# Self-powered triboelectric-responsive microneedles with controllable release of optogenetically engineered extracellular vesicles for intervertebral disc degeneration repair

Weifeng Zhang[1,3], Xuan Qin[2,3], Gaocai Li [1,3], Xingyu Zhou[1,3], Hongyang Li[2], Di Wu[1], Yu Song [1], Kangcheng Zhao[1], Kun Wang[1], Xiaobo Feng[1], Lei Tan[1], Bingjin Wang[1] ✉, Xuhui Sun [2] ✉, Zhen Wen [2] ✉ & Cao Yang [1] ✉

Excessive exercise is an etiological factor of intervertebral disc degeneration (IVDD). Engineered extracellular vesicles (EVs) exhibit excellent therapeutic potential for disease-modifying treatments. Herein, we fabricate an exercise self-powered triboelectric-responsive microneedle (MN) assay with the sustainable release of optogenetically engineered EVs for IVDD repair. Mechanically, exercise promotes cytosolic DNA sensing-mediated inflammatory activation in senescent nucleus pulposus (NP) cells (the master cell population for IVD homeostasis maintenance), which accelerates IVDD. TREX1 serves as a crucial nuclease, and disassembly of TRAM1-TREX1 complex disrupts the subcellular localization of TREX1, triggering TREX1-dependent genomic DNA damage during NP cell senescence. Optogenetically engineered EVs deliver TRAM1 protein into senescent NP cells, which effectively reconstructs the elimination function of TREX1. Triboelectric nanogenerator (TENG) harvests mechanical energy and triggers the controllable release of engineered EVs. Notably, an optogenetically engineered EV-based targeting treatment strategy is used for the treatment of IVDD, showing promising clinical potential for the treatment of degeneration-associated disorders.

Low back pain (LBP) is the most prevalent musculoskeletal disorder and the leading cause of disability worldwide, posing a severe socio-economic burden and decreasing quality of life[1-3]. LBP spans a spectrum of different types of pain anatomically extending from the 12th rib to the iliac crest, with the characteristics of recurrence, refractoriness, and unsatisfactory patient outcomes[1,3]. The degeneration and herniation of intervertebral discs (IVDs), a highly prevalent affliction of aging individuals, are important causative factors of LBP[1,4,5]. IVDs are avascular fibrocartilaginous joints situated between adjacent vertebral bodies with anatomical regional structures containing a central hydrogel-like nucleus pulposus (NP), a peripherally concentrically arranged annulus fibrosus (AF), and hyaline cartilaginous endplates

[1]Department of Orthopedics, Union Hospital, Tongji Medical College, Huazhong University of Science and Technology, Wuhan, China. [2]Institute of Functional Nano and Soft Materials (FUNSOM), Jiangsu Key Laboratory for Carbon-Based Functional Materials and Devices, Soochow University, Suzhou, China. [3]These authors contributed equally: Weifeng Zhang, Xuan Qin, Gaocai Li, Xingyu Zhou. ✉e-mail: wangbingjin@hust.edu.cn; xhsun@suda.edu.cn; wenzhen2011@suda.edu.cn; caoyangunion@hust.edu.cn

(CEPs) lying between the vertebra and the disc[4,6]. Given that exercise is a powerful driver that can enhance neurogenesis, angiogenesis, myogenesis, and osteogenesis, it is increasingly promoted for the prevention and treatment of chronic degenerative diseases to alleviate symptoms and postpone progression[7–11]. Nevertheless, excessive exercise has damaging effects on musculoskeletal systems by disrupting biomechanical homeostasis, especially in individuals with potential musculoskeletal abnormalities, including dysplasia and postural dysfunction[12–14]. In addition, efficient biolubrication on synovial joint surfaces is essential for continued physical activity, but excessive exercise destroys biolubrication and remodels the biomechanical interface, leading to inflammation-associated pain and friction-related musculoskeletal disorders[13]. Persistent abnormal biomechanical loading during physical activity serves as an etiological initiating event of IVD degeneration (IVDD), accompanied by the acquisition of a cellular inflammatory phenotype and the destruction of biomechanical integrity[12,14]. Therefore, a sustainable strategy requires balancing the effects of exercise by enhancing beneficial functions and decreasing deleterious functions.

Globally, a large gap persists between the evidence and clinical practice involved in the therapeutic treatment of IVDD[15]. Currently, no approved medical therapies are available, and pharmacological interventions tend to alleviate pain-related symptoms rather than reverse or postpone degenerative processes but are accompanied by extensive medical consultation[4]. Tissue regenerative engineering offers promising drug carriers, including liposomes, ceramic nanoparticles, and polymer nanoparticles; however, investigations of the long-term efficacy and safety of these tissue regenerative strategies, which have a weak capacity for sustained drug release or poor biodegradability, are lacking[16–21]. Cell-derived extracellular vesicles (EVs) serve as vectors of cell-cell communication via the transfer of member-encapsulated cargoes to recipient cells to modulate the phenotype of recipient cells[22–24]. Cell-derived EVs, which have the ability to bypass biological barriers and deliver active molecular cargoes, are suitable for development as a potential therapeutic strategy for IVDD[25,26]. Cell-derived EVs potentially mimic the therapeutic effects of cells, yet native EVs lacking the desired cargoes have a poor ability to target master regulators involved in the degenerative process of IVDs[25]. An excellent delivery system for targeting EVs to degenerative sites and responsive triggers for the sustainable release of controllable EVs are key considerations when designing EV-based therapeutic strategies to minimize off-target effects and enhance therapeutic efficiency[25].

Microneedles (MNs) are an array of microscale needle projections that are promising for minimally invasive transdermal drug delivery and have the capacity to penetrate the stratum corneum and communicate with the epidermis/dermis via theranostic nanoparticles or signals[27–29]. With the prominent advantages of high penetration ability, minimal invasiveness, self-administration and adjustable capacity, MNs exhibit excellent ability to deliver transdermally administered drugs either as coated or encapsulated cargo during the insertion process or via convective effects in the field of macromolecule distribution and vaccine self-administration[27,28,30,31]. Wearable stimulation-responsive MNs that respond to physiological factors (biomolecular and chemical signals) or external physical stimuli (high-energy photons and magnetic field) expand the potential of controllable drug release for "site-to-site" treatment with high efficiency and excellent utilization, resulting in the current trend in drug and gene delivery[20,27,32–36]. The concept of "etiology drives treatment" focuses on the treatment of disease-modifying targets during the IVDD process with the goal of slowing the degenerative process. Therefore, the integration of etiology-related stimulation and controllable drug release in MN array is a promising strategy for disease-targeted treatment.

Triboelectric nanogenerators (TENGs) are a state-of-the-art technology for converting ambient mechanical energy to triboelectric energy based on the combination of triboelectrification and electrostatic induction[37]. By periodic relational contact and separation between dissimilar frictional materials with different electron affinities, TENGs efficiently harvest irregular and low-frequency human biomechanical energy derived from body motion, breathing, heart beating, and pressure differentials into electrical energy[38–43]. With the advantages of low cost, light weight, structural flexibility and efficient energy conversion, TENGs are widely and increasingly used for health monitoring, minimally invasive diagnosis, and individual disease therapeutics[44–48]. TENGs exhibit excellent bioelectric effects to modulate cell activities involved in functional regulation and fate determination[49,50]. Given the lack of evidence that NP tissues or residual NP cells have the capacity for electroreactivity, it is impractical to directly integrate the electrical stimulation driven by TENGs with the phenotypic alteration of damaged NP cells. However, exercise-induced triboelectric energy is promising as a responsive trigger to drive the electrically controllable release of active molecules for targeted IVDD-modifying treatment.

In this study, we fabricated an exercise self-powered triboelectric-responsive MN array for the controllable release of optogenetically engineered (EXPLOR) EVs for IVDD repair via an inflammatory regulation strategy (Fig. 1A). Exercise accelerated the degenerative process of IVDs from lumbar surgical instability (LSI), which is involved in cytosolic DNA sensing-mediated inflammatory activation, similar to the degenerative patterns of the clinical IVDD process (Fig. 1B). TREX1 is a crucial nuclease with dual functions: physiologically, it participates in cytosolic DNA elimination, and pathologically, it causes aberrant nuclear genomic DNA damage. Disassembly of TRAM1-TREX1 disrupted the localization of TREX1 in the endoplasmic reticulum (ER) (Fig. 1B). Dislocation of TREX1 results in a nuclease effect that leads to nuclear genomic DNA damage (Fig. 1B). EXPLOR technology applies the concept of optically reversible protein-protein interactions to load TRAM1 protein into EVs (Fig. 1B)[51,52]. Self-powered triboelectric-responsive MNs, using the triboelectric effect from the relational motion of dissimilar frictional layers (ITO and PTFE) and the changeable electrostatic adsorption characteristic of polypyrrole (PPy) exposed to triboelectric stimulation, could harvest mechanical energy to drive TRAM1-engineered EV release, block TREX1 nuclear localization and anchor TREX1 to the ER, thereby facilitating immune surveillance for the elimination of damaged DNA (Fig. 1B, C). When the motion frequency of the linear motor varied from 0.5 to 2.5 Hz, the open-circuit voltage ($V_{oc}$) remained almost constant at -255 V, and the transferred charge ($Q_{tr}$) was also maintained at -80 nC, while the short-circuit current ($I_{sc}$) increased apparently from -3 to -16 μA (Fig. 1E). An integrated self-powered triboelectric-responsive MN array with wearable portability and excellent triboelectric characteristics effectively alleviated the degeneration of IVDs and provided a promising repair strategy for exercise-associated disorders (Fig. 1D, E).

## Results

### Exercise accelerated the degeneration of lumbar IVDs from surgical lumbar instability, similar to the clinical degeneration process

Mechanical stresses, including compulsion position and aberrant mechanical loading, are regarded as etiological factors of IVDD, and to further investigate the effect of mechanical exercise during the degeneration of IVDs, we surgically resected the lumbar 4th–lumbar 5th (L4-L5) spinous processes of the rat spine along with the supraspinous and interspinous ligaments to induce LSI, ensuring that the surgical operation did not interfere with the exercise ability of the LSI rats (Fig. 2A) (Supplementary Fig. 1A, B). Surprisingly, X-ray and microcomputed tomography (μCT) showed no significant differences in the L3-6 vertebra, subchondral bone or IVD height between LSI rats with or without exercise 8 weeks after the surgery (Fig. 2B). Moreover, magnetic resonance imaging (MRI) revealed that the T2-weighted

signals in the L3-L4, L4-L5, and L5-L6 IVDs of exercising LSI rats (LSI-E) clearly became heterogeneous, suggesting reduced water content and degenerated phenotype (Fig. 2B, E). Histologically, hematoxylin and eosin (H&E) staining and safranin O-fast green (SO&FG) staining of L4-L5 IVDs revealed that NP cells of LSI-E rats lost vacuolated or stellar-shaped morphology accompanied by cellular chondrogenic proliferation at the interface of the NP and AF (Fig. 2C, D; Supplementary Fig. 1C). The collagen lamellae of the AF bulged inwards, and the border between the NP and AF was nebulous, indicating fibrosis of the IVDs with the loss of well-organized NP and AF regional structures (Fig. 2C, D, F; Supplementary Fig. 1C). Together, MRI-based radiological results and histological results verified that rats with LSI that ran on a running treadmill exhibited accelerated degeneration of lumbar IVDs.

To further describe the degeneration-associated molecular atlas of the LSI model during exercise, L4-L5 IVDs from the sham operation, LSI and LSI-E groups were collected and subjected to RNA-sequencing analysis (RNA-seq) (Fig. 2A). Gene Ontology (GO) analysis revealed that inflammation- or immune activation-associated pathways, including

"Regulation of immune system process", "Regulation of cytokine production", "Positive regulation of immune system process", and "Inflammatory response" were substantially enriched in the differentially expressed genes (DEGs) between the LSI-E group and sham operation group and between the LSI-E group and LSI group (Fig. 2G; Supplementary Fig. 2A). Similarly, Kyoto Encyclopedia of Genes and Genomes (KEGG) analysis indicated activation of the NF-κB signaling pathway and osteoclast differentiation during exercise-related degeneration (Supplementary Fig. 2B). Furthermore, gene set enrichment analysis (GSEA) demonstrated that activation of the inflammatory response accompanied by immune mobilization was associated with multiple immune effectors (Supplementary Fig. 2C). Notably, the "cytosolic DNA-sensing pathway", which acts as a critical immune-damaging pathway and contributes to a severe cytokine storm via the NF-κB signaling pathway after being triggered by excessive accumulation of damaged DNA during numerous inflammatory degenerative disorders, was significantly more strongly enriched in the LSI-E group than the sham operation group or LSI group (Fig. 2H, I) (Supplementary Fig. 2D). A heatmap revealed that the expression of genes involved

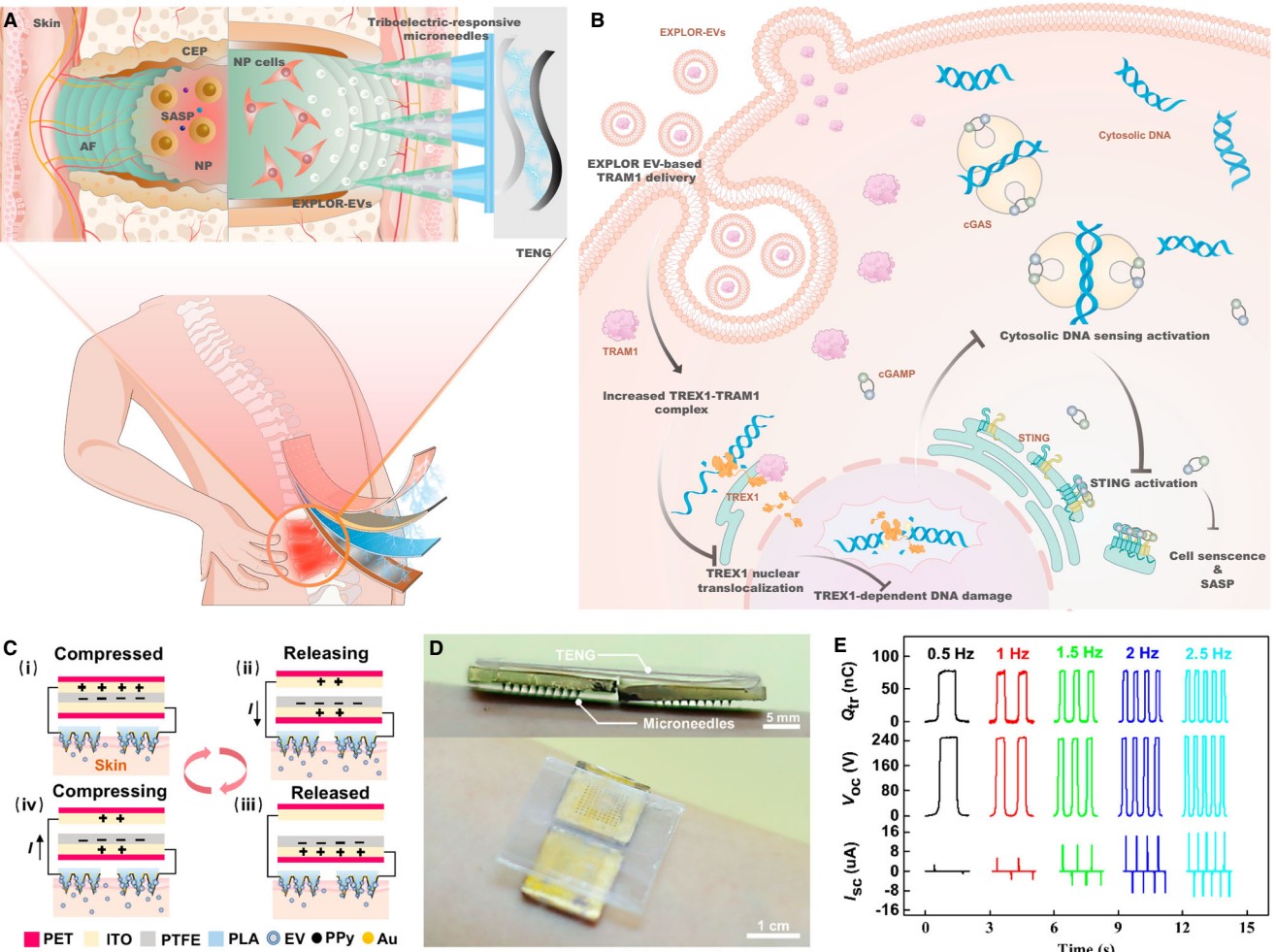

**Fig. 1 | Self-powered triboelectric-responsive MNs with controllable release of EXPLOR engineered EV for IVDD repair. A** Self-powered triboelectric-responsive EXPLOR engineered EV release for biologically targeted IVDD treatment via optically reversible protein-protein interactions. **B** EXPLOR engineered EV-delivered TRAM1 protein increased TRAM1-TREX1 complex assembly, blocking TREX1 nuclear localization and promoting TREX1 anchoring in the ER, which exhibited protective effects for cytosolic damaged DNA elimination, inhibited cGAS-STING axis activation-mediated inflammatory response and alleviated the progression of IVDD. **C** Schematic of EXPLOR engineered EV loading and release of triboelectric-responsive MNs using the electrochemical characteristics of polypyrrole (PPy). Polytetrafluoroethylene (PTFE) and indium tin oxide-polyester (ITO-PET) acted as two dissimilar frictional layers. Polylactic acid (PLA) and the Aurum (Au) layer acted as triboelectric-responsive MNs for controllable release of EXPLOR engineered EVs. **D** Wearable self-powered triboelectric-responsive MNs for controllable release of EXPLOR engineered EVs. **E** Electrical output under various motion frequencies ranging from 0.5 to 2.5 Hz, including $V_{oc}$, $I_{sc}$, and $Q_{tr}$ (Representative plot of three independent technical experiments).

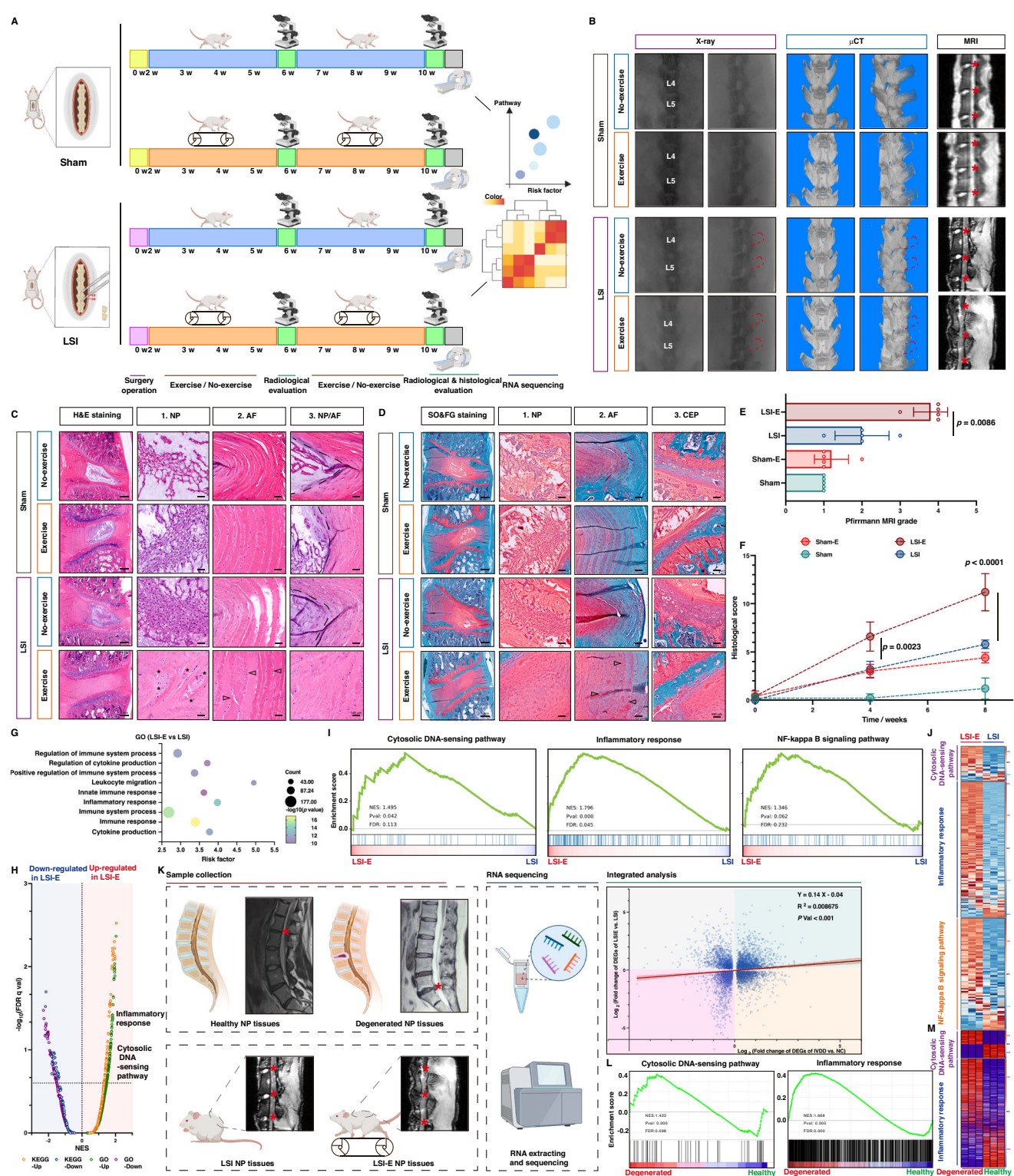

in the "Cytosolic DNA-sensing pathway", "Inflammatory response", and "NF-κB signaling pathway" was upregulated in the transcriptome of the LSI-E group (Fig. 2J) (Supplementary Fig. 2E). These transcriptomic maps indicated that cytosolic DNA-sensing-mediated inflammatory activation plays a damaging role during exercise-related degenerative processes.

To explore the potential correlation between exercise-induced degenerative process in the LSI model and clinical IVD degenerative process, healthy and degenerated NP tissue samples were collected from volunteers with idiopathic scoliosis and lumbar intervertebral

herniation who underwent orthopedic surgeries (Fig. 2K). Subsequently, RNA-seq analysis was used to construct a degeneration-associated molecular atlas, and importantly, correlation analysis of DEGs between degenerated and no-degenerated patients and between LSI-E and LSI rats revealed that exercise-induced degeneration of IVDs from surgically induced lumbar instability was similar to that of clinical degenerative process, with a positive correlation coefficient ($0.14 \pm 0.04$, $R^2 = 0.0087$) (Fig. 2K). "Cytosolic DNA-sensing pathway" and "Inflammatory response" were significantly enriched in clinically degenerated samples, and these molecular pathways were also

**Fig. 2 | Exercise accelerated the degenerative process of IVDs from surgical lumbar instability, similar to the clinical degeneration process. A** Schematic workflow for the establishment of a surgically LSI rat model and exercise intervention (*n* = 5 rats per group) (Created with BioRender.com). **B** X-ray, μCT, and MR images of the lumbar from rats undergoing sham surgery (Sham) or LSI surgery with (LSI-E) or without exercise intervention (LSI) for 8 weeks. **C** H&E staining and **D** SO&FG staining of lumbar 4th–lumbar 5th (L4-L5) IVDs from rats with the indicated treatment. Asterisks showed the large, round cells and cellular chondrogenic proliferation; triangles showed inwards bulging, thickened AF collagen lamellae and dashed lines showed the border between the NP and AF. Bar: 1 mm, 100 μm. **E** Pfirrmann MRI grading of rat IVDs with the indicated treatment for 8 weeks (*n* = 5 rats per group). **F** Histological score of rat IVDs with the indicated treatment for 8 weeks (*n* = 5 rats per group). **G** Upregulated inflammation-associated pathways by Gene Oncology (GO) analysis of differentially expressed genes (DEGs) between the LSI-E group and LSI group (*n* = 3 biological independent samples). **H, I** Enriched pathways by GSEA (Gene set enrichment analysis) between the LSI-E group and LSI group (*n* = 3 biological independent samples). **J** Gene expression heatmaps in the LSI-E and LSI group (*n* = 3 biological independent samples). **K** Schematic workflow for RNA sequencing (RNA-seq) of clinical normal and degenerated NP tissue samples and correlation analysis of overlapping DEGs between LSI-E group and LSI group and between degenerated and normal NP samples (*n* = 3 biological independent samples) (Created with BioRender.com). **L** Enriched pathways by GSEA in degenerated NP tissues (*n* = 3 biological independent samples). **M** Gene expression heatmaps of enriched pathways in normal and degenerated NP tissues (*n* = 3 biological independent samples). A significant *p* value was determined by two-tailed ANOVA (**F**) and Pearson's correlation analysis (**K**). Mean ± SD are shown for (**F**).

involved in the exercise-induced degenerative process of IVDs, suggesting that the cytosolic DNA-sensing pathway participates in crucial immune surveillance for inflammatory activation during the IVDD process (Fig. 2L, M).

## Disassembly of the TRAM1-TREX1 complex disrupted the anchoring of TREX1 to the ER and triggered TREX1-dependent nuclear genomic DNA damage during NP cell senescence

The senescence of resident NP cells is fundamental for the initiation and progression of IVDD[4,53–55]. The acquisition of senescent phenotype in NP cells initiates damage-associated responses and disrupts the balance between anabolism and catabolism of the extracellular matrix, leading to the dehydration of NP tissue accompanied by the disorganization of AF collagen lamellae. The destruction of IVD integrity triggers an inflammatory response and mechanically compresses neighboring structures, contributing to discogenic pain and activity limitations[4,53]. NP cellular senescence accompanied by inflammatory mediator secretion occurs in response to extracellular matrix anabolic-catabolic imbalance and tissue biomechanical homeostasis destruction during IVDD, which was also obviously upregulated in the LSI-E group and clinical degenerative samples according to RNA-seq GSEA (Fig. 3A) (Supplementary Fig. 3A). cGAS-mediated sensing of aberrant cytoplasmic DNA is responsible for triggering the inflammatory response and promoting inflammatory mediator secretion during the cellular senescence process, defined as the senescence-associated secretion phenotype (SASP)[56,57]. To further evaluate the critical role of cytosolic DNA-sensing-mediated inflammatory activation in NP cell inflammatory senescence, we generated a senescence-associated NP cell model by serial passage and predictably found that compared with NP cells from the second passage, NP cells from the eighth passage exhibited a senescence status, characterized by increased cell cycle arrest-associated markers phosphorylated p53 (p-p53), p21, and p16, enhanced senescence-associated β-galactosidase (SA-β-Gal) activity, HP1γ-heterochromatin foci formation, cellular swelling morphology and hyperactivation of senescence-associated secretome (Fig. 3B, C; Supplementary Fig. 3B, C). The levels of cytosolic accumulated DNA, a genomic DNA damage marker (γH$_2$A) and cGAS-STING axis-related effectors were drastically increased in senescent NP cells (Fig. 3D–F). In addition, GSEA revealed that "STING-mediated induction of host immune response" was enriched during the degenerative process of IVDs (Supplementary Fig. 3D). Overall, senescent NP cells exhibited excessive cytosolic DNA accumulation and aberrant cGAS-STING axis activation. cGAS-STING axis activation causes inflammatory damage, whereas TREX1 (three-prime repair exonuclease 1) serves as an ER-associated exonuclease that protects cells from excessive cytosolic accumulated DNA fragments and may be a promising therapeutic target for blocking the activation of the inflammatory response during IVDD. However, unexpectedly, no marked changes in TREX1 expression were observed between normal and senescent NP cells (Fig. 3G). TREX1 anchors to the ER membrane via a carboxyl (C)-terminal single-pass transmembrane helix, and the amino (N)-terminal nuclease

domain is located in the cytoplasm. The ER tethering of TREX1 is critical for its nucleolytic activity, and frameshift mutations of the C-terminal domain compromise the association between TREX1 and the ER with no disturbance of the catalytic activity site, which contributes to aberrant activation of cytosolic DNA-sensing pathways and abnormal phenotypes of autoimmunity[56,58,59]. We wondered whether TREX1 localization plays a critical role in the senescence process of NP cells, and immunofluorescence (IF) staining revealed that TREX1 spread across the cytoplasm and nucleus in senescent cells, but accumulated mainly in the cytoplasm in normal cells (Fig. 3J). Subsequently, western blot analysis of cytosolic and nuclear components isolated from normal and senescent NP cells revealed that the ratio of nuclear TREX1 to cytosolic TREX1 was drastically increased in senescent NP cells (Fig. 3H, I). The destruction of nuclear membrane integrity is a characteristic of cell senescence that allows TREX1 to be transferred to the inner nuclear membrane networks and gain access to chromatin, leading to TREX1-dependent nuclear DNA damage[59]. In addition, the administration of TREX1 wild-type (TREX1-WT) plasmids increased the level of genomic DNA damage in senescent NP cells, whereas the N-terminus of the TREX1 construct lacking a C-terminal ER-associated site and the TREX1 D18N (18th amino acid residue replaced aspartate with asparagine) mutant with dysfunctional DNA-degrading activity were able to partially rescue the increased level of DNA damage and cytosolic accumulated genomic DNA (Fig. 3K–O). Together, the results revealed that TREX1 detachment from the ER and translocating to the nucleus were responsible for the immune elimination dysfunction of TREX1 and aberrant activation of the cytosolic DNA-sensing pathway involved in inflammatory NP cellular senescence.

To further explore the potential mechanism that triggers TREX1 detachment from ER and nuclear translocation, we performed immunoprecipitation (IP) of endogenous TREX1 protein from normal and senescent NP cells, and mass spectrometry (IP-MS) analysis of proteins pulled down by using an anti-TREX1-specific antibody revealed 413 potential proteins that interacted with endogenous TREX1 in normal NP cells and 30 candidates that interacted with endogenous TREX1 in senescent NP cells (Fig. 4A). Notably, a KEGG analysis of differential interactome of TREX1 revealed that the "Protein processing in ER" pathway was significantly enriched in normal but not in senescent NP cells, which indicated that processing in the ER may be critical for the protective effect of TREX1 against aberrant activation of the cytosolic DNA-sensing pathway and TREX1-dependent genomic DNA damage (Fig. 4B; Supplementary Fig. 3E, F). A comparative analysis of the TREX1 interactomic data and ER localization-associated molecular pathways involved in "Protein processing in ER" and "Protein localization in ER" suggested that TRAM1 was the only potential protein interacting with TREX1 and thus could be responsible for regulating TREX1 localization in the ER (Fig. 4C; Supplementary Fig. 3E). Translocating chain-associating membrane protein 1 (TRAM1) is a multipass membrane protein in the ER that is sufficient for the translocation of several secretory

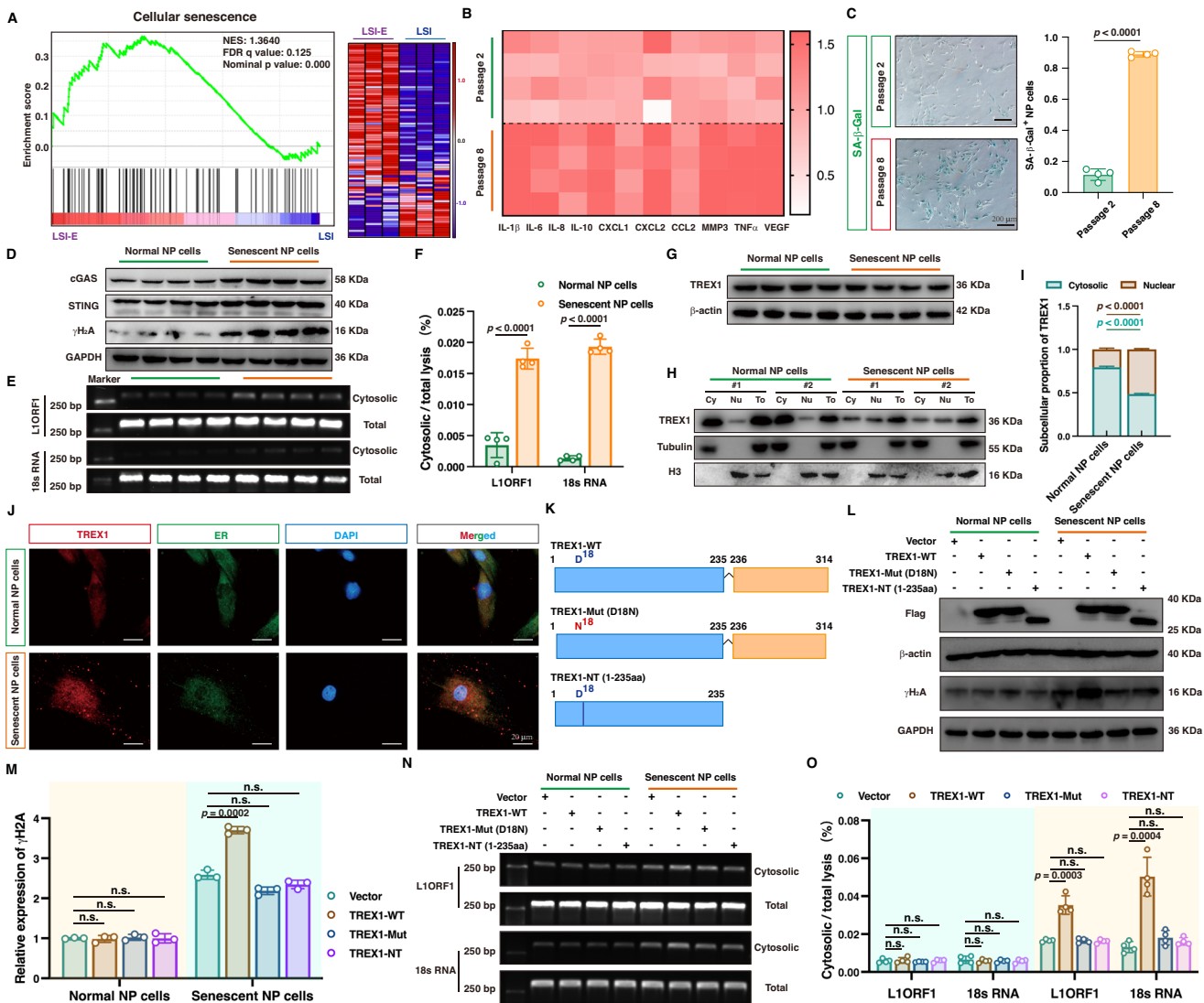

**Fig. 3 | TREX1 detachment from the ER and translocating to the nucleus were responsible for the elimination dysfunction of TREX1 and aberrant activation of the cytosolic DNA-sensing pathway. A** Upregulated "Cellular senescence" pathways and expression heatmaps between LSI and LSI-E group ($n = 3$ biological independent samples). **B** Expression heatmaps of senescence-associated secretory phenotype (SASP) genes from NP cells ($n = 4$ biological independent experiments). **C** Senescence-associated β-galactosidase (SA-β-gal) staining and quantitative analysis in NP cells ($n = 4$ biological independent experiments). **D** Representative western blotting images of cGAS, STING, and $\gamma H_2A$ in NP cells ($n = 4$ biological independent experiments). **E, F** Representative DNA electrophoresis images (**E**) and quantitative analysis (**F**) of cytosolic DNA fragments in NP cells ($n = 4$ biological independent experiments). **G** Representative western blotting images of TREX1 in NP cells ($n = 4$ biological independent experiments). **H, I** Representative western blotting images of (**H**) and quantitative analysis (**I**) of cytosolic, nuclear, or total TREX1 protein in NP cells ($n = 4$ biological independent experiments). **J** Representative IF staining of TREX1 in NP cells, bar: 20 μm (Representative image

of three independent technical experiments). **K** Schematic illustration of Flag-tagged TREX1 wide-type (TREX1-WT), Flag-tagged TREX1 D18N mutant (TREX1-Mut, 18th amino acid residue replaced aspartate with asparagine) and Flag-tagged N-terminus of TREX1 construct lacking a C-terminal ER-associated site (TREX1-NT, the anime-terminus from the 1st to 235th amino acids). **L** Representative western blotting images of Flag-tagged TREX1 variants and $\gamma H_2A$ in NP cells after transfected with vector and Flag-tagged TREX1 variant plasmids (Representative blot of four independent technical experiments). **M** Quantitative analysis of western blotting results showing $\gamma H_2A$ protein expression in NP cells (Quantification of three independent technical experiments). **N, O** Representative agarose gel electrophoresis images (**N**) and quantitative analysis (**O**) of cytosolic DNA fragments in NP cells (Representative blot of four independent technical experiments, and quantification of four independent technical experiments). A significant $p$ value was determined by two-tailed unpaired $t$ test (**C, F, I**) and two-tailed ANOVA (**N, O**). Mean ± SD are shown for (**C, F, M, O**). n.s. not significant.

proteins, ER-membrane integration and the transport of misfolded ER-membrane proteins from the ER[60–62]. Transient transfection of HEK293T cells with Flag-tagged TREX1 and His-tagged TRAM1 followed by bidirectional Co-IP assays demonstrated that exogenous TREX1 interacted with TRAM1 (Supplementary Fig. 3G). Subsequently, bidirectional Co-IP experiments showed that the physical interaction between TREX1 and TRAM1 was reduced in senescent NP cells (Fig. 4D). To further examine the role of decreased TREX1-TRAM1 complex assembly during the senescence process in NP cells,

TRAM1 was genetically silenced by two specific small interfering RNAs (siRNAs) in normal NP cells (Supplementary Fig. 4A, B). TRAM1 knockdown significantly promoted the acquisition of the senescence phenotype in NP cells, accompanied by enhanced nuclear localization of TREX1 and aberrant activation of the cGAS-STING axis (Supplementary Fig. 4C–H). Therefore, in TRAM1-deficient NP cells, maintaining TREX1 localization in the ER and ensuring the protective effect of cytosolic exonuclease are critical strategies for alleviating NP cell senescence. In contrast, the genetic overexpression of TRAM1

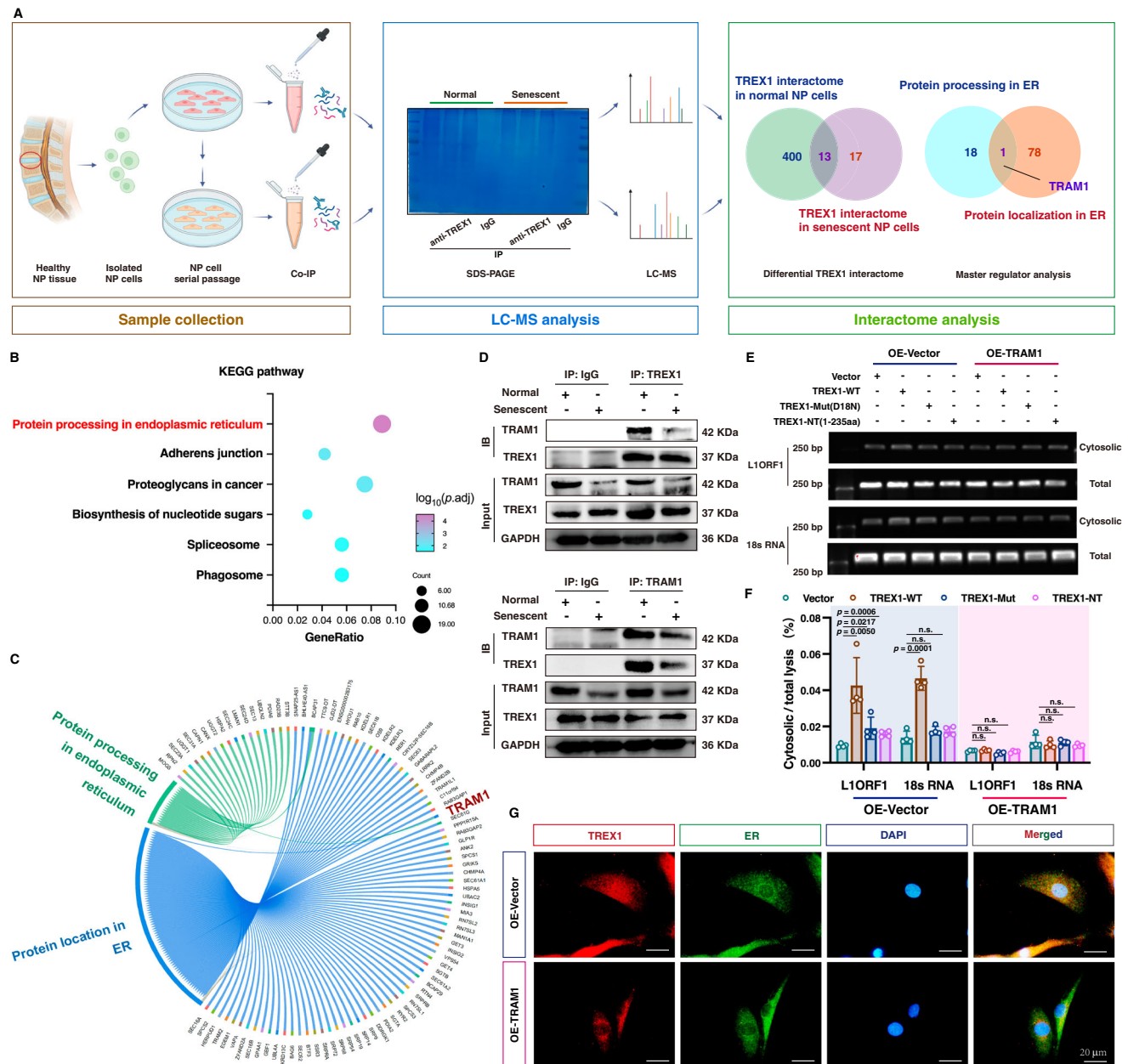

**Fig. 4 | Interactomic analysis revealed that disassembly of the TRAM1-TREX1 complex promoted TREX1 mislocalization and activation of cytosolic DNA-sensing pathway during NP cell senescence. A** Schematic workflow for TREX1 interactomic analysis via Co-IP and LC-MS using a specific anti-TREX1 antibody or control antibody in normal and senescent NP cells ($n = 3$ biological independent samples) (Created with BioRender.com). **B** Top 6 enriched KEGG pathways of the differential TREX1 interactome in normal NP cells and red-marked pathways were associated with protein processing in the ER. **C** Chordal graph of potential proteins in KEGG pathways involved "Protein location in ER" and "Protein processing in ER". **D** Endogenous forward and reverse Co-IP assay to detect the interaction of TRAM1 and TREX1 in normal and senescent NP cells (Representative blot of three independent technical experiments). **E** Representative agarose gel electrophoresis

images of DNA fragments from the cytosolic component in P8 NP cells after co-transfection with TRAM1-overexpressing plasmid and different Flag-tagged TREX1 variant plasmids (Representative blot of four independent technical experiments). **F** Quantitative analysis of cytosolic genomic DNA fragments in P8 NP cells after co-transfection with the TRAM1-overexpressing plasmid and different Flag-tagged TREX1 variant plasmids (Quantification of four independent technical experiments). **G** Representative IF staining images of TREX1 to show subcellular localization changes of TREX1 after TRAM1 overexpression (Representative image of three independent technical experiments), bar: 20 μm. A significant $p$ value was determined by two-tailed ANOVA (**F**). Mean ± SD are shown for (**F**). n.s. not significant.

via the exogenous transfection of plasmids into senescent NP cells clearly blocked the translocation of TREX1 to the nucleus, inhibited the activation of the genomic DNA damage-dependent cGAS-STING inflammatory response, and ultimately alleviated the senescence status of NP cells by mimicking the senotherapeutic effects of increased assembly of the TREX1-TRAM1 complex (Fig. 4E–G) (Supplementary Fig. 5A–H). Overall, loss- and gain-of-function assays

revealed that disassembly of the TREX1-TRAM1 complex disrupted the translocation of TREX1, resulting to TREX1 destroying genomic integrity via its exonuclease activity, and TRAM1 was essential for the functional regulation of TREX1 tethering to the ER or translocation to the nucleus, which is involved in immune elimination or immune damage. Overall, TRAM1 might be a promising target for the inhibition of the inflammatory response during progressive IVDD.

## EXPLOR-engineered EVs alleviated the acquisition of inflammatory senescence phenotype in NP cells via the delivery of TRAM1

An appropriately selective vector is necessary for an excellent gene and drug delivery system to deliver target molecules stably to a specific site. EVs, with their strong biocompatibility, minimal immunogenicity, and intrinsic capacity to target tissues, perform a critical function in mediating cell-cell communication by transferring active molecules to recipient cells, which has been highlighted as a promising strategy for the delivery of proteins and nucleotides in disease-targeting therapeutics[22,25]. EXPLOR technology, defined as optogenetically controlled engineered EV biogenesis, integrates optically reversible protein-protein interactions with targeting proteins for EV loading and can effectively deliver soluble proteins into the recipient cytoplasm[51,52,63,64]. We successfully utilized optical illumination-induced reversible conjugation of two photoreactive binding proteins, CRY2 and CIBN, fused with either TRAM1 (CRY2-conjugated mCherry-tagged TRAM1) or the EV surface marker CD9 (CIBN-conjugated EGFP-tagged CD9) to load the TRAM1 protein into EVs via the localization of CD9 to the inner surface of early endosomes (Fig. 5A, B). Western blotting analysis was performed to confirm the packaging of TRAM1 into engineered EVs, accompanied by the expression of the EV-associated markers Alix, TSG101 and CD63, which was expectedly absent in native EVs (Fig. 5C). Transmission electron microscopy (TEM) revealed that TRAM1-engineered EVs and native EVs had a prominent cup-shaped morphology (Fig. 5D). In addition, nanoparticle tracking analysis (NTA) revealed that engineered EVs had a particle size ranging from 27.3 to 468 nm with a mean size of 149 nm; similarly, native EVs were between 14.8 nm and 647 nm in size with a mean size of 147.5 nm, which indicated no significant changes in morphology or size between TRAM1-engineered EVs and native EVs (Fig. 5D). To evaluate the therapeutic effects of TRAM1-loaded EVs against inflammatory senescence, we incubated senescent NP cells with engineered or native EVs labeled with PKH26 and DiO, and flow cytometry confirmed that NP cells were able to efficiently take up both types of EVs without any particular preference (Fig. 5F). In addition, IF staining revealed that the green fluorescence of EGFP-conjugated CD9 spread across the cell membrane and that the red fluorescence of mCherry-conjugated TRAM1 accumulated in the cytoplasm (Fig. 5E). Western blotting analysis further verified that the mCherry-conjugated TRAM1 protein was efficiently delivered to the NP cell cytosol via EXPLOR-engineered EVs (Supplementary Fig. 6A). Interestingly, incubation of senescent NP cells with TRAM1-engineered EVs drastically decreased the levels of cell cycle arrest-associated effectors and genomic DNA damage markers, inhibited the enzymatic activity of SA-β-Gal and reduced the secretion of proinflammatory factors (Supplementary Fig. 6B–D). Furthermore, the administration of TRAM1-engineered EVs alleviated the nuclear translocation of TREX1, which confirmed the protective effect of TREX1 (Fig. 5G, H). More importantly, the amount of damaged genomic DNA in the cytosol was notably reduced, and the cytosolic DNA sensing cGAS-STING pathway was also suppressed after TRAM1-engineered EV administration (Fig. 5I; Supplementary Fig. 6E, F). In summary, TRAM1-engineered EVs can deliver TRAM1 to the NP cell cytoplasm and regulate TREX1 to exert a protective effect by blocking cytosolic DNA-cGAS-STING pathway activation.

## Self-powered triboelectric-responsive MNs had excellent mechanical and electrical characteristics

After chemical and electrochemical doping, polypyrrole (PPy) changes from an insulator to a conductive polymer with extensively conjugated π bonds[65–67]. Due to their advantages of low polymerization potential, high conductivity and suitable biocompatibility, PPy films formed by electrochemical polymerization have been widely studied in the fields of electrocatalysis, biosensors and controlled drug release. PPy will undergo a continuous oxidation reaction during electrochemical polymerization and have a positive charge, and anionic biomolecules can be doped into the skeleton of PPy by electrostatic action to balance the positive charge of PPy and achieve the loading of anionic biomolecules[66]. When PPy is reduced to a zero-valence state, it loses the ability to adsorb anionic biomolecules, resulting in the controlled release of anionic biomolecules[66]. EVs have a lipid bilayer with negative charges on their surface[68,69]. Therefore, to realize the controlled release of TRAM1-engineered EVs and improve their release efficiency at degenerative intervertebral discs, we proposed the preparation of self-powered triboelectric-responsive MNs based on the electrochemical characteristics of PPy (Fig. 1C, D) (Supplementary Fig. 7A).

First, a wearable TENG was designed to provide pulsed electric fields by converting mechanical energy into electrical energy[37,39]. Supplementary Fig. 7A shows the structure of the TENG, which includes a triboelectric layer and two electrodes. Polytetrafluoroethylene (PTFE) was used as a triboelectric layer, and indium tin oxide-polyester (ITO-PET) films were used as electrodes according to the working mechanism of "contact-separation mode". Here, we used ITO as the positive tribomaterial and PTFE as the negative tribomaterial due to their different electronegativities. Initially, the PTFE film contacted the ITO electrode, causing charge transfer from the ITO to the PTFE film, and accordingly, the PTFE was negatively charged (State i). When the PTFE film began to move away, the surface of the bottom ITO electrode continuously generated a positive charge due to electrostatic induction and created a potential difference between the two ITO electrodes, resulting in a current signal in the electric circuit (State ii). The subsequent contact process between two charged films generated a reverse current with an alternating current output during periodic contact-separation movements (State iii and iv) (Fig. 1C). To evaluate the output performance of the TENG, the TENG was connected to series with external load resistors. The peak power density increased with increasing frequency and reached a maximum value of 500 mW/m² at 2.5 Hz (Supplementary Fig. 7B). When the frequency was constant, the output voltage increased, and the output current decreased with increasing resistance to external loads. (Supplementary Fig. 7C). Since the TENG was small and lightweight, we tested the open-circuit voltage of it when the device was worn on different parts of the body to verify the usefulness of the TENG in the wearable electronics field. When the TENG was attached to the chest, it could generate a 3 V voltage upon breathing. Clenching the fists caused the TENG at the arm to generate an open-circuit voltage of approximately 5 V. Upon shaking or bending the arm, the TENG at the arm generated an open-circuit voltage of approximately 25 V (Supplementary Fig. 7D).

The preparation process of triboelectric-responsive MNs is shown in Fig. 5J. First, a silicone rubber template with conical holes was fabricated by the laser engraving method with great consistency. To improve the biocompatibility of the MNs, polylactic acid (PLA) particles were placed in the silicone rubber template and heated at 200°C for approximately one hour. As a result, the PLA particles fully melted and then became the substrate of the MNs at room temperature (Supplementary Fig. 8B (i)). The height of the PLA MN was approximately 600 μm, while the diameter was approximately 300 μm. Then, highly conductive PLA-Au MNs were fabricated by sputtering gold on the surface of the PLA MNs (Supplementary Fig. 8B (iii)). Electropolymerization of polypyrrole was performed with an electrochemical workstation at a constant current density of 2 mA/cm², and the current was very stable throughout the deposition process (Supplementary Fig. 8A). During the deposition process, PLA-Au MNs were used as working electrodes, platinum electrodes were used as counter electrodes, and saturated calomel electrodes were used as reference electrodes. EVs and PPy were successfully loaded on the surface of the PLA-Au MNs and appeared black (Supplementary Fig. 8B (iv)). However, without EVs as dopants, pyrrole alone cannot form polypyrrole through the electrodeposition process. By scanning electron microscopy (SEM), the thickness of the polypyrrole on the surface of the PLA-Au MNs was approximately 6 μm when deposited for 300 s

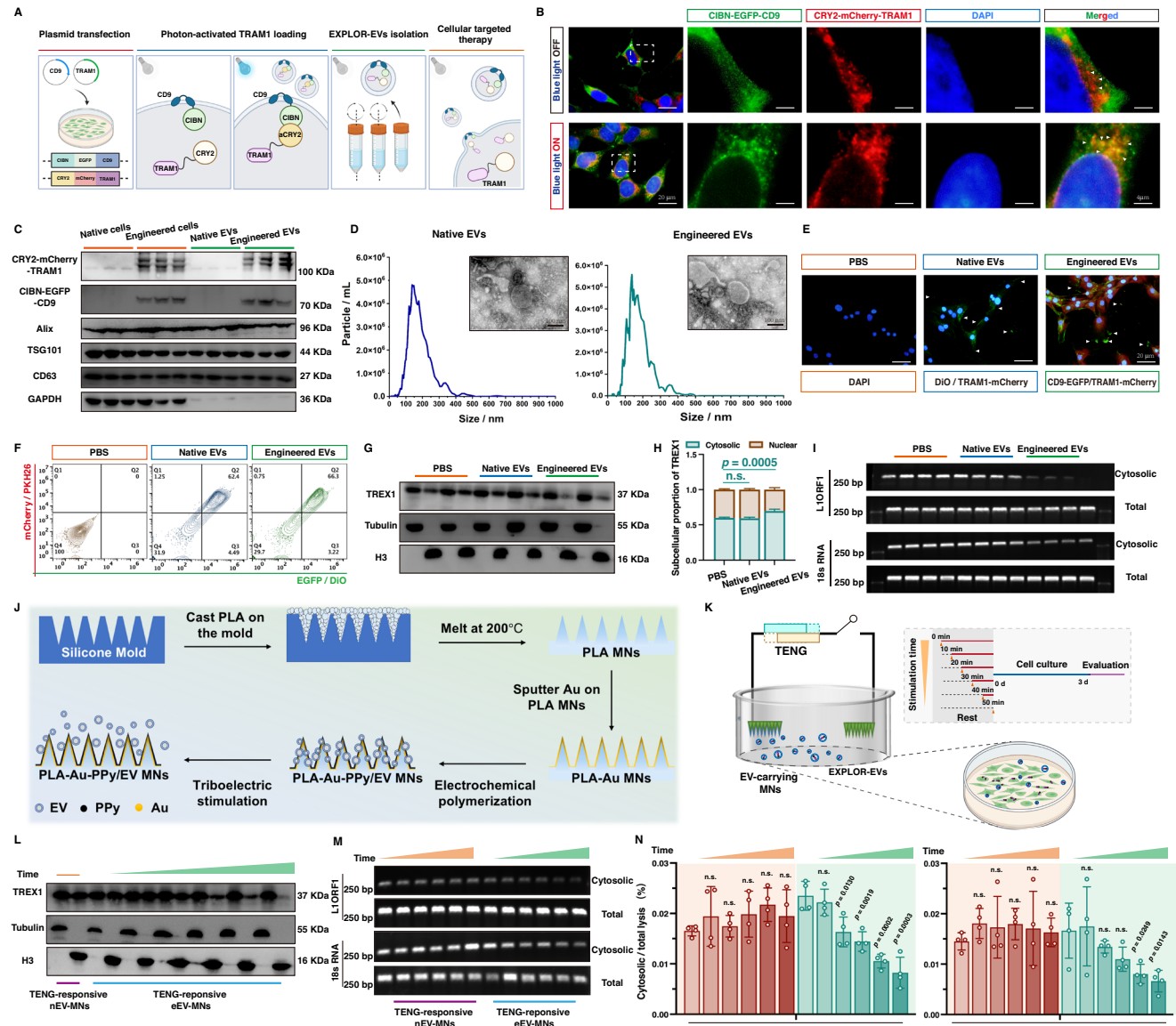

**Fig. 5 | EXPLOR-engineered EVs alleviated the inflammatory senescence phenotype acquisition of NP cells via triboelectric-responsive delivery of TRAM1.**
**A** Schematic workflow for EXPLOR-engineered EVs generation (Created with BioRender.com). **B** Representative IF staining of EGFP and mCherry in HEK-293T cells co-transfected with CIBN-EGFP-CD9 and CRY2-mCherry-TRAM1 plasmids with or without 460-nm laser stimulation (20 μW/cm²), bar: 20 μm (Representative image of three independent technical experiments). **C** Representative western blotting images of CRY2-mCherry-TRAM1, CIBN-EGFP-CD9, Alix, TSG, and CD63 from native or transfected cells and EVs (*n* = 3 biological independent experiments). **D** Nanoparticle tracking analysis (NTA) and representative transition electron microscope (TEM) images of native and engineered EVs, bar: 100 nm (Representative image of three independent technical experiments).
**E** Representative IF staining of EGFP and mCherry in senescent NP cells after treatment with PBS, native or engineered EVs for 72 h (native EVs were labeled by DiO), bar: 20 μm (Representative image of three independent technical experiments). **F** Flow cytometry images to analyze the uptake of EVs in senescent NP cells after treatment with PBS, native or engineered EVs for 72 h (native EVs were labeled

by PKH26 and DiO) (Representative plot of three independent technical experiments). **G, H** Representative western blotting images and quantitative analysis of cytosolic and nuclear TREX1 protein in senescent NP cells with the indicated treatments (Representative blot of four independent technical experiments).
**I** Representative agarose gel electrophoresis images of cytosolic DNA fragments in senescent NP cells with the indicated treatments (*n* = 4 biological independent experiments). **J** Schematic of the preparation process and EV-release of triboelectric-responsive MNs. **K** Schematic workflow for senescent NP cells incubated with triboelectric-responsive MNs (Created with BioRender.com).
**L** Representative western blotting images of cytosolic and nuclear TREX1 protein in senescent NP cells with the indicated treatments (Representative blot of three independent technical experiments). **M, N** Representative agarose gel electrophoresis images (**M**) and quantitative analysis (**N**) of cytosolic DNA fragments in senescent NP cells with the indicated treatments (Representative blot of four independent technical experiments). A significant *p* value was determined by two-tailed unpaired *t* test (**H**) and two-tailed ANOVA (**N**). Mean ± SD are shown for (**N**, **H**). n.s. not significant.

(Supplementary Fig. 8C). The electrical and mechanical properties of the EV-carrying PLA-Au-PPy MNs obtained at different deposition charge densities were also characterized. Cyclic voltammetry is an electrochemical measurement of potentiodynamic potential that can be used to study the redox behavior of EV-carrying PLA-Au-PPy MNs. The electrochemical activity of EV-carrying PLA-Au-PPy MNs at

different deposition charge densities was tested by cyclic voltammetry. The voltage sweep range was set from −1 V to 1 V, and the sweep rate was set to 100 mV/s (Supplementary Fig. 8D). Oxidation and reduction reactions occur on the surface of the membrane during the scanning process, as ions are able to freely enter and exit the EV-carrying PLA-Au-PPy MNs, during which currents are generated on the

surface of the MNs, which are shown on the CV diagram as peak currents for the anode and cathode. With deposition times of 150 s, 200 s, 250 s, and 300 s, EV-carrying PLA-Au-PPy MNs were finally obtained at four deposition current densities: 300 mC/cm², 400 mC/cm², 500 mC/cm², and 600 mC/cm². When the charge density of the electrodeposition process was 300 mC/cm², the maximum reduction current of EV-carrying PLA-Au-PPy MNs was −0.6 V, and an oxidation peak began to appear at 0.12 V. As the charge density of the electrodeposition process increased, the reduction current and oxidation current increased sharply, and the position of the reduction peak and the oxidation peak deviated from 0 V. Compared to those of the PLA-Au MNs, the conductivity of the EV-carrying PLA-Au-PPy MNs decreased. However, with increasing polypyrrole deposition time, the current of the measured EV-carrying PLA-Au-PPy MNs gradually increased, indicating that their conductivity gradually increased. The charge transfer of PLA-Au MNs and EV-carrying PLA-Au-PPy MNs in the frequency range of 0.1 Hz ~ 100 kHz was studied by electrochemical impedance methods. It could be concluded that the impedance of the EV-carrying PLA-Au-PPy MNs gradually decreased as the charge density of the electrodeposition increased (Supplementary Fig. 8E). To test the ability of the MNs to penetrate the skin, we used pig skin to imitate human skin. The mechanical properties of the PLA MNs, PLA-Au MNs, and EV-carrying PLA-Au-PPy MNs were quantitatively tested by using a dynamometer (Supplementary Fig. 8F) (Supplementary Fig. 9). The results showed that with increasing electrodeposition time, the force required by the MNs to pierce the skin gradually increased, and the MNs did not break after insertion into the skin. These results show that the MNs we prepared have excellent hardness and ability to pierce the skin.

### Triboelectric-responsive MNs mitigated inflammatory senescence in NP cells in vitro

To investigate the EXPLOR-engineered EV release kinetics of the triboelectric-responsive MN system, EV-carrying MNs immersed in cell culture medium were connected to the two dissimilar electrode layers by using copper wire (Fig. 5K). TENG output triboelectric stimulation, and the PPy backbone obtained electrostatic electrons, which triggered the EVs to drop and diffuse into the cell culture medium. We collected the medium after TENG triboelectric stimulation over a time course, detected the fluorescence intensity of mCherry-conjugated protein in the collected medium to indirectly calculate the release amount of EVs during different triboelectric stimulations, and unexpectedly found that the accumulated amount of EVs in the collected medium increased with triboelectric stimulation time, characterized by three different stages, namely, a relatively slow release before stimulation for 30 min, a rapid release during 30-90 min, and a relatively slow release after 90 min (Supplementary Fig. 10A). During the process of EV release, the PPy layer under triboelectric stimulation continuously transforms between oxidative and reductive states, and to compensate for the reduced amount of EVs released from the PPy layer and maintain the electrostatic balance, the PPy film may be released with other anions into the medium. Once a major portion of the EVs had been released, the compensated anion was released from the PPy layer into the medium, which slowed the release of EVs after long-term triboelectric stimulation. To some extent, the self-powered triboelectric-responsive EV release mode exhibited sustainable characteristics and improved the utilization of EVs to efficiently alleviate disease processes. In addition, the zeta potentials of EVs exposed to triboelectric stimulation were not significantly different, indicating that triboelectric-responsive release did not affect the stability of EVs (Supplementary Fig. 10B), and NTA analysis revealed the release of TRAM1-engineered EVs at different triboelectric stimulation times (Supplementary Fig. 10C). Senescent NP cells were cultured in media supplemented with native EV-carrying MNs connected to the TENG or with TRAM1-engineered EV-carrying MNs connected to the TENG for

different triboelectric stimulation durations (0, 10, 20, 30, 40 and 50 min) (Fig. 5J). TRAM1-engineered EV-carrying MNs carried EGFP-tagged CD9 and mCherry-tagged TRAM1, and flow cytometry showed that the number of EGFP⁺ mCherry⁺ NP cells increased with increasing stimulation time, which indicates that NP cells increasingly took up EVs in a time-dependent manner (Supplementary Fig. 10D). Notably, western blot analysis revealed that CRY2-conjugated mCherry-tagged TRAM1 accumulated in NP cells in a time-dependent manner, further confirming that the EXPLOR-engineered EVs exhibited an excellent ability to deliver TRAM1 protein into NP cells (Supplementary Fig. 10E). Importantly, after stimulation with the TRAM1-engineered EV-carrying MN system, senescent NP cells exhibited drastically decreased levels of cell cycle arrest-associated markers and reduced secretion of proinflammatory factors, indicating that TRAM1-engineered EVs efficiently rejuvenated senescent NP cells (Supplementary Fig. 10F, G). Senescent NP cells exhibited increased cytosolic TREX1 accompanied by decreased cytosolic damaged DNA, reduced numbers of genomic damage foci and inhibition of the cGAS-STING axis after treatment with TRAM1-engineered EV-loaded MNs for increased durations (Fig. 5L, M) (Supplementary Fig. 10H, I). In addition, we also verified that the lack of illumination stimulation failed to load the TRAM1 into CIBN-CD9-engineered EVs, and the delivery of TRAM1 protein into senescent NP cells was essential for the therapeutic effects of optogenetically engineered EV-carrying MNs for remodeling TREX1 elimination and inhibiting NP cell senescence acquisition, which prevented the effects of cofounding factors of illumination stimulation and photoreactive proteins (Supplementary Fig. 11A–Q).

### An exercise self-powered triboelectric-responsive MN system alleviated the senescence-associated degenerative process of rat coccygeal IVDs

To further evaluate the efficacy of self-powered triboelectric-responsive MNs in the treatment of the IVDD in a rat model, we surgically established needle puncture-induced degeneration of rat coccygeal IVDs. Subsequently, the self-powered TENG was sutured to the lumbar skin of the rat, and lumbar exercise-induced mechanical force was applied vertically to two dissimilar layers of the TENG to drive the layers to periodically contact and separate (Fig. 6A, B) (Supplementary Mov. 1). TRAM1-engineered EV- or native EV-carrying PLA-Au-PPy MNs were assembled with TENGs to fabricate a self-powered triboelectric-responsive MN system by copper wire connecting the MNs to the electrode layers. Triboelectric energy induced by periodical relational contact and separation of the TENG stimulated the MNs and triggered the controllable release of EVs. Pulsed triboelectric stimulation drove the transformation between the oxidative and reductive states of the PPy layer, which ensured the electrical conduction of the PPy layers and promoted the delivery of EVs into the rat coccygeal IVDs. The rats wearing native EV-carrying MNs connected to the TENG (TENG-responsive nEV-MNs group) served as the control group for comparison to those wearing TRAM1-engineered EV-carrying MNs connected to the TENG (TENG-responsive eEV-MNs group) to evaluate the protective role of the TRAM1 protein and correct the exact effects of triboelectric energy or other cargoes in native EVs, and the rats wearing TRAM1-engineered EV-carrying MNs without connection to the TENG (eEV-MNs group) acted as the control group to focus on the triboelectrically controllable release of TRAM1-engineered EVs (Fig. 6B). The EV dynamics detected by in vivo imaging of the EVs revealed that the fluorescence signals of PKH26-labeled EVs and TRAM1-engineered EVs obviously accumulated and increased with increasing stimulation time in the coccygeal intervertebral space of rats in the TENG-responsive nEV-MNs group and TENG-responsive eEV-MNs group, and little fluorescence signal appeared in the coccygeal intervertebral space of the eEVs-MNs group, suggesting that native or TRAM1-engineered EVs could be controllably delivered, driven by lumbar exercise, to allow the TENGs to transform mechanical energy

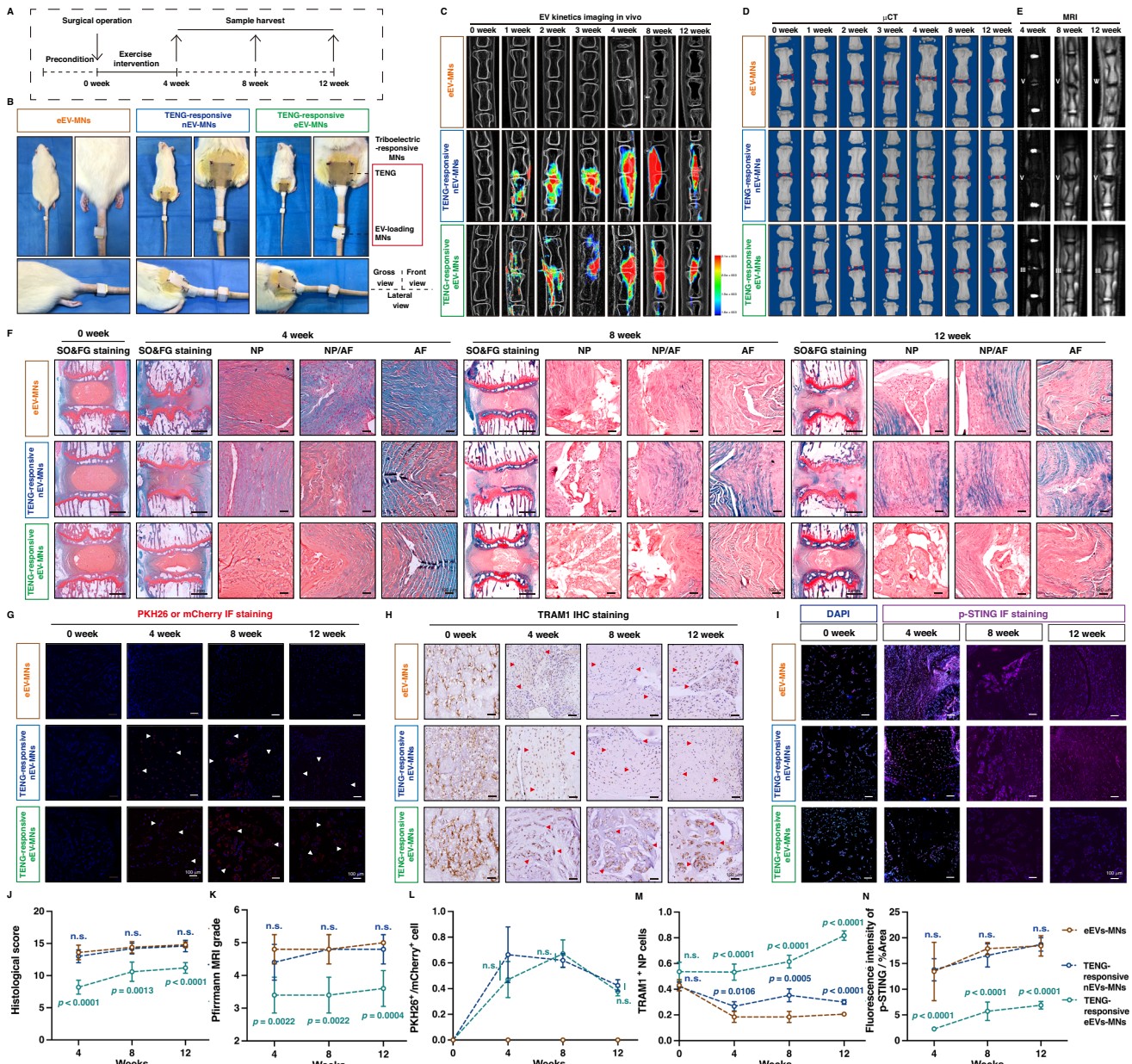

**Fig. 6 | Self-powered triboelectric-responsive MN system alleviated the senescence-associated degenerative process of IVDs. A** Schematic workflow for the needle puncture-induced degeneration of rat coccygeal IVDs treated with triboelectric-responsive MNs ($n = 5$ rats per group). **B** Gross pictures of coccygeal IVD model wearing with different equipments. **C** Representative X-ray images and EVs in vivo imaging showing the EV release kinetics in coccygeal IVDs at different time points. **D** Representative μCT images of coccygeal IVDs at different time points. **E** MR T2-weighted images and Pfirrmann MRI grades of rat IVDs with indicated treatments. **F** SO&FG staining of rat IVDs with the indicated treatments. **G** IF staining of PKH26-labeled EVs or mCherry in rat IVDs with the indicated treatments. The arrows indicated the PKH26+ or mCherry+ NP cells. **H** IHC staining

of TRAM1 in rat IVDs with the indicated treatments. The arrows indicated TRAM1+ NP cells. **I** IF staining of p-STING in rat IVDs with the indicated treatments. **J** Pfirrmann MRI grades of rat IVDs with indicated treatments. **K** Histological score of rat IVDs with the indicated treatments. **L** Quantitative analysis of PKH26+ or mCherry+ NP cell proportions in rat IVDs with the indicated treatments. **M** Quantitative analysis of the TRAM1+ NP cell proportion in rat IVDs with the indicated treatments. **N** Quantitative analysis of the fluorescence intensity of p-STING in rat IVDs with the indicated treatments. Representative images of five independent biological replicates, and quantification of five independent biological experiments (**J**–**N**). A significant $p$ value was determined by two-tailed ANOVA (**J**–**N**). Mean ± SD are shown for (**J**–**N**). n.s. not significant.

to triboelectricity (Fig. 6C). In addition, we verified that the EV-carrying MN systems with or without triboelectric stimulation did not contribute to unexpected inflammatory responses in the inserted skin or paravertebral tissues, including paravertebral muscles and paravertebral ligaments (Supplementary Fig. 12A, B). Furthermore, radiological results from X-ray and μCT analysis showed that the delivery of TRAM1-engineered EVs into surgically damaged IVDs improved the height of coccygeal intervertebral discs in the TENG-responsive eEV-MNs group, (Fig. 6C, D), and MRI results revealed that treatment with

TRAM1-engineered EVs could increase the water content and reduce fibrosis in coccygeal IVDs in the TENG-responsive eEV-MNs group (Fig. 6E, J). Histologically, as revealed by H&E staining and SO&GF staining, the coccygeal IVD tissues of rats in the TENG-responsive eEV-MNs group exhibited a dramatically reduced degenerative phenotype characterized by a vacuolated and stellar morphology of NP cells with a peripheral proteoglycan matrix, a partially clear boundary between the AF and NP tissue, fibroblast-like AF cells and well-organized AF collagen lamellae (Fig. 6F, K, Supplementary Fig. 13A). Coccygeal IVD

tissues from the eEV-MNs and TENG-responsive nEV-MNs groups showed a loss of well-organized structure and severe interruption of the boundary between the NP and AF regions, but TENG-responsive eEV-MN treatment prevented the collapse of the IVD space and partially preserved the concentrical arrangement of the AF lamellae (Fig. 6F, K, Supplementary Fig. 13A). More collagen was deposited in the extracellular matrix and arranged radically in the well-structured AF tissue of rats in the TENG-responsive eEV-MNs group (Supplementary Fig. 13B). Senescence-associated marker immunohistochemical (IHC) staining showed that the number of p-p53+ NP cells significantly decreased in the TENG-responsive eEV-MNs group, which suggested that the delivery of the TRAM1 protein efficiently reduced the number of senescent cells (Supplementary Fig. 13D, F). As expected, PKH26-labeled native EVs and mCherry-conjugated TRAM1-carrying EVs were successfully delivered into IVD tissues and accumulated in IVD tissues after 12 weeks, as shown by IF staining (Fig. 6G, L). Interestingly, the expression of TRAM1 protein obviously decreased in degenerated coccygeal IVDs, and TRAM1-engineered EVs increased the number of TRAM1+ cells and exhibited excellent delivery efficiency of TRAM1 into NP cells to maintain the protective effects of TRAM1 against cellular senescence (Fig. 6H, M). The levels of effector of cGAS-STING axis activation (phosphorylated-STING, p-STING) and inflammatory mediator (IL-1β) were obviously decreased in rat IVD tissues via the triboelectric-responsive release of TRAM1-engineered EVs (Fig. 6I, N, Supplementary Fig. 13E, G). In addition, we verified that long-term treatment with mCherry-conjugated TRAM1-carrying EVs had anti-degenerative effects, including improving the historical structure and reducing the senescent phenotype of coccygeal IVDD (for at least 12 weeks), which indicated that the anti-degenerative effects of these EVs were sustainable upon the cessation of EVs release (Fig. 6C–N). In addition, we also verified that the CIBN-CD9-engineered EV-carrying MN system without illumination-induced TRAM1 loading failed to alleviate the degenerative progression of coccygeal IVDs, and the delivery of TRAM1 protein into senescent NP cells was essential for the therapeutic effects of optogenetically engineered EVs for inhibiting the NP cell inflammatory response, which avoided the effects of cofounding factors of illumination stimulation and photoreactive proteins (Supplementary Fig. 14A–N).

To evaluate the therapeutic effects of different delivery strategies, including TRAM1 protein injection, lentivirus (LV) transfection, and EV-based delivery, we directly injected exogenous GST-tagged TRAM1 protein into rat coccygeal NP tissues or used TRAM1-overexpressing plasmid-carrying LVs to transfect coccygeal NP cells (Supplementary Fig. 15A–E). It was difficult for exogenously injected TRAM1 protein to be stored in NP tissues, and exogenous protein administration failed to deliver TRAM1 into NP cells and alleviate needle picture-induced degeneration of coccygeal IVDs (Supplementary Fig. 15A–E). LV transfection is an efficient delivery method for improving the number of TRAM1+ NP cells and migrating coccygeal IVDD (Supplementary Fig. 15A–E). Interestingly, via IHC staining, we found that the number of TRAM1+ NP cells increased until 8 weeks and subsequently decreased at 12 weeks, and we inferred that the increase in the number of TRAM1 gene copies limited the ability to upregulate the expression of TRAM1 and remodel the function of TRAM1 because the endogenous etiological factors driving the progression of degeneration were not eliminated and the endogenous molecular mechanism involved in TRAM1 deficiency or dysfunction was still unknown (Supplementary Fig. 15A–E). Similarly, the EV-based strategy exhibited the ability to deliver functional proteins to NP cells to overcome the challenges associated with protein delivery and the limited efficiency of LVs in regulating endogenous protein expression (Supplementary Fig. 15D, E). In summary, mechanical energy driven by lumbar exercise could efficiently control the release of TRAM1 into IVD tissues via the triboelectric-responsive MN system, which alleviated senescence-associated coccygeal IVD degeneration.

## Discussion

IVDD afflicts a large number of adults worldwide due to the increase in the aging population and a modern sedentary lifestyle, which serves as an important causative factor of LBP[1,15]. Gelatinous NP tissue acts not only as a conduit with excellent biomechanical properties to distribute compressive and tensile strength but also as a signaling hub for maintaining IVD integrity[53]. NP cell senescence accompanied by disrupted biomechanical homeostasis and inflammatory phenotype acquisition attributed to etiological factors, including aging, altered mechanical stress, and poor posture, is regarded as a prominent characteristic of the initiation and progression of IVDD[14,53]. Although tissue engineering strategies have shown promising and innovative biomaterials with excellent biomechanical properties for the treatment of early-stage IVDD in preclinical studies, none of those approaches have been approved for clinical application due to the lack of integration of etiology-driven interventions and treatments intrinsically targeting degenerative molecules[16,21,70].

Herein, we innovatively designed a self-powered triboelectric-responsive MN array integrated with the exercise-controlled release of EXPLOR-engineered EVs for IVDD repair. Resembling the pathological progression of IVDD, physical exercise triggered the activation of cytosolic DNA sensing-mediated inflammatory response in NP cells and promoted IVD degeneration in surgically induced lumbar instability. Mechanistically, an interactomic array combined with loss- and gain-of-function analysis demonstrated that the disassembly of TRAM1-TREX1 disrupted the localization of TREX1 in the ER and converted its functional activity from immune clearance to immune damage, which ultimately contributed to the activation of cytosolic DNA sensing-mediated inflammatory senescence in senescent NP cells. EXPLOR-engineered EVs integrated reversible optical-responsive CRY2-CIBN interactions with TRAM1 protein loading, blocked TREX1 nuclear translocation, promoted TREX1 anchoring to the ER and eliminated cytosolic damaged DNA. Self-powered triboelectric-responsive MNs efficiently control the release of TRAM1-engineered EVs via electrostatic attraction and show promise for alleviating the degenerative process of IVDs, providing an innovative repair approach for exercise-related disorders (Fig. 1D).

### Exercise self-powered MN-based band aids for minimally invasive IVDD treatment

Current pharmacological therapies focus on anti-inflammatory or pain-relieving drugs such as nonsteroidal anti-inflammatory drugs (NSAIDs) to alleviate pain-associated symptoms, resulting in substantial drug consumption and unavoidable side effects[15,71]. Secreted inflammatory factors, including cytokines and chemokines, exhibit damaging effects on all aspects of the degenerative process, and anti-cytokine agents and inflammatory signaling pathway inhibitors are used to alleviate structural damage and mitigate radicular pain[14]. Despite their satisfactory efficacy in cell or organ culture models or animal models, their long-term efficacy and safety are unable to meet patient demands in human clinical trials[14]. Surgical discectomy efficiently improves LBP-related outcomes and discogenic disability in patients who are nonresponsive to conservative intervention or suffer from progressive neurological deficits, and minimally invasive lumbar discectomy is currently popular and promising because it is superior to conventional open surgery in reducing tissue damage, shortening hospital stays and improving patient satisfaction[70,72]. Minimally invasive technology avoids destruction of the lamina, paraspinal muscles and soft tissues, which maintains the mechanical integrity of the spine and consequently reduces the risk of postoperative segmental instability[72–74]. However, minimally invasive discectomy does not cure or reverse the degenerative process but only relieves neurological compression, which leaves a large IVD defect with high reherniation rates and accelerates the degenerative progress of the remaining discectomized IVDs[70]. In this study, the self-powered triboelectric-responsive MN array takes advantage of the minimal

invasiveness of MNs to deliver EXPLOR-engineered EVs across the epidermis and into the superficial dermis with reduced damage to capillaries and reduced activation of nociceptors[27,75]. Notably, the band aid-like MN array was designed to target the degeneration-associated molecules and to block the aberrant activation of the cGAS-STING axis-mediated inflammatory response without destroying the mechanical integrity of IVDs.

## Triboelectric-responsive EXPLOR-engineered EV release system for biologically targeted IVDD treatment

Excellent strategies to deliver therapeutic proteins to NP tissue are essential for molecule-targeted repair in IVDD, which blocks the initial cue of the degenerative process. Synthetic nanoparticles, including liposomes and polymeric nanoparticles, are regarded as preferred protein delivery systems; however, liposomes have the capacity to fuse or aggregate, which affects the mature release of cargo, and polymeric nanoparticles lack suitable biocompatibility and long-term safety[21,52]. In addition, the manufacturing of protein-encapsulated liposomes or nanoparticles is limited due to the complicated processes of generating purified proteins and ex vivo encapsulation[21,52]. Cell-derived EVs with weak immunogenicity exhibit a natural capacity for surface-membrane trafficking and the delivering of signaling factors to mediate cell-cell communication and modulate the functional status of recipient cells[22,25]. To overcome the challenges of effective cargo loading and EV engineering for biological targeted therapeutics, we applied EXPLOR technology to integrate optically reversible CYR2-CIBN interactions with conjugated TRAM1 protein loading in EVs, which improved TRAM1 protein loading efficiency, drastically expanded the molecule- or pathway-targeting capacity of EVs and enhanced the therapeutic effects of EVs in inhibiting the inflammatory response during IVDD. In addition, stimulation-responsive EVs-carrying system induced by exogenous physical signals provide the controllable release of EVs for highly efficient and sustainable applications[22,25]. Cell-derived EVs are secreted membrane-delimited particles constructed of an anionic phospholipid bilayer, which indicates that EVs have potential electrical properties[68,69]. Given that PPy has an excellent capacity to load and release drugs by switching between the oxidative and reductive states, we utilized a TENG to convert biomechanical energy from irregular motion to electrical energy, which drove the loss of electrostatic attraction between anionic EVs and the PPy layer in the reductive state, and thereby fabricated a smart self-powered electroresponsive system to release therapeutic EVs for IVDD.

In summary, a smart self-powered electroresponsive system for IVDD repair exhibited excellent performance by integrating MN-based minimally invasive therapeutics with the triboelectric-controlled release of EXPLOR-engineered EVs for disease-modifying intervention. This system inhibited the activation of cytosolic DNA-mediated inflammatory response involved in NP cell senescence during the degenerative process of IVDD, which provides an innovative strategy for the precise regulation of the inflammatory response and shows promising clinical potential in the treatment of degeneration-associated disorders.

# Methods

## Ethics approval

Ethics involved in experimental arrangement and implementation in this study, including patient medical data consultation, NP tissue sample collection and application, animal obtainment, animal surgical operation, and animal sample collection, were approved by Ethics Committee of Tongji Medical College, Huazhong University of Science and Technology (No. S341) (No. S2394).

## Human NP tissue sample collection

Healthy human NP tissue samples were collected from patients who had idiopathic scoliosis and underwent orthopedic surgeries involving in lumbar segments, and the MR-T2 weighed image-based Pfirrmann grading system identified the no-degenerated IVD tissue as Grade I. Degenerated NP tissue samples were collected from patients with lumbar intervertebral disc herniation who underwent discectomy, and Pfirrmann grading system evaluated the degenerated IVD tissue as Grade III to Grade IV. Furthermore, these patients without cardio-cerebral-vascular disease, cancers, infection, immune and endocrine diseases or organ dysfunction approved the application of NP tissue samples in scientific studies. Informed consent for the use of human NP tissues for medical research was obtained from all volunteers. Three pairs, including three healthy and three degenerated samples, were used for transcriptome RNA sequencing analysis, and NP cells were isolated from other healthy samples to perform in vitro experiments.

## RNA sequencing and GSEA (Gene set enrichment analysis)

After collection, fresh NP tissues were washed with PBS to remove surface residues, and placed into liquid nitrogen for freezing. Frozen NP tissues were ground under liquid nitrogen. Suspensions were filtered, and Trizol was added to isolate RNA. After qualification, mRNA of isolated NP cell was enriched with magnetic beads with Oligo (dT), broken into fragments, and mixed with primers to synthesize cDNA. cDNA was eluted, purified, and terminally modified, and then sequencing connector treatment was carried out. Targeting fragments were recovered by gel electrophoresis, amplified by PCR, and used to prepare a whole cDNA library. After qualification, libraries were sequenced.

After the above sequencing was completed, raw data were filtered to obtain high-quality comprehensive transcript information of NP cells on the human reference genome, and gene quantification was carried out. GSEA 4.2.3 software (Massachusetts Institute of Technology, USA) with the set parameters of NES (Normalized enrich score) < −1 or >1, unbiased $p$ value < 0.05, FDR $q$-value < 0.25 was used to perform enrichment analysis on sequencing results.

## Cell isolation and culture

Fresh healthy NP tissue samples obtained by surgical operation were placed into PBS (Phosphate buffered saline, Biosharp, China) and cut into 1–3 mm³ pieces. After washing small pieces three times with PBS to remove superficial residues, 0.4% type II collagenase (Invitrogen, USA) was used to digest the samples at 37 °C for 4–6 h. The digested cell suspension was centrifuged at 800 rpm for 5 min, and washed three times with PBS, and the cell precipitate was resuspended in cell culture medium (1:1 DMEM: F12, 1:1 Dulbecco's modified Eagle medium: F12, Gibco, USA) containing 10% FBS (Fetal bovine serum, Gibco, USA) and 1% penicillin-streptomycin (Gibco, USA). NP cells were cultured at 37 °C and 5% $CO_2$, and after one week, the culture medium was first replaced. Then, the culture medium was replaced every three days until NP cells covered the bottom of the culture flask and reached over 95% confluence. Cells were passaged at a ratio of 1:3 or 1:4. NP cells in the second passage were regarded as normal state and used to perform experiments. To induce cellular senescence, NP cells were subjected to serial passage and continuous culture, and those in the eighth passage were identified as having a senescence-like phenotype and used to perform experiments.

The HEK-293T cell line was obtained from ATTC (American Type Tissue Culture Collection, USA). We used DMEM with 10% FBS (Gibco, USA) to culture HEK-293T cells at 37 °C and 5% $CO_2$ and replaced the cell culture medium every two days. Until HEK-293T cells covered the bottom of the culture dish, the cells were passaged at a ratio of 1:10.

## Small interfering RNA (siRNA) technology and plasmid transfection

To knock down the expression of TRAM1, two specific siRNAs (100 nM) targeting TRAM1 were used to transfect NP cells using Lipofectamine

2000 (Thermo Fisher, USA). When the cells reached over 70% confluence, the transfection mixture with two specific siRNAs was added to replace the culture medium, and the cells were cultured at 37 °C with 5% $CO_2$. After 72 h of culture, transfected NP cells were collected for further experiments.

Gene fragments encoding wild-type, mutant (18th amino acid residue replaced aspartate with asparagine) and truncated TREX1 (the anime-terminal from 1st to 235th amino acid) were amplified from human cDNA and inserted into the pcDNA5-Flag vector for cloning into TREX1-WT, TREX1-MUT, and TREX1 amine-terminal plasmids.

NP cells were transfected with plasmids using Lipofectamine 2000. After the cells reached over 70% confluence, the transfection mixture was added to replace the culture medium, and the cells were cultured at 37 °C with 5% $CO_2$. After 72 h of culture, transfected NP cells were collected for further experiments.

### Lentiviral transduction
Gene fragment encoding rat TRAM1 protein was amplified and inserted into the Ubi-MCS-3FLAG-SV40-EGFP-IRES-Puromycin vector for cloning into TRAM1-overpressing plasmid. And this TRAM1-overpressing plasmid or vector plasmid were loading into LVs for plasmid transduction in vivo.

### EXPLOR technology to engineer EVs for TRAM1 loading
**Optical-responsive plasmid generation.** Full-length TRAM1 was amplified from human cDNA and cloned into the pLV3-6×His vector. A gene fragment encoding full-length CIBN-EGFP was amplified from *CIBN-EGFP* constructs, and the gene fragment encoding CD9 was amplified from human cDNA. The two gene fragments were ligated into the GV219 (CMV-MCS-SV40-Neomycin) vector to generate the CIBN-conjugated EGFP-tagged CD9 plasmid. To generate the CRY2-conjugated mCherry-tagged TRAM1 plasmid, a gene fragment encoding full-length CRY2-mCherry was amplified from *CRY2-mCherry* constructs, and the gene fragment encoding TRAM1 was amplified from human cDNA. The two gene fragments were ligated into the GV219 (CMV-MCS-SV40-Neomycin) vector.

### HEK-293T cell transfection and EXPLOR technology engineered EV generation.
HEK-293T cells were cultured in DMEM containing 10% FBS and transfected with CIBN-conjugated EGFP-tagged CD9 and CRY2-conjugated mCherry-tagged TRAM1 plasmids by using Lipofectamine 2000 (Thermo Fisher, USA). After transfection for 12 h, the cell culture medium was replaced with 10% EV-depleted FBS (System Biosciences, USA). Then, the cells were exposed to continuous 460 nm blue light illumination with an intensity of 20 μW/cm² at 37 °C with 5% $CO_2$. After 48 h of light illumination, the cell culture supernatant was collected, and the differential centrifugation method was used to isolate EVs according to the experimental protocol published in the Minimal information for studies of extracellular vesicles 2018 (MISEV 2018) guidelines. In brief, the culture supernatant was centrifuged at 500 × $g$ for 10 min, at 2000 × $g$ at 4 °C for 40 min, and at 10,000 × $g$ at 4 °C for 60 min to remove cells or debris and finally at 20,000 × $g$ at 4 °C for 60 min to isolate EVs, then washed twice with PBS. The suspended EVs were then filtered through a syringe filter (0.22 μm, Sartorius, Germany). Native EVs were isolated from the differential centrifugation of the cell culture supernatant of HEK-293T cells without plasmid transfection and light illumination.

### EV identification.
Western blot analysis was used to identify the protein cargoes in EVs. After isolation by using the differential centrifugation method, the EV precipitate was lysed in RIPA buffer (Boster, China) containing 1% PMSF (Boster, China) and EV-loaded proteins were extracted for SDS-PAGE electrophoresis. Transmission electron microscopy (TEM) was used to show the morphology of native and engineered EVs. 20 μl of EV solution was loaded on a formvar/carbon-coated grid for 5 min. After drying, the grid was negatively stained with 3% phosphotungstic acid for 1 min, dried overnight, and observed by TEM (H-7000FA, Hitachi LTD, Japan). Nanoparticle tracking analysis (NTA) was used to detect the concentration and size of EVs. The EV solution was appropriately diluted with PBS within the recommended concentration range and analyzed by a NANOSIGHT NS300 system (Malvern, UK). Zeta potential analysis (Malvern Zetasizer Nano ZS90, UK) was used to detect the stability of EVs.

**EV labeling.** Isolated EVs were incubated with 5 μM PKH26 (Sigma-Aldrich, USA) at 4 °C for 20 min and washed with PBS twice to remove nonconjugated PKH26.

**EV treatment and uptake analysis.** After treatment according to the experiment arrangement, NP cells were washed twice with PBS, and incubated with labeled EVs (1 × 10⁶ particle/ml) or engineered EVs (1 × 10⁶ particle/ml) for 72 h at 37 °C and 5% $CO_2$. After incubation with labeled or engineered EVs for 72 h, flow cytometer (BD Biosciences, USA) was used to detect the fluorescence intensity.

### Flow cytometer
After the indicated treatment, NP cells were washed twice with PBS and digested into a cell suspension by using trypsin. Then, FACSCalibur flow cytometer (BD Biosciences, USA) was used to detect the fluorescent intensity of EGFP (Excitation wavelength: 488 nm; Emission wavelength: 507 nm), mCherry (Excitation wavelength: 580 nm; Emission wavelength: 610 nm), PKH26 (Excitation wavelength: 551 nm; Emission wavelength: 567 nm) or DiO (Excitation wavelength: 484 nm; Emission wavelength: 501 nm) in NP cells.

### Western blotting
After treatment according to the experiment arrangement, NP cells were washed twice with PBS, incubated with RIPA buffer containing 1% protease inhibitor PMSF, lysed on ice for 15 min, ultrasonically crushed for 30 s, and centrifuged at 13,000 × $g$ for 15 min at 4 °C. The lysed supernatant was aspirated into precooled EP tubes, and the concentration of cellular lysed protein was detected by a BCA protein assay kit. 40 μg of protein was separated by 8–12% SDS-PAGE and transferred to 0.22 nm or 0.45 nm PVDF membranes (Sigma-Aldrich, USA) according to the protein molecular weight. PVDF membranes were incubated in 5% BSA (Bovine serum albumin) to block the nonspecific sites for 1 h, washed four times with 0.1% TBST, and incubated with primary antibodies at 4 °C overnight and secondary antibodies at room temperature for 1 h. After washed with 0.1% TBST four times to remove nonspecific antibodies, the membranes were exposed by chemiluminescence (Affinity, USA), and images were captured by a ChemiDoc Imaging System (Bio-Rad, USA).

### Co-immunoprecipitation (Co-IP)
After treatment according to the experiment arrangement, NP cells were washed twice with PBS, incubated with NP-40 containing 1.5% PMSF and 1.5% protease inhibitor cocktail (Boster, China), lysed on ice for 25 min, and centrifuged at 13,000 × $g$ for 15 min at 4 °C. The lysed supernatant was aspirated into precooled EP tubes, and the concentration of cellular lysed protein was detected by a BCA protein assay kit. A total of 1000–2000 μg of protein was used for Co-IP analysis, and 3–5 μg of primary antibody or control antibody was added to the extracted protein solution. The protein-antibody mixture was shaken moderately at 4 °C overnight. After washing with NP-40, protein A/G magnetic beads (MedChemExpress, USA) were added to the protein-antibody mixture, and the protein-antibody-magnetic bead mixture was shaken moderately at 4 °C for 6 h. After washing with NP-40, the bound proteins were eluted with 1 × protein loading buffer (Boster, China) at 95 °C for 10 min and analyzed by western blotting.

## TREX1 interactome analysis

After treatment according to the experiment design, normal NP cells (NP cells in the second passage) and senescent NP cells (NP cells in the eighth passage) were used to extract TREX1-binding proteins by incubation with anti-TREX1 antibody and control antibody according to Co-IP analysis. The eluted proteins were separated by SDS-PAGE. The protein gel was washed, decolorized, reductively alkylated and digested with trypsin to obtain peptides. The peptides were extracted and dried under vacuum. Then, 0.1% formic acid was added to redissolve the samples, and l-2 μg sample was removed for separation by using EASY-nLC1200 (Thermo Scientific, USA) and self-filled Trap column (Cl8, 5 μm) and analytical column (Cl8, 1.9 μm) with a flow rate of 200 nl/min. Using DDA (Data-dependent acquisition) mode, Orbitrap Fusion Lumos (Thermo Scientific, USA) was applied for mass spectrometric detection. Proteome Discoverer (Version 1.4) was used to analyze the raw data.

## RNA extraction and reverse transcription

After treatment according to the experiment design, total RNA was extracted from NP cells according to the manufacturer's protocol. In brief, NP cells were washed twice with PBS, incubated with TRIzol (Invitrogen, USA), lysed on ice for 10 min, added to chloroform and shocked in a vortex mixer for 15 s to extract RNA. The cell lysate was incubated on ice for 15 min, and was centrifuged at $13,000 \times g$ for 15 min at 4 °C. The aqueous phase was precipitated by isopropanol and centrifuged at $13,000 \times g$ for 30 min at 4 °C. After washed with 75% ethanol twice, RNA precipitate was dissolved in DEPC water. HiScript III RT SuperMix (Vazyme, China) was used to synthesize cDNA via the reverse transcription of RNA.

## Cytosolic and total DNA extraction

After treatment according to the experiment design, cytosolic and total genomic DNA of NP cells were extracted according to the manufacturer's protocol[76]. In brief, adherent NP cells were washed twice with PBS, digested into a cell suspension via trypsin and separated into two parts. One-half of the cell suspension was lysed by precooled 0.1% NP-40 solution on ice for 30 min and centrifuged at $14,000 \times g$ for 25 min at 4 °C to collect the supernatant as the cytosolic component. The other half of the cell suspension was lysed by strong digestion buffer as the total cellular component. DNA from cytosolic or total components was extracted by PureLink Genomic DNA kits (Invitrogen, USA).

## Polymerase chain reaction (PCR) and real-time quantitative PCR (RT-qPCR)

Cytosolic DNA and total genomic DNA levels were detected by PCR technology via 2 × Taq Master Mix (Vazyme, China). The relative transcription levels of mRNA were detected by RT-qPCR of cDNA via Taq Pro Universal SYBR qPCR Master Mix (Vazyme, China), performed on a Real-Time PCR System (Bio-Rad, USA). The transcription level of GAPDH was used as a control to normalize the transcription levels of targeted molecules, and $2^{-\Delta\Delta Ct}$ was calculated as the quantitative index.

## DNA agarose gel electrophoresis

DNA amplification products were visualized by DNA agarose gel electrophoresis. After PCR amplification, the products were mixed with 6 × DNA loading buffer (Vazyme, China), separated by 1.5% agarose gel electrophoresis at 120 V for 45 min, developed via 0.01% GelRed dye (Invitrogen, USA) and exposed by a ChemiDoc Imaging System (Bio-Rad, USA).

## Immunofluorescence (IF) staining

After treatment according to the experiment design, NP cells were washed twice with PBS, fixed with 4% paraformaldehyde fixative for 15 min, permeabilized with 0.5% Triton for 20 min, blocked with 3% BSA for 30 min, incubated with primary antibodies at 4 °C for 8 h, washed with 0.1% PBST (PBS-Tween) four times and incubated with fluorescein-conjugated secondary antibodies at room temperature for 8 h. After washing with 0.1% PBST and staining with DAPI (Invitrogen, USA), NP cells were observed and captured by fluorescence microscopy (Olympus, Japan).

## Senescence-associated β-galactosidase staining (SA-β-Gal)

After treatment according to the experiment design, NP cells were washed by PBS twice, fixed with SA-β-Gal staining fixative at room temperature for 15 min, washed with PBS three times for 3 min, added with staining solution at 4 °C for 8 h. After washed with PBS, NP cells were observed and captured in the microscopy (Olympus, Japan).

## Fabrication of the TENG

The device was a triboelectric nanogenerator in contact-separation mode, including a PTFE membrane and ITO/PET conductive films. PTFE is a commercial single-sided adhesive film, and ITO/PET is a flexible conductive film for commercial use. The spacer between the PTFE film and ITO/PET film was prepared with 3 M tape.

## Self-powered triboelectric-responsive MNs in vitro

EV-carrying MNs immersed in 1:1 DMEM:F12 cell culture medium were connected to the two dissimilar electrode layers of the TENG by using copper wire. The TENG was driven by a motor at a frequency of 2 Hz to push two dissimilar electrode layers into relative contact and separate to generate triboelectric energy. After triboelectric stimulation for different times (0, 10, 20, 30, 40, 50 min), NP cells were cultured with the treated culture medium for 72 h to evaluate the effects of EVs in vitro experiments. To evaluate the release dynamics of EVs, the cell supernatant was collected after different triboelectric stimulation duration (range from 0 to 110 min, interval of 10 min), and a microplate reader (BioTek, USA) was used to detect the fluorescence intensity of the supernatant.

## Animal operation

Three-month-old, 200 g ± 20 g male Sprague Dawley rats (SD rats) were obtained from Laboratory Animal Center, Huazhong University of Science and Technology. SD rats were raised under a standard and specific pathogen-free environment with a constant temperature of 21–24 °C and a 1:1 dark:light cycle. Animals were randomly and blindly assigned to groups for in vivo experiments.

**Lumbar spine instability (LSI) animal model.** After anesthetization with 3% pentobarbital sodium, randomly selected SD rats were surgically resected from the lumbar 4th–lumbar 5th (L4-L5) spinous processes of the rat spine along with the supraspinous and interspinous ligaments to induce lumbar spine mechanical instability (LSI), and the posterior paravertebral muscles and other soft tissues from the L4-L5 vertebrae were detached in rats from the sham operation group (Sham).

**SD rat exercise.** A transparent enclosed cage was installed to cover the treadmill. One week before the surgical operation, SD rats were placed on the treadmill every day to precondition them to the running exercise equipment. SD rats underwent lumbar spinous process resection and recovery for 2 weeks after surgery. Over the first week of exercise, rats ran at a rate of 0.5 km/h and slowly reached a speed of 1 km/h. During the second week, the running exercise was kept at a speed of 1 km/min for 30 min every day. After 4 weeks of exercise intervention, randomly selected rats were euthanized and the degeneration levels of lumbar IVDs were evaluated by histological staining. After 8 and 12 weeks, rats were sacrificed, and radiological analysis and histological staining were used to assess the degeneration of lumbar IVDs.

**Needle puncture-induced coccygeal IVD degeneration model.**
After anesthetization with 3% pentobarbital sodium, randomly selected SD rats were fixed on the operating table. We marked the position of the coccygeal 7th–coccygeal 8th (C7-C8) IVDs and then vertically inserted a 29-G needle for 3 mm to induce coccygeal IVD degeneration parallel to the CEP. The needle was rotated 360° and kept in the coccygeal IVDs for 30 s.

**Exercise self-powered triboelectric-responsive MNs for degenerated IVD repair.** A needle puncture-induced coccygeal IVD degeneration model was used to evaluate the therapeutic effects of exercise self-powered triboelectric-responsive MNs. The TENG was sutured to the lumbar skin of the rat, and lumbar exercise-induced mechanical force was vertically applied to two dissimilar layers of the TENG to drive the layers to periodically relationally contact and separate. EV-carrying MNs were pierced into the C7-C8 skin of rats and assembled with or without TENGs by copper wire connecting the MNs to the electrode layers to evaluate the specific protective role of triboelectrically controllable release of TRAM1-engineered EVs. Rats wearing different equipment underwent exercise interventions according to the abovementioned protocol, namely, running on the treadmill at a speed of 1 km/h for 30 min every day for 4 weeks. Every week, rats were randomly selected to detect EVs dynamics and radiologically evaluate degenerative levels. A new EVs-carry MNs were replaced to ensure the biological activity of TRAM1 protein packaged in engineered EVs for 7 days. At 4, 8, and 12 weeks after surgical operation and exercise intervention, histological staining was used to assess the degenerative characteristics of coccygeal IVDs.

**Intradiscal injection of TRAM1 protein.** A needle puncture-induced coccygeal IVD degeneration model was used to evaluate the therapeutic effects of TRAM1 protein. After the initial coccygeal IVDs needle puncture, rats were randomly and double-blindly assigned to 2 treatment groups. After anesthesia with 3% pentobarbital, a total of 2 μl solution containing GST-tagged TRAM1 protein (100 ng/μl) or solvent was slowly injected into the center of NP tissues of Co7-8 coccygeal IVDs every week for four weeks, and after 4, 8, and 12 weeks, the rats were sacrificed for radiographic and histological analysis.

**Intradiscal injection of LVs.** A needle puncture-induced coccygeal IVD degeneration model was used to evaluate the therapeutic effects of TRAM1-overexpressing plasmids-carrying LVs. After the initial coccygeal IVDs needle puncture, rats were randomly and double-blindly assigned to 2 treatment groups. After anesthesia with 3% pentobarbital, a total of 2 μl solution containing TRAM1-overexpression plasmids-carrying LVs or vector-carrying LVs ($10^6$ plaque-forming unit) was slowly injected into the center of NP tissues of Co7-8 coccygeal IVDs every week for 4 weeks, and after 4, 8, and 12 weeks, the rats were sacrificed for radiographic and histological analysis.

**RNA sequencing and GSEA of rat lumbar NP tissues**
According to the experiment design, SD rats were sacrificed, and fresh NP tissues were separated under microscopy and placed into liquid nitrogen for freezing. The cDNA library was prepared according to the abovementioned methods. Raw data were filtered to obtain high-quality comprehensive transcript information of NP cells on the rat reference genome, and gene quantification was carried out. GSEA 4.2.3 software was used to perform enrichment analysis on the sequencing results.

**EVs in vivo imaging**
According to the experimental design, randomly selected SD rats were sacrificed, and rat tails were collected. EVs in vivo imaging at different time points were performed by using an FX PRO imaging system (Bruker, Germany), and the scanning parameters were set at an excitation wavelength of 550 nm and an emission wavelength of 600 nm.

**Radiological analysis**
According to the experimental design, SD rats were sacrificed, and lumbar or coccygeal samples were collected to perform radiological detections including X-ray, μCT, and MRI. The X-ray-related scanning parameters, including 0.06 s exposure time, 100 cm distance, 160 mA current, and 50 kV voltage, were set by using the DRX Ascend System (Carestream, Canada). The μCT related scanning parameters, including 18 μm pixel size, 100 μA current and 60 kV voltage were set by using a μCT Scanning System (Bruker, Germany) and the three-dimensional structure of the samples was reconstructed by CT-Vox Software (Bruker, Germany). The MR-related scanning parameters, including a fast spin echo sequence, 0.5 mm slice thickness, 3000 ms repetition time, 70 ms echo time and 7.0 T magnetic field intensity, set by using an MRI Scanning System (Bruker, Germany).

**Histological staining**
According to the experimental design, SD rats were sacrificed, and lumbar or coccygeal samples were collected, washed with PBS to remove superficial residues, fixed in 4% paraformaldehyde, decalcified for 1 month, embedded in paraffin and cut into 30 μm slides. After gradient hydration, H&E staining and SO&FG staining were used to assess the degenerative characteristics of lumbar or coccygeal IVD slides. Slides were incubated with primary antibodies and horseradish peroxidase-conjugated or fluorescein-conjugated secondary antibodies, IHC or IF staining was used to explore the expression changes of molecules in coccygeal IVD tissues, and the staining results were observed by microscopy (Olympus, Japan) or fluorescence microscopy (Olympus, Japan).

**Statistics and reproducibility**
We independently performed all the experiments and results more than three times. The continuous data were showed as "Mean ± Standard deviation (SD)" and statistical analysis was conducted by Prism 9.0 software (GraphPad, USA). Two groups of counting data with a normal distribution were tested by Shapiroe Wilk's test and homogeneous variance testing was performed by Levene's variance test, two-tailed $t$ test was used for significance analysis. More than two groups of counting data with a normal distribution were tested by Shapiroe Wilk's test and homogeneous variance testing was performed by Levene's variance test, two-tailed Analysis of variance (ANOVA) was used for significance analysis. All samples were randomly assigned, and analyzed together in each experiment. The investigators were blinded to group allocation during data collection and analysis. No data or samples were excluded from the study.

**Reporting summary**
Further information on research design is available in the Nature Portfolio Reporting Summary linked to this article.

## Data availability
The RNA-sequencing data generated in this study have been deposited in the Gene Expression Omnibus (GEO) database under accession code GSE266883. The interacting protein candidates binding with TREX1 are available in Source Data files. The information of the volunteers enrolled in the study, primer sequences used in PCR genotyping, siRNA sequences used in siRNA transfection and antibodies used in this study are provided in the Supplementary Information (Supplementary Tables 1, 2, 3 and 4). All data supporting the findings of this study are available within the article and its supplementary files. Any additional requests for information can be directed to, and will be fulfilled by, the corresponding authors. Source data are provided with this paper.

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

## Acknowledgements

The work was supported by the National Natural Science Foundation of China (NSFC) (No. 82130072, No. 82072505, No. 82202725) and Natural Science Foundation of Hubei Province (2023AFB805). This work was also supported by Collaborative Innovation Center of Suzhou Nano Science & Technology, the 111 Project and Joint International Research Laboratory of Carbon-Based Functional Materials and Devices, the International Joint Research Center for Intelligent Nano Environmental Protection New Materials and Testing Technology (Grant No. SDGH2108), and the Major Independent Research Project of Jiangsu Key Laboratory for Carbon-Based Functional Materials & Devices (Grant No. L421490022). We thanked BioRender for providing some illustration in this manuscript (Fig. 2A (CC26P9LH2R), Fig. 2K (LE26P9N5TZ), Fig. 4A (RX26P9NI4J), Fig. 5A (MV26P9MB19), and Fig. 5K (HP26P9OMFR)).

## Author contributions

C.Y., Z.W., X.S., and B.W. designed the experiments. W.Z., X.Q., G.L., and X.Z. performed most of the experiments and analyzed the data. H.L., D.W., and Y.S. partly performed the animal experiments. K.Z., K.W., X.F., and L.T. collected the clinical samples. C.Y., Z.W., X.S., B.W., W.Z., X.Q., G.L., and X.Z. wrote this manuscript.

## Competing interests

The authors declare no competing interests.
