## [Peer Review File · Nature Communications]

REVIEWER COMMENTS

Reviewer #1 (Remarks to the Author):

In this manuscript, Zhang et. al. fabricated a self-powered triboelectric-responsive microneedle array driving the controllable release of optogenetically engineered extracellular vesicles (EVs) for intervertebral disc degeneration (IVDD) repair. The self-powered microneedle device combines wearable triboelectric nanogenerator (TENG) with engineered EVs-based targeting molecular modulation, aiming to block etiology-associated IVD degenerative process. Authors revealed the degenerative-associated molecular atlas involved in nucleus pulposus cell senescence via the analysis of transcriptomics, interactomics and gain-/loss-of-function experiment and designed an optogenetically controlled EVs-targeting treatment strategy. Overall, this study is a creative and interesting with a significance of transitional medicine, and the experiments are meticulously performed. Below are just some minor comments to be solved to further improve the content of the manuscript.

1. I'm not sure whether title could include the abbreviation in Nat. Commun. and the abbreviation "EVs" should be replaced by "extracellular vesicles" when it occurs firstly.
2. The character font in the manuscript should be the same and suit the demand of Nat. Commun. such as in line 216, line 223, line 266, line 278 and line 281 and so on.
3. In Fig 1C, authors used H&E staining and SO&FG staining to show histological degenerative levels in different treatments, please use some marks such as asterisk or triangle to focus the degenerative characteristics which help to be understood.
4. cGAS-STING axis-mediated sensing of aberrant cytoplasmic DNA is responsible for triggering the inflammatory response and promoting inflammatory mediator secretion during the cellular senescence process. Why authors chose the ER-associated exonuclease, TREX1, as the master regulator to block the excessive activation of cGAS-STING axis? Please explain.
5. Which method used in Fig 2B, Extended Fig 4D and Extended Fig 5D and other similar heatmaps, and please add it to figure legends.
6. Cell-derived EVs serve as the vector of cell-cell communication via the transfer of membrane-encapsulated cargoes to recipient cells, are suitable for development as a potential therapeutic disease-modified strategy. Please explain why authors used optogenetically engineering technology to deliver targeting cargo into EVs.
7. The radiological method, micro-computer tomography, should be unified in the manuscript using micro-CT or μ CT.
8. In Fig 5D, please mark the MRI-degenerative grades in different treatment in MRI images, which help to be understood.

Reviewer #2 (Remarks to the Author):

In general the manuscript is very interesting providing a new approach to release EVs into the IVD, using the triboelectric generators, a current fashion and attractive tool. However, during the text the final application is somehow confusing since the authors refer several times IVD degeneration and IVD aging which do not overlap in the majority of cases. Also, the peak of LBP is between 40 and 50 years old, which is not considered aged population. The results obtained by in vitro and in vivo analysis do not include age-related models, so it is an over assumption consider this therapy for aged IVD. This should be corrected in abstract, MM, results and discussion.~

Envisaging results translation, how human IVD will be in contact with the microneedles, if the distance to the skin is much higher than in rat tale? This should be properly addressed in the manuscript.

In the materials and methods and figure legends it is not clear the exact number of replicates for all the graphs. The authors just mention data analysis by parametric tests, but this is unlikely with n=3, so in this case non-parametric tests should be used. Important, refer when using paired vs unpaired analysis.

In Fig 5, the images of EVs in rat tales during several weeks suggest increase of EVs concentration during time, which is unlikely. How do the authors interpret this? With such release profile outside the IVD, how these EVs affect the surrounding tissues as muscle or skin?

Also there is a lack of information concerning the inflammatory response associated with the treatment. The authors should provide additional insights regarding the inflammatory response for example by analyzing the draining lymph nodes.

In the end, how do this treatment affect the IVD in a long term? Are the effects sustainable with time, upon stopping the EVs release?

Reviewer #3 (Remarks to the Author):

In their manuscript titled "Self-powered triboelectric-responsive microneedles of optogenetically engineered-EV controllable release for intervertebral disc degeneration repair", by Zhang et al. fabricated an exercise self-powered triboelectric-responsive microneedle array driving the sustainable release of optogenetically engineered EVs for IVDD repair. The authors established a LSI rat model and found that exercise accelerates the degenerative process of IVDs, which involved in cytosolic DNA sensing-mediated inflammatory activation, similar to the degenerative phenotypes of the clinical IVDD process. They also demonstrated the importance of TRAM1-TREX1 complex and NP cells in the development of IVDD, and delivered the TRAM1 protein packaged in optogenetically controlled engineered EV using self-powered triboelectric-responsive MNs to coccygeal IVD rat model, which alleviates the senescence-associated degenerative process of IVDs.

Overall, this is a very interesting and innovative work. This study is well-executed and supported by solid data to validate the results. I believe that the manuscript is a strong candidate for publication in NC, once the points below have been addressed.

Major concerns:

1. In this work, the EXPLOR-engineered EVs were loaded to the triboelectric-responsive MNs through electrochemical polymerization. I wonder if the EV will be lost when the MNs penetrate the skin, and how much EV can be delivered into lesion location. If these MNs loaded with EVs are not ready-to-use, what advantages do they offer compared to directly injecting EVs? When using EVs, there is a risk of off-target effects due to their non-specific distribution. Did the authors examine whether TRAM1 has an impact on non-target tissues in mouse experiment, such as skin, et al? I suggest testing the effects of the protein in non-target tissues and considering the addition of targeted peptides to enhance specificity.
2. The authors chose EXPLOR (CRY2) as an optogenetic tool. As mentioned earlier, pMag/nMag proteins have smaller sizes and more sensitive photo responsive properties. I wonder if fusing with optogenetic protein has any impact on the functionality of TRAM1 itself. If so, protein of small size should be a better choice.
3. In EXPLOR-engineered EVs, it is essential to include a control group without light exposure (Dark). For example, Figure 4C lacks data for the non-illuminated group. I recommend adding this control group and presenting the corresponding data in the manuscript.
4. The authors claimed that their MNs can achieve "sustained release" of EVs (see line 36). It would be beneficial to complement and design specific experiments to substantiate this claim. Similarly,

the authors stated that their EVs "demonstrate excellent delivery efficiency into NP cells" (see line 657). Conducting appropriate experiments to validate this statement would enhance the reliability of their findings.

5. To strengthen the context and background of the study, it is advisable to provide an introduction to the research progress regarding the TRAM1-TREX1 complex and NP cells in the development of IVDD. This will help readers understand the significance of the current study and its contribution to the existing body of knowledge.

6. In Fig. 5, there are several aspects that require clarification. Firstly, the meaning of the arrows in Fig. 5J, Fig. 5L, and Fig. 5N should be clearly labeled to avoid any confusion for readers. Secondly, in Fig. 5K, noticeable differences seem to exist between group TENG-responsive eEVs-MNs and the other groups at 0 weeks. Additionally, differences between group eEVs-MNs and group TENG-responsive nEVs-MNs at 4 weeks are also observed. An explanation of the factors contributing to these differences would be helpful in understanding the results. Thirdly, considering that protein degradation rates are typically rapid, even when packaged in EVs, it is essential to understand how the authors managed to maintain the TRAM1 protein for 4 weeks in EVs with only one operation. A detailed description of the methodology used to achieve consistency in the degradation and supplementation amount of the TRAM1 protein within a four-week period in the group TENG-responsive eEVs-MNs presented in Fig. 5O would be valuable for the readers and researchers in the field.

Minor concerns:

I strongly recommend that the author should rewrite the manuscript carefully, because:

1. The description of Fig. 1B, as well as Extended Data 7 and 8, contains an abundance of repetitive statements, making the presentation unnecessarily lengthy. In particular, the author reiterates certain points multiple times, which could be condensed for a more concise and effective communication of the findings.

2. The manuscript lacks explanations for key abbreviations, such as "eEV" and "nEV". Additionally, the values corresponding to the abbreviations "NEMT group", "TEMT group", and "TEM group" are not provided because the authors did not use these abbreviations in the figures. Moreover, there appears to be a mistake in the usage of the "TEM group" in line 647.

3. In line 322, 326 and 331, there should be Extended Data Fig. 3E, 3F and 3G. Please pretend the similar mistakes.

4. In Fig. 1a, it seems that the author mistakenly used a human as the experimental subject instead of a rat, which could cause confusion or misinterpretation of the results.

5. The authors seem to have confused between "PHK26" and "PKH26." It is crucial to conduct a careful review and rectify this inconsistency to ensure the accuracy of the information presented.

6. There are inconsistencies in the format of references. Such as line 1173, 1186 and 1198, there are three versions of "ACS nano".

7. The authors are expected to include comprehensive materials and methods concerning flow cytometry.

Reviewer #4 (Remarks to the Author):

The manuscript entitled 'Self-powered triboelectric-responsive microneedles of optogenetically engineered-EV controllable release for intervertebral disc degeneration repair' by Zhang et al. describes a multi-layered system with which EVs, engineered to carry the TRAM1 protein can be delivered to nucleus pulposus cells, using a self-powered triboelectric-responsive microneedle platform, in order to alleviate degenerative processes.

Novelty.

To date, EVs from various sources have been used extensively in a multitude of regenerative medicine approaches, including for IVDs (reviewed in doi: 10.3389/fbioe.2020.00311 and doi: 10.1002/adhm.202100596 a.o.). The current work I believe has some novelty in the field, by using an engineered EV type, in-depth characterization of biological samples and a delivery system that may provide some advantages. It would be nice for the authors to also describe whether they observe functional recovery in their animal models. Still, the whole scheme seems aimed at the delivery of TRAM1 protein into coccygeal IVDs. In that respect, the use of a rather complicated system (i.e. EVs) may not have benefit over other protein delivery vehicles or even the 'naked' TRAM1 protein. In itself, the EVs itself seem to have limited function.

Lack of experimental detail provided.

Essential experimental detail is lacking. There is no mention of the EV concentration used in any of the experiments (in vitro nor in vivo). This makes it impossible to tell whether EV numbers used are in any way relevant or achievable. It is not mentioned what number of HEK cells was used for EV production. Again, this makes it impossible to gauge the value of the results provided.

The authors use a transient overexpression in HEK cells of their plasmids. What was the efficacy of transfection? I assume both plasmids need to be taken up by one cell in order for that cell to generate the EVs with both desired proteins? How many cells (and therefore EVs), did not carry the constructs and could this negatively impact the results?

Tracking of native and engineered EVs relied on different labelling strategies. The native EVs were labelled with PKH26 or DiO, the engineered expressed mCherry or EGFP. This is a potential concern as it has recently been shown that the label used can affect results (see doi: 10.1021/acsnano.0c09873). A better control for the engineered EVs could be a truncated/non-functional version of TRAM1 to avoid results being influenced by different labelling strategies. This would at the same time solve the potential issue of the Cry2 protein having bioactivity. This is currently not controlled for.

Potential issues with the EV prep used.

The authors should consider declaring whether they adhere to the MISEV guidelines and what the purity of their EV prep is. As they have used differential centrifugation with no additional purification step to isolate their EVs, there is significant risk of the presence of other, non-EV associated secreted proteins. Furthermore, with EV-depleted serum additional carry-over of proteins is an inherent risk that needs to be assessed, but currently isn't. Involvement of non-EV factors cannot be excluded completely in my opinion.

Minor issues.

In line 511, it is not clear to me whose parts of the body the authors are referring to. Are these the animals (rats) that were used?

In line 616, Fig. 6A and 6B are referenced, but I don't think these exist.

Nomenclature could be more consistent. E.g. Use TEMT and NEMT also in the labels in figure 5.

Amount of data presented in one manuscript.

Though the work reflects an impressive amount of work, reading the paper, I found it challenging to fully comprehend all the information that was provided. It is a complex manuscript with several elements, that makes it challenging to see what the main message and novelty is. I sincerely

wonder whether the authors may have put too much information in the main body of the manuscript.

Language.

The manuscript contains numerous typo's in various places (also in the axes of the figures), which should be addressed. English language should also be improved. In some instances it even hinders understanding with the reader. The abstract comes across as quite incoherent and could do with additional connection between the different topics.

Overall, I think publication in its current form would be premature for this manuscript.

Response to Reviewers

Many thanks for the positive comments, constructive suggestions and kind recommendations from the reviewers! We have carefully checked and revised the manuscript according to their insightful advice, which should significantly improve the clarity and quality of our work. Below is a list of the point-by-point responses to the reviewers' comments shown in *italics* and the corresponding changes that we made highlighted in red.

Reviewer #1 (Remarks to the Author):

In this manuscript, Zhang et. al. fabricated a self-powered triboelectric-responsive microneedle array driving the controllable release of optogenetically engineered extracellular vesicles (EVs) for intervertebral disc degeneration (IVDD) repair. The self-powered microneedle device combines wearable triboelectric nanogenerator (TENG) with engineered EVs-based targeting molecular modulation, aiming to block etiology-associated IVD degenerative process. Authors revealed the degenerative-associated molecular atlas involved in nucleus pulposus cell senescence via the analysis of transcriptomics, interactomics and gain-/loss-of-function experiment and designed an optogenetically controlled EVs-targeting treatment strategy. Overall, this study is a creative and interesting with a significance of translational medicine, and the experiments are meticulously performed. Below are just some minor comments to be solved to further improve the content of the manuscript.

Response:

- Thank the reviewer for your efforts to review our manuscript. We appreciate the reviewer's opposite comments and valuable suggestions, and we are very pleased to revise the manuscript according to your insightful advice. Please read our point-by-point responses below. Thanks again for the reviewer's efforts.

1. I'm not sure whether title could include the abbreviation in Nat. Commun. and the abbreviation "EVs" should be replaced by "extracellular vesicles" when it occurs firstly.

Response:

- Many thanks for the useful suggestions from the reviewer. According to the general

formation requirement for submitted manuscript of *Nat. Commun.*, abbreviations should be defined in the text or legends at their first occurrence, and abbreviations should be used thereafter. We replaced the phrase “EVs” with “extracellular vesicles” to ensure that the concept will be readily intelligible to any scientist.

2. The character font in the manuscript should be the same and suit the demand of Nat. Commun. such as in line 216, line 223, line 266, line 278 and line 281 and so on.

Response:

- Many thanks for the careful review from the reviewer. We are very pleased to correct the character fonts in the manuscript for suiting the requirement of *Nat Commun.* and more readability.

3. In Fig 1C, authors used H&E staining and SO&FG staining to show histological degenerative levels in different treatments, please use some marks such as asterisk or triangle to focus the degenerative characteristics which help to be understood.

Response:

- We appreciate the reviewer for pointing it out. The degenerative histological characteristics contain the changes involved in NP morphology and cellularity, AF morphology and cellularity as well as the border of NP and AF (Ji ML, et. al., *Nat Commun.* 2018 Nov 28;9(1):5051; Li G, et. al., *Nat Commun.* 2022 Mar 18;13(1):1469.). And histologically, H&E staining and SO&FG staining of L4-L5 IVDs revealed that NP cells of LSI-E rats lost vacuolated or stellar-shaped morphology accompanied by cellular chondrogenic proliferation at the interface of NP and AF in the exercise group (Fig. 2C, 2D) (Extended Data Fig. 1C). The collagen lamellae of the AF bulged inwards, and the border between the NP and AF was nebulous, indicating fibrosis of the IVDs with the loss of well-organized NP and AF regional structures (Fig. 2C, 2D) (Extended Data Fig. 1C). According to the insightful advice of reviewer, we added some asterisks to show the large, round cells and cellular chondrogenic proliferation, some triangles to show bulging inwards, thickened AF collagen lamellae and some dashed line to show the border between the NP and AF in Fig. 2C, 2D for clarity.

Figure 2 C) Hematoxylin and eosin (H&E) staining and D) Safranin O and fast green (SO&FG) staining of lumbar 4th-lumbar 5th (L4-L5) IVDs from rats with the indicated treatment. **Asterisks showed the large, round cells and cellular chondrogenic proliferation; triangles showed inwards bulging, thickened AF collagen lamellae and dashed lines showed the border between the NP and AF.** Bar: 1 mm, 100 μ m.

4. cGAS-STING axis-mediated sensing of aberrant cytoplasmic DNA is responsible for triggering the inflammatory response and promoting inflammatory mediator secretion during the cellular senescence process. Why authors chose the ER-associated exonuclease, TREX1, as the master regulator to block the excessive activation of cGAS-STING axis? Please explain.

Response:

- We appreciate the reviewer for pointing it out. In our previous studies, we integrated high-throughput multi-omics technologies with biological experiments, and revealed senescent NP cells harbored excessive cytosolic DNA accumulation and aberrant cGAS-STING axis activation. The sensing of cytosolic accumulated DNA was sufficient for cGAS-STING axis activation and the inflammatory effects formation in driving the inflammatory phenotypic acquisition of senescent NP cells, whereas TREX1 serves as an ER-associated exonuclease that protects cells from excessive cytosolic accumulated DNA fragments and may be a promising therapeutic target to block activation of the inflammatory response during the IVDD process. However, unexpectedly, no marked changes in TREX1 expression were observed between normal and senescent NP cells (Fig. 3G).

TREX1 anchors in the ER membrane via the carboxyl (C)-terminal single-pass transmembrane helix and places the amino (N)-terminal nuclease domain in the cytoplasm. The ER tethering of TREX1 is critical for its nucleolytic activity, and frameshift mutations of the C-terminal domain compromise the association between TREX1 and the ER with no disturbance of the catalytic activity site, which contributes to aberrant activation of cytosolic DNA sensing pathways and abnormal phenotypes of autoimmunity. We wondered whether TREX1 localization could play a critical role in the senescent process of NP cells, and IF staining and western blotting analysis of isolated cytosolic and nuclear components from normal and senescent NP cells showed that TREX1 detachment from the ER and translocation to the nucleus disrupted cytosolic DNA elimination function of TREX1, and worse, promoted the functional transformation of TREX1 to cleave nuclear genomic DNA, which aggravated the cytosolic accumulation of damaged nuclear DNA and aberrant activation of cytosolic DNA-sensing pathway during NP cellular senescence process.

Mechanically, we analyzed differential TREX1 interactomics to identify that TRAM1 might be responsible for regulating TREX1 localization in ER and functional activity. Furthermore, loss-/gain-of-function assays revealed that the disassembly of the TREX1-TRAM1 complex disrupted the translocation of TREX1, which resulted in the exonuclease activity of TREX1 destroying genomic integrity. TRAM1 is essential for the functional regulation of TREX1 tethering in the ER or escape to the nucleus involved in immune elimination or immune damage, which is accordingly a promising target for the inhibition of the inflammatory response during progressive IVDD.

5. Which method used in Fig 2B, Extended Fig 4D and Extended Fig 5D and other similar heatmaps, and please add it to figure legends.

Response:

- We appreciate the reviewer for pointing it out. We used Real-time quantitative PCR (RT-qPCR) technology to detect the expression levels of senescence-associated inflammatory genes, and showed the transcriptional results via heatmaps, which could visualize the differential changes in different treatment groups in Fig 2B and son so. For clarity, we added the experimental methods in the related figure legends.

6. *Cell-derived EVs serve as the vector of cell-cell communication via the transfer of member-encapsulated cargoes to recipient cells, are suitable for development as a potential therapeutic disease-modified strategy. Please explain why authors used optogenetically engineering technology to deliver targeting cargo into EVs.*

Response:

- Cell-derived EVs, with the capacities to bypass biological barriers and deliver bioactive cargoes, are suitable for development as a potential therapeutic strategy for IVDD. Cell-derived EVs potentially mimic the therapeutic effects of cells, yet native EVs lacking the desired cargoes have a poor ability to target master regulators involved in the degenerative process of IVDs. Additionally, an excellent delivery system for targeting EVs to degenerative sites and controllable EV release is important when designing EV-based therapeutic strategies to minimize off-target effects and enhance therapeutic efficacy.

To solve the problems of efficient cargo loading and engineered EVs for biological targeted therapeutics, we applied EXPLOR technology to integrate optically reversible CYR2-CIBN interactions with conjugated TRAM1 protein loading in EVs, which improved TRAM1 protein loading efficiency, drastically expanded the molecule- or pathway-targeting capacity of EVs and enhanced the therapeutic effects of EVs in inhibiting inflammatory response during IVDD.

To solve the problem involved in the controllable delivery system for sustainable release of engineered EV to degenerative sites, we applied TENG to harvest mechanical energy from motion and drive the controllable release of engineered EVs via the varying electrostatic adsorption of PPy-based MNs to fabricating a smart self-powered triboelectric-responsive system to release therapeutic EVs for IVDD.

7. *The radiological method, micro-computer tomography, should be unified in the manuscript using micro-CT or μ CT.*

Response:

- Many thanks for the constructive suggestion from the reviewer. Micro-CT, namely μ CT, is an important radiological technology to evaluate the vertebra, subchondral bone and disc

height of rat coccygeal IVDs. For clarity, we used the abbreviations “ μ CT” to represent this technology “micro-computer tomography” in the manuscript.

8. In Fig 5D, please mark the MRI-degenerative grades in different treatment in MRI images, which help to be understood.

Response:

➤ Many thanks for the constructive suggestion from the reviewer. MRI-based Pfirrmann degenerative grade is the classical radiological grade system of IVDD, which uses the change of T2-weighted signal in IVD tissues to evaluate the water content change and fibrosis degree of NP tissues (Grade I: The structure of the disc is homogeneous, with a bright hyperintense white signal intensity and a normal disc height. Grade II: The structure of the disc is inhomogeneous, with a hyperintense white signal. The distinction between NP and AF is clear, and the disc height is normal, with or without horizontal gray bands. Grade III: The structure of the disc is inhomogeneous, with an intermediate gray signal intensity. The distinction between NP and AF is unclear, and the disc height is normal or slightly decreased. Grade IV: The structure of the disc is inhomogeneous, with a hypointense dark gray signal intensity. The distinction between NP and AF is lost, and the disc height is normal or moderately decreased. Grade V: The structure of the disc is inhomogeneous, with a hypointense black signal intensity. The distinction between NP and AF is lost, and the disc space is collapsed). According to the useful advice of reviewer, we added the MRI-degenerated grade of T2-weighted images in the indicated treatments for clarity.

Figure 5 E) MR T2-weighted images and Pfirrmann MRI grades of coccygeal 7th-coccygeal 8th (C7-C8) IVDs from rats with indicated treatments for 4, 8, and 12 weeks (n = 5).

Reviewer #2 (Remarks to the Author):

In general the manuscript is very interesting providing a new approach to release EVs into the IVD, using the triboelectric generators, a current fashion and attractive tool. However, during the text the final application is somehow confusing since the authors refer several times IVD degeneration and IVD aging which do not overlap in the majority of cases. Also, the peak of LBP is between 40 and 50 years old, which is not considered aged population. The results obtained by in vitro and in vivo analysis do not include age-related models, so it is an over assumption consider this therapy for aged IVD. This should be corrected in abstract, MM, results and discussion.

Response:

- We thank the reviewer so much for the positive comments and valuable suggestions on our work involved in IVDD. We are very pleased to further edit this manuscript according to your insightful advice.

IVDD represents a highly prevalent affliction of adults, which causes LBP and is accompanied by progressively limited activity for patients with a pivotal socioeconomic burden. IVD aging without activation of inflammatory response doesn't contribute to LBP and disability, which might act as the nature phenomenon of human organic aging progression. Globally, discogenic pain and abominable long-term consequences are probably attributable to lack of efficient first-line treatments and excessive use or abuse of radiologic imaging, opioids or surgery. Therefore, understanding of IVDD pathogenesis is urgent for the development of highly relevant biomarkers for sensitive diagnosis and reliably curative targets in disease-modifying treatment. Therefore, our work aimed to investigate the molecular mechanism of IVDD but not IVD aging, and design a targeting treatment strategy for alleviating IVDD. Recently, studies on IVDD have emphasized the senescence of resident NP cell and aberrant NP cell-mediated response to progressive IVD tissue integrity destruction, including senescence-associated phenotype alteration in NP cells and extracellular matrix remodeling. Mechanically, we integrated multi-omics technologies to reveal that cGAS-STING axis-dependent senescent phenotypic acquisition of NP cells promoted inflammatory degeneration of IVDs, and the functional liberation of TREX1 was responsive for inhibiting the activation of cGAS-STING axis and NP cell senescence. Thus,

we designed a smart self-powered triboelectric-responsive system to target the functional liberation of TREX1 for IVDD repair.

As the reviewer said, the peak of LBP is between 40 and 50 years old, which is not considered aged population. Although it isn't claimed that each aged person would suffer with the affliction of LBP and IVDD, aging is the only clear risk factor associated with IVDD incidence and progression because aging accompanied with the accumulated damaged stresses exacerbates the degenerative progression of IVDs. Aging and other stimulations, including oxidative stress and abnormal mechanical loading, contribute to a higher percentage of senescent NP cell clusters, furthermore, NP cellular senescence limits the self-renewal capacity of somatic cells, which leads to the acquisition of senescent phenotype. Limitation of self-renewal potential and continuous affliction of environmental stresses may disrupt the functions of NP cells for maintaining IVD tissue homeostasis, and contribute to IVD inflammatory degeneration and ultimately discogenic pain.

Given the pathological effects of NP cell senescence during IVDD progression, we designed a smart self-powered triboelectric-responsive system for IVDD repair by integrating MNs-based minimally invasive therapeutics with the triboelectric-controlled release of engineered EVs for targeting the aberrant activation of inflammatory response during NP cell senescence progression. For clarity, according to the constructive advice of reviewer, we have corrected some descriptions in abstract, results and discussion, and we have focused on IVDD and NP cell senescence in our manuscript.

Envisaging results translation, how human IVD will be in contact with the microneedles, if the distance to the skin is much higher than in rat tale? This should be properly addressed in the manuscript.

Response:

- Thank the reviewer for the insightful advice. Human IVD tissues are located between two adjacent bony vertebrae near complex nerves and blood vessels, and the degeneration and herniation of IVD tissues lead to progressive neurological deficits, activity limitation, and ultimately disability. The surgical operations of lumbar disc herniation for the resection of degenerative NP tissues don't cure or reverse the degenerative process but only relieve

neurological compression, which leaves a large IVD defect with high reherniation rates and accelerates the degenerative progress of the remaining discectomized IVDs. And the surgical operations may contribute to the neurological or vessel damage due to the complex anatomical structure and high technical requirements. Therefore, we are dedicated to develop the efficient biological therapeutic strategy for targeting master regulators of IVDD, and we successfully designed minimally invasive delivery system via EV-loading MNs. Furthermore, we aimed to promote the minimally invasive therapeutic system to clinical translation, and we would integrate the percutaneous transforaminal endoscopic technology and radiology-guided masking to expose the degenerated IVD tissues for equipping our EV-loading MNs system, namely “Percutaneous transforaminal endoscopic interventional operation (PTEIO)” (Response Fig. 1).

Percutaneous transforaminal endoscopic technology (PTED) is one of these proposed less invasive techniques of spinal surgery (Gadjradj PS, et al. *Bmj* 376, e065846 (2022)). PTED is expected to lead to less postoperative back pain, shorter hospital admission, and a faster recovery because paraspinal muscles are not detached from their insertion, bony anatomy is not changed. The designed operation procedures of PTEIO as following:

Radiological technology via C-arm guides the inserted needle to reach the correct position via transforaminal approach, and the cannula is constructed. Then, the endoscope with the working channels is introduced via the cannula. With the help of endoscope, the operators would expose the degenerated disc, fix the two frictional layers in the two adjoined bony vertebrae and equip the EV-carrying MN system in the degenerated disc to repair the tissues in situ with the triboelectric controlling driven by the movements of bony vertebrae. We are confident that PTEIO will overcome the high distance between skin and IVD tissues, reduce the neurological or vessel damage, and increase the efficacy of EV delivery, which may be a promising strategy for disease-modifying treatment of IVDD.

Response Fig. 1 Percutaneous transforaminal endoscopic interventional operation.

A) Operation illustration of percutaneous transforaminal endoscopic technology. B) Radiology-guided localization via the transforaminal approach. C) The expose of NP tissues, nerves and paravertebral tissues via endoscopic technology.

In the materials and methods and figure legends it is not clear the exact number of replicates for all the graphs. The authors just mention data analysis by parametric tests, but this is unlikely with $n=3$, so in this case non-parametric tests should be used. Important, refer when using paired vs unpaired analysis.

Response:

- We appreciate the reviewer for pointing it out. For clarity, we have added the exact number of replicates of all graphs in figure legends, and corrected the statistical analysis of relevant graphs according to the data characteristics and the requirement of each statistical test.

In Fig 5, the images of EVs in rat tales during several weeks suggest increase of EVs concentration during time, which is unlikely. How do the authors interpret this?

Response:

- We appreciate the reviewer for pointing it out. We are pleased to interpret this, and the EV-based MN delivery system exhibited the capacity of triboelectric-responsive release. We sutured self-powered TENG was to the lumbar skin of the rat, and engineered EV-carrying MNs were assembled with TENGs to fabricate a self-powered triboelectric-responsive controlling system by copper wire connecting the MNs to the electrode layers to drive controllable release of engineered EVs. Rats were placed on the treadmill at a speed of 1

km/min for 30 min every day for 4 weeks after surgery. With the accumulation of exercise time, the amount of EVs release would increase during 4 weeks (Fig. 5C). Upon rats stopped exercise, the fluorescence intensity of masked EVs decreased as time went on, especially after the elution of 8 weeks (Surgical operation at 12 weeks) (Fig. 5C). The increase of EVs concentration during time indicated that triboelectric-responsive controllable release performance of this EV-carrying MN system.

With such release profile outside the IVD, how these EVs affect the surrounding tissues as muscle or skin?

Response:

➤ Many thanks for the insightful comment from the reviewer. Aberrant activity of cGAS-STING axis driven by excessive cytosolic accumulated DNA fragments promotes inflammatory response formation and inflammatory mediator secretion, which has been associated with sterile inflammation and multiple chronic age-associated diseases. TREX1 serves as an ER-associated exonuclease that protects cells from excessive cytosolic accumulated DNA, and TRAM1 is essential for the functional regulation of TREX1 tethering in the ER, which ensures the elimination function of TREX1 for the inhibition of the inflammatory response during progressive IVDD. Theoretically, the engineered EVs for delivering TRAM1 aimed to inhibit the damaged effects of inflammatory response during the progression of IVDD. Expectedly, we also experimentally verified that the EV-carrying MN systems with or without triboelectric stimuli didn't contribute to unexpected inflammatory response in the inserted skin and paravertebral tissues including paravertebral muscles and paravertebral ligaments via histological staining (Extended Data Fig. 12).

Extended Data Fig. 12 EV-carrying MN systems didn't contribute to unexpected inflammatory

response in the inserted skin and paravertebral tissues.

A) Representative images of the H&E staining of skin and subcutaneous tissues from different groups (eEV-MNs and TENG-responsive eEV-MNs). Scale bars, 500 μm , 100 μm . B) Representative images of the H&E staining of paravertebral tissues including paravertebral muscles and paravertebral ligaments from different groups (eEV-MNs and TENG-responsive eEV-MNs). Scale bars, 500 μm , 100 μm .

Also there is a lack of information concerning the inflammatory response associated with the treatment. The authors should provide additional insights regarding the inflammatory response for example by analyzing the draining lymph nodes.

Response:

➤ We appreciate the reviewer for pointing it out. IVD tissue is a hostile microenvironment for native NP cells with high mechanical loading and high osmotic pressure, which is regarded as a unique anatomical area of “immune privilege”, therefore, we focused on the intrinsic regulatory mechanism of inflammatory response within IVD tissues, especially the role of NP cells in the sensing and response of damaged stimulations, and the crosstalk between NP cells and microenvironmental homeostasis of IVD tissues.

Recently, studies on IVDD have emphasized the senescence of resident cell populations and aberrant cell-mediated inflammatory response to progressive IVD tissue integrity destruction. And our work focused on the mechanism involved in inflammatory phenotypic acquisition of senescent NP cells, and aimed to develop a promising therapeutic strategy to inhibit the inflammatory response of NP cells during IVDD progression. We integrated high-throughput multi-omics technologies with biological experiments to reveal that the activation of cGAS-STING axis and the dysfunction of TREX1 triggered the inflammatory response of senescent NP cells, and varied that engineered EVs released from triboelectric-responsive MN system exhibited anti-inflammatory effects and alleviated NP senescence during IVDD. The effector of cGAS-STING axis activation (phosphorylated-STING) and inflammatory mediator (IL-1 β) were obviously decreased in senescent NP cells and needle-puncture-induced rat degenerated IVD tissues via the triboelectric-responsive release of TRAM1-engineered EVs. Additionally, to evaluate whether these EVs affected the

surrounding tissues as muscle or skin, we corrected inserted skin and paravertebral tissues including paravertebral muscles and paravertebral ligaments, and histological staining indicated that the systems didn't contribute to unexpected inflammatory response in the inserted skin and paravertebral tissues. Overall, the repair system would decrease the intrinsic inflammatory response within IVD tissues.

In the end, how do this treatment affect the IVD in a long term? Are the effects sustainable with time, upon stopping the EVs release?

Response:

- Many thanks for the constructive suggestion from the reviewer. According to the insightful advice, we are pleased to evaluated the therapeutic efficacy of EV-carrying MN system in a long term. And we also verified that the TRAM1-carrying EV treatment exhibited the anti-degenerative effects, including improved historical structure and reduced senescent phenotype of coccygeal IVDD in a long term (12 weeks at least), although the anti-degenerative effects and the accumulation of engineered EVs were decreasing progressively (Fig. 5) (Extended Data Fig. 13).

Fig. 5 Self-powered triboelectric-responsive MN system alleviated the senescence-associated degenerative process of IVDs.

A) Schematic workflow for the needle puncture-induced degeneration of rat coccygeal IVDs treated with triboelectric-responsive MNs ($n = 5$). B) Gross pictures of coccygeal IVDs for needle puncture model wearing with different equipments (eEV-MNs, TENG-responsive nEV-MNs, TENG-responsive eEV-MNs) ($n = 5$). C) Representative X-ray images and EVs in vivo imaging showing the EV release kinetics in coccygeal IVDs at different time points. D) Representative μ CT images of coccygeal IVDs at different time points. E) MR T2-weighted images and Pfirrmann MRI grades of coccygeal 7th-coccygeal 8th (C7-C8) IVDs from rats with indicated treatments for 4, 8, and 12 weeks ($n = 5$). F) SO&FG staining of C7-C8 IVDs from rats with the indicated treatments for 4, 8, and 12 weeks ($n = 5$). G) IF staining of PKH26-labelled EVs or mCherry in C7-C8 IVDs from rats with the indicated treatments for 4, 8, 12 weeks ($n = 5$). The

arrows indicated the PKH26⁺ or mCherry⁺ NP cells. H) IHC staining of TRAM1 in C7-C8 IVDs from rats with the indicated treatments for 4, 8 and 12 weeks (n = 5). The arrows indicated TRAM1⁺ NP cells. I) IF staining of p-STING in C7-C8 IVDs from rats with the indicated treatments for 4, 8 and 12 weeks (n = 5). J) Pfirrmann MRI grades of C7-C8 IVDs from rats with indicated treatments for 4, 8, and 12 weeks (n = 5). K) Histological score of C7-C8 IVDs from rats with the indicated treatments for 4, 8, and 12 weeks (n = 5). L) Quantitative analysis of PKH26⁺ or mCherry⁺ NP cell proportions in C7-C8 IVDs from rats with the indicated treatments for 4, 8 and 12 weeks (n = 5). M) Quantitative analysis of the TRAM1⁺ NP cell proportion in C7-C8 IVDs from rats with the indicated treatments for 4, 8 and 12 weeks (n = 5). N) Quantitative analysis of the fluorescence intensity of p-STING in C7-C8 IVDs from rats with the indicated treatments for 4, 8 and 12 weeks (n = 5). All the experiments and results performed independently more than three times and number of asterisk (*) presented significant p value: * p value < 0.05, ** p value < 0.01, *** p value < 0.001, ns. not significant.

Extended Data Fig. 13 Self-powered triboelectric-responsive MN system alleviated the senescence and inflammatory response of NP cells in rat coccygeal IVDD.

A) H&E staining of C7-C8 IVDs from rats with the indicated treatments for 4, 8, 12 weeks (n = 5).
B) IF staining of type II collagen in C7-C8 IVDs from rats with the indicated treatments for 4, 8 and 12 weeks (n = 5). C) IHC staining of p-p53 in C7-C8 IVDs from rats with the indicated treatments for 4, 8 and 12 weeks (n = 5). The arrows indicated p-p53⁺ NP cells. D) IF staining of IL-1 β in C7-C8 IVDs from rats with the indicated treatments for 4, 8 and 12 weeks (n = 5). E) Quantitative analysis of the p-p53⁺ NP cell proportion in C7-C8 IVDs from rats with the indicated treatments for 4, 8 and 12 weeks (n = 5). F) Quantitative analysis of the fluorescence intensity of IL-1 β in C7-C8 IVDs from rats with the indicated treatments for 4, 8 and 12 weeks (n = 5). All the experiments and results performed independently more than three times and number of asterisk (*) presented significant p value: * p value < 0.05, * * p value < 0.01, * * * p value < 0.001, ns. not significant.

Reviewer #3 (Remarks to the Author):

In their manuscript titled "Self-powered triboelectric-responsive microneedles of optogenetically engineered-EV controllable release for intervertebral disc degeneration repair", by Zhang et al. fabricated an exercise self-powered triboelectric-responsive microneedle array driving the sustainable release of optogenetically engineered EVs for IVDD repair. The authors established a LSI rat model and found that exercise accelerates the degenerative process of IVDs, which involved in cytosolic DNA sensing-mediated inflammatory activation, similar to the degenerative phenotypes of the clinical IVDD process. They also demonstrated the importance of TRAM1-TREX1 complex and NP cells in the development of IVDD, and delivered the TRAM1 protein packaged in optogenetically controlled engineered EV using self-powered triboelectric-responsive MNs to coccygeal IVD rat model, which alleviates the senescence-associated degenerative process of IVDs.

Overall, this is a very interesting and innovative work. This study is well-executed and supported by solid data to validate the results. I believe that the manuscript is a strong candidate for publication in NC, once the points below have been addressed.

Response:

- We thank the reviewer so much for the positive comments and valuable suggestions on our work involved in self-powered triboelectric-responsive MN array for IVDD repair. We are very pleased to further edit this manuscript according to your insightful advice. Based on the constructive comments, we have carefully done some experiments and revised the manuscript with all changes highlighted in red. Please read our point-by-point responses below. Thank again for giving us this chance to revise our study.

Major concerns:

1. In this work, the EXPLOR-engineered EVs were loaded to the triboelectric-responsive MNs through electrochemical polymerization. I wonder if the EV will be lost when the MNs penetrate the skin, and how much EV can be delivered into lesion location. If these MNs loaded with EVs are not ready-to-use, what advantages do they offer compared to directly injecting EVs?

Response:

- Thank the reviewer for your insightful opinions. Compared to direct injection of engineered EVs, we inferred that EV-carrying MN array exhibited the following advantages:

Firstly, triboelectric-responsive EV controllable release. Stimuli-responsive EV release system driven by exogenous physical signals provides the controllable release of EVs for highly efficient and sustainable applications. Cell-derived EVs are membrane-delimited particles constructed by an anionic phospholipid bilayer, which indicates that EVs have potential electrical properties. Given that PPy has an excellent capacity to load and release drugs by switching between the oxidative and reductive states, we utilized a TENG to convert biomechanical energy from motion to electrical energy, which drove the loss of electrostatic attraction between anionic EVs and the PPy layer in the reductive state, and thereby fabricated a smart self-powered triboelectric-responsive system to release therapeutic EVs for IVDD.

Secondly, minimally invasive delivery to pathological sites for biological targeting treatment. Cell-derived EVs are suitable for development as a potential therapeutic strategy for IVDD, and an excellent delivery system for targeting EVs to degenerative sites and responsive triggers for EV controllable release are key considerations when designing EV-based therapeutic strategies to minimize off-target effects and enhance therapeutic efficiency. MNs serve as a promising technology for minimally invasive transdermal drug delivery with the capacity to deliver transdermally administered drugs either as coated or encapsulated cargo during the insertion process or via convective effects in the field of macromolecule distribution and vaccine self-administration. Stimuli-responsive MNs which respond to physiological factors or external physical stimuli expand the potential of controllable drug release for “site-to-site” treatment with a high efficiency and an excellent utilization, resulting in the current trend in drug and gene delivery. Therefore, we integrated EV-based bioactive molecule delivery, stimuli-responsive MNs and triboelectric technology to fabricate self-powered IVDD repair system.

- It was difficult to measure the accurate concentrations of EVs in rat coccygeal IVDs due to lack of methods, but EV dynamics detected by in vivo imaging confirmed the enrichment and accumulation of native EVs and engineered EVs in rat coccygeal NP tissues, and showed the release characteristic of EVs from triboelectric-responsive MNs. Additionally, IF staining

showed NP cells had similar uptake rates of native EVs and engineered EVs. Multi-experimental technologies have verified that the therapeutic efficiency of TRAM1-engineered EVs in alleviating IVDD. We confirmed that the successful delivery of engineered EVs into rat coccygeal IVDDs, and focused on the therapeutic efficacy of TRAM1-engineered EVs in alleviating IVDD progression (Fig. 5) (Extended Data Fig. 13).

When using EVs, there is a risk of off-target effects due to their non-specific distribution. Did the authors examine whether TRAM1 has an impact on non-target tissues in mouse experiment, such as skin, et al?

Response:

➤ Thank the reviewer for pointing out it. Theoretically, the engineered EVs for delivering TRAM1 aimed to inhibit the damaged effects of inflammatory response. The sensing of cytosolic accumulated DNA via cGAS-STING axis activation promotes inflammatory response formation in various sterile inflammation. TREX1 serves as an ER-associated exonuclease that protects cells from excessive cytosolic accumulated DNA fragments, and TRAM1 is essential for the functional regulation of TREX1 tethering in the ER, which ensures the elimination function of TREX1 for the inhibition of the inflammatory response during progressive IVDD. Non-target tissues including the inserted skin and paravertebral tissues didn't harbor an inflammatory imbalance state, and we also experimentally verified that the EV-carrying MN systems with or without triboelectric stimulation didn't contribute to unexpected inflammatory response in the inserted skin and paravertebral tissues including paravertebral muscles and paravertebral ligaments via histological staining (Extended Data Fig. 12).

Extended Data Fig. 12 EV-carrying MN systems didn't contribute to unexpected inflammatory

response in the inserted skin and paravertebral tissues.

A) Representative images of the H&E staining of skin and subcutaneous tissues from different groups (eEV-MNs and TENG-responsive eEV-MNs). Scale bars, 500 μm , 100 μm . B) Representative images of the H&E staining of paravertebral tissues including paravertebral muscles and paravertebral ligaments from different groups (eEV-MNs and TENG-responsive eEV-MNs). Scale bars, 500 μm , 100 μm . All the experiments and results performed independently more than three times.

I suggest testing the effects of the protein in non-target tissues and considering the addition of targeted peptides to enhance specificity.

Response:

➤ Many thanks for the constructive suggestions from the reviewer. There lacks of high-specific molecular markers of NP cells, and the functional elucidation of NP cell subpopulations is uncovered, contributing to a problem involved in targeting NP cells for IVDD treatment. It's difficult to use the high-specific targeted peptides to concentrate therapeutic EVs in NP tissues. Thus, we investigated the master molecular mechanism of IVDD pathogenesis and designed an excellent delivery strategy to enhance the enrichment of targeting EVs in degenerated NP tissues. Overall, we applied high-throughput multi-omics technologies to excavate highly relevant biomarkers for reliably curative targets in disease-modifying treatment and fabricated triboelectric-responsive MNs for targeting EV delivery and EV controllable release to minimize off-target effects and enhance therapeutic efficacy. TRAM1-engineered EVs exhibited the capacity of anti-inflammatory and remodeled the elimination function of TREX1 to cytosolic damaged DNA for alleviating IVDD progression with no obvious negative effects on surrounding tissues.

2. The authors chose EXPLOR (CRY2) as an optogenetic tool. As mentioned earlier, pMag/nMag proteins have smaller sizes and more sensitive photo responsive properties. I wonder if fusing with optogenetic protein has any impact on the functionality of TRAM1 itself. If so, protein of small size should be a better choice.

Response:

- Many thanks for the constructive suggestion from the reviewer. EXPLOR technology, defined as optogenetically controlled engineered EV biogenesis, integrates optically reversible protein-protein interactions with targeting proteins for EV loading, which could effectively deliver soluble proteins into the recipient cells. Researchers successfully utilized the optogenetical reversible conjugation of CRY2 and CIBN, fused with Bax, Cre or super-repressor IκB (srIκB) protein (CRY2-conjugated mCherry-tagged Bax/Cre/srIκB) and the EV surface marker CD9 (CIBN-conjugated EGFP-tagged CD9) to guide Bax/Cre/srIκB protein to load into EVs. Fusing with optogenetic protein didn't have any effects on the functionality of proteins themselves, and these proteins also exhibited themselves physiological functions for disease-modifying treatment in vitro or in vivo (Sheller-Miller S, et al. Science advances 7, (2021).; Kim S, et al. Kidney Int 100, 570-584 (2021).; Yim N, et al. Nature Communications. 7, 12277 (2016)).

We designed the recombinant protein CRY2 fused with TRAM1 (CRY2-conjugated mCherry-tagged TRAM1) and CIBN fused with CD9 (CIBN-conjugated EGFP-tagged CD9) to guide TRAM1 protein to load into EVs via the localization of CD9 to the inner surface of the early endosomes. Western blotting analysis was performed to confirm the packaging of TRAM1 into engineered EVs. And we experimentally varied that the incubation of TRAM1-engineered EVs drastically inhibited inflammatory phenotype acquisition of senescent NP cells accompanied with reduced cytosolic accumulated DNA and less activation of cGAS-STING axis. Furthermore, the administration of TRAM1-engineered EVs alleviated the nuclear translocation of TREX1 and ensured the exhibition of a protective effect of TREX1, whose protective effects was similar to those of TRAM1-overexpression treatment. Additionally, engineered EVs released from triboelectric-response MNs also exhibited the anti-inflammatory effects and anti-degenerated effects to alleviate NP cell senescence and IVDD progression. Expectedly, native EVs or engineered EVs derived from plasmid-transfected cells without light exposure didn't exhibit the protective effects to avoid the confounding factors from cargoes of EVs, optogenetic proteins and triboelectric effects (Extended Data Fig. 11 and Fig.14).

3. In EXPLOR-engineered EVs, it is essential to include a control group without light exposure

(Dark). For example, Figure 4C lacks data for the non-illuminated group. I recommend adding this control group and presenting the corresponding data in the manuscript.

Response:

- Many thanks for the constructive suggestion from the reviewer. According to the insightful advice of reviewer, we have added this control group and presented the corresponding data in the relevant parts. We verified that the lack of illuminated stimulation failed to load the TRAM1 into CIBN-CD9-engineered EVs, and the delivery of TRAM1 protein into senescent NP cells was essential for the therapeutic effects of optogenetically engineered EV-carrying MNs for and inhibiting NP cell senescent acquisition in vitro and alleviating the degenerative progression of coccygeal IVDs in vivo, which avoided the effects of cofounding factors of illuminated stimulation and photoreactive proteins (Extended Data Fig. 11 and Fig.14).

Extended Data Fig. 11 Illumination-induced TRAM1 loading into EVs was essential for the therapeutic effects of engineered EVs for remodeling TREX1 function and inhibiting NP cell senescent phenotypic acquisition.

A) Representative western blotting images of CRY2-mCherry-TRAM1, CIBN-EGFP-CD9 and EV markers in HEK-293T cells and isolated engineered EVs from HEK-293T cells after cotransfection with CIBN-EGFP-CD9 and CRY2-mCherry-TRAM1 plasmids with (“Blue light ON”) or without (“Blue light OFF”) 460-nm laser stimulation (20 μ W/cm²). B) Representative IF staining images of EGFP and mCherry in P8 NP cells after treatment with engineered EVs for 72 h, bar: 20 μ m. C) Flow cytometry images to analyze the uptake of EVs in P8 NP cells after treatment with engineered EVs for 72 h. D) Representative western blotting images of

CRY2-mCherry-TRAM1, CIBN-EGFP-CD9 from P8 NP cells after treated with PBS or engineered EVs for 72 h. E) Representative western blotting images of p-p53 and p21 in P8 NP cells after treated with PBS or engineered EVs for 72 h. F) SA- β -gal staining and G) quantitative analysis of SA- β -gal⁺ NP cell proportion in P8 NP cells after treated with PBS, native or engineered EVs for 72 h, bar: 200 μ m. H) Differential expression heatmap of SASP in P8 NP cells after treated with PBS or engineered EVs for 72 h. I) Representative western blotting and J) quantitative analysis of cytosolic and nuclear TREX1 protein in P8 NP cells after treated with PBS or engineered EVs for 72 h. K) Representative agarose gel electrophoresis images and L) quantitative analysis of DNA fragments from the cytosolic component in P8 NP cells after treatment with PBS or engineered EVs for 72 h. M) Representative western blotting images of cGAS, STING and γ H2A in P8 NP cells after treated with PBS or engineered EVs for 72 h. N) Representative western blotting images of CRY2-mCherry-TRAM1 protein from P8 NP cells after cultured in the medium with engineered EV-carrying MNs connected to TENG with different triboelectric stimulation durations. O) Representative electrophoresis images of DNA fragments from the cytosolic component in P8 NP cells after culture in medium with engineered EV-carrying MNs connected to TENGs with different triboelectric stimulation durations. P) Representative western blotting images of cGAS, STING and γ H2A in P8 NP cells after cultured in the medium with engineered EV-carrying MNs connected to TENG with different triboelectric stimulation durations. Q) Representative western blotting images of p-p53 and p21 in P8 NP cells after cultured in the medium with engineered EV-carrying MNs connected to TENG with different triboelectric stimulation durations. R) Differential expression heatmap of SASP in P8 NP cells after cultured in the medium with engineered EV-carrying MNs connected to TENG with different triboelectric stimulation durations. All the experiments and results performed independently more than three times and number of asterisk (*) presented significant p value: * p value < 0.05, ** p value < 0.01, *** p value < 0.001, ns. not significant.

Extended Data Fig. 14 Illumination-induced TRAM1 loading into EVs was essential for the therapeutic effects of engineered-EV carrying MN system for alleviating needle puncture-induced rat coccygeal IVDD.

A) Isolated engineered EVs from HEK-293T cells after cotransfection with CIBN-EGFP-CD9 and CRY2-mCherry-TRAM1 plasmids with (“Blue light ON”) or without (“Blue light OFF”) 460-nm laser stimulation ($20 \mu\text{W}/\text{cm}^2$) were loaded into triboelectric-responsive MN system. And rat coccygeal IVDs for needle puncture model wearing with different equipments (TENG-responsive “Blue light ON” EV-MNs or TENG-responsive “Blue light OFF” EV-MNs) ($n = 5$) were performed. A) H&E staining of C7-C8 IVDs from rats with the indicated treatments for 4, 8, 12 weeks ($n = 5$). B) SO&FG staining of C7-C8 IVDs from rats with the indicated treatments for 4, 8 and 12 weeks ($n = 5$). C) IHC staining of TRAM1 in C7-C8 IVDs from rats with the indicated

treatments (n = 5) for 4, 8 and 12 weeks. D) IHC staining of p-p53 in C7-C8 IVDs from rats with the indicated treatments for 4, 8 and 12 weeks (n = 5). G) IF staining of PKH26-labelled EVs and mCherry in C7-C8 IVDs from rats with the indicated treatments for 4, 8, 12 weeks (n = 5). E) IF staining of type II collagen in C7-C8 IVDs from rats with the indicated treatments for 4, 8 and 12 weeks (n = 5). F) IF staining of EGFP-CD9 in C7-C8 IVDs from rats with the indicated treatments for 4, 8, 12 weeks (n = 5). G) IF staining of p-STING in C7-C8 IVDs from rats with the indicated treatments for 4, 8 and 12 weeks (n = 5). H) Quantitative analysis of the fluorescence intensity of p-STING in C7-C8 IVDs from rats with the indicated treatments for 4, 8 and 12 weeks (n = 5). I) Histological score of C7-C8 IVDs from rats with the indicated treatments for 4, 8, and 12 weeks (n = 5). J) Quantitative analysis of the TRAM1⁺ NP cell proportion in C7-C8 IVDs from rats with the indicated treatments for 4, 8 and 12 weeks (n = 5). K) Quantitative analysis of the p-p53⁺ NP cell proportion in C7-C8 IVDs from rats with the indicated treatments for 4, 8 and 12 weeks (n = 5). L) Quantitative analysis of EGFP-CD9⁺ NP cell proportions in C7-C8 IVDs from rats with the indicated treatments for 4, 8 and 12 weeks (n = 5). M) Quantitative analysis of the fluorescence intensity of p-STING in C7-C8 IVDs from rats with the indicated treatments for 4, 8 and 12 weeks (n = 5). N) Quantitative analysis of the fluorescence intensity of IL-1 β in C7-C8 IVDs from rats with the indicated treatments for 4, 8 and 12 weeks (n = 5). All the experiments and results performed independently more than three times and number of asterisk (*) presented significant p value: * p value < 0.05, ** p value < 0.01, *** p value < 0.001, ns. not significant.

4. The authors claimed that their MNs can achieve "sustained release" of EVs (see line 36). It would be beneficial to complement and design specific experiments to substantiate this claim.

Response:

- We appreciate the reviewer for pointing it out. We sutured the self-powered TENG to the lumbar skin of the rat, and engineered EV-carrying MNs were assembled with TENGs to fabricate a self-powered triboelectric-responsive controlling system, which converted lumbar exercise-induced mechanical force to triboelectricity to drive controllable release of engineered EVs. Rats were placed on the treadmill at a speed of 1 km/min for 30 min every day for 4 weeks after surgery. With the accumulation of exercise time, the amount of EV release would increase during 4 weeks (Fig. 5C). Upon rats stopped exercise, the

fluorescence intensity of masked EVs decreased as time went on, especially after the elution of 8 weeks (Surgical operation at 12 weeks) (Fig. 5C). The increase of EV concentration during time indicated that triboelectric-responsive sustained release of EVs of this MNs system. Similarly, histochemical staining confirmed that the number of TRAM1⁺ and mCherry-labelled cells increased during 4 weeks, and decreased as the elution time went on, upon the exercise stopped (Fig. 5G and 5H). Collectively, the time-dependent imaging and histochemical change indicated the "sustained release" of EVs.

Figure 5 C) Representative X-ray images and EVs in vivo imaging showing the EV release kinetics in coccygeal IVDs at different time points. G) IF staining of PKH26-labelled EVs or mCherry in C7-C8 IVDs from rats with the indicated treatments for 4, 8, 12 weeks (n = 5). The arrows indicated the PKH26⁺ or mCherry⁺ NP cells. H) IHC staining of TRAM1 in C7-C8 IVDs from rats with the indicated treatments for 4, 8 and 12 weeks (n = 5). L) Quantitative analysis of PKH26⁺ or mCherry⁺ NP cell proportions in C7-C8 IVDs from rats with the indicated treatments

for 4, 8 and 12 weeks (n = 5). M) Quantitative analysis of the TRAM1⁺ NP cell proportion in C7-C8 IVDs from rats with the indicated treatments for 4, 8 and 12 weeks (n = 5).

Similarly, the authors stated that their EVs "demonstrate excellent delivery efficiency into NP cells" (see line 657). Conducting appropriate experiments to validate this statement would enhance the reliability of their findings.

Response:

- Thank the reviewer for the constructive advice. To evaluate the therapeutic effects of different delivery strategies including TRAM1 protein injection, lentivirus (LV) transfection and EV-based delivery, we directly injected exogenous GST-tagged TRAM1 protein into rat coccygeal NP tissues or used TRAM1-overexpressing plasmid-carrying LVs to transfect coccygeal NP cells (Extended Data Fig. 15A-E). It was difficult that the exogenously injected TRAM1 protein to be stored in NP tissues and exogenous protein administration failed to deliver TRAM1 into NP cells and alleviate needle puncture-induced degeneration of coccygeal IVDs (Extended Data Fig. 15A-E). LV transfection exhibited an efficient delivery effects for improving the number of TRAM1⁺ NP cells and migrating the coccygeal IVDD (Extended Data Fig. 15A-E). Interestingly, via IHC staining, we found that the TRAM1⁺ NP cells increased until 8 weeks, and subsequently decreased at 12 weeks, and we inferred the increasement of TRAM1 gene copies limited the ability to upregulate the expression of TRAM1 and remodel the function of TRAM1 because the endogenous etiological factors driving the progression of degeneration were not eliminated and the endogenous molecular mechanism involved in TRAM1 deficiency or dysfunction was still unknown (Extended Data Fig. 15A-E). Similarly, the EV-based strategy exhibited the ability to delivering functional protein to NP cells to overcome the challenges associated with protein delivery and the limited efficiency of LVs in regulating endogenous protein expression (Extended Data Fig. 15D-E).

Extended Data Fig. 15 The therapeutic efficacy of different delivery strategies for alleviating rat coccygeal IVDD.

To compare the delivery efficacy of different delivery strategies, we used exogenous GST-tagged TRAM1 protein solution or TRAM1-overexpressing plasmid-carrying LVs to inject into the NP tissues of needle puncture-induced rat coccygeal IVDD model. A) H&E staining of C7-C8 IVDs from rats with the indicated treatments for 4, 8, 12 weeks (n = 5). B) SO&FG staining of C7-C8 IVDs from rats with the indicated treatments for 4, 8 and 12 weeks (n = 5). B) IHC staining of TRAM1 in C7-C8 IVDs from rats with the indicated treatments for 4, 8 and 12 weeks (n = 5). C) IF staining of GST or GFP in C7-C8 IVDs from rats with the indicated treatments for 4, 8, 12 weeks (n = 5). D) Histological score of C7-C8 IVDs from rats with the indicated treatments for 4, 8, and 12 weeks (n = 5). E) Quantitative analysis of the TRAM1⁺ NP cell proportion in C7-C8 IVDs from rats with the indicated treatments for 4, 8 and 12 weeks (n = 5). All the experiments and results performed independently more than three times and number of asterisk (*) presented significant p value: * p value < 0.05, ** p value < 0.01, *** p value < 0.001, ns. not significant.

5. To strengthen the context and background of the study, it is advisable to provide an introduction to the research progress regarding the TRAM1-TREX1 complex and NP cells in the

development of IVDD. This will help readers understand the significance of the current study and its contribution to the existing body of knowledge.

Response:

- Many thanks for the constructive suggestion from the reviewer. In this study, focusing on the research mode of “phenotype-mechanism-treatment”, we utilized high-throughput multi-omics technologies with biological experiments to explore the master regulators in promoting the progression of IVDD, and designed exercise-driven repair system for targeting disease-modifying of IVDD via the integration of triboelectric effects and minimally invasive delivery.

IVDs are regional and avascular fibrocartilaginous tissues located between two adjacent bony vertebrae, and the highly elastic NP tissue is located in the center of IVD where it is constrained by the concentrically arranged AF collagen lamellae, which acts as hydrogel-like adapters that withstand circumferential loads and constitutes a signaling hub to maintain the structural and functional integrity of an IVD. Native NP cells exhibit crucial functions for regulating extracellular matrix homeostasis, constructing an accommodating biomechanical environment and maintaining the gelatinous property of NP tissue. The accumulation of senescent NP cells due to aging and other damaged factors is a distinctive hallmark of the initiation and progression of IVDD.

Recently, studies on IVDD have emphasized the senescence of resident cell populations and aberrant cell-mediated response to progressive IVD tissue integrity destruction. Various stresses, including oxidative stress, abnormal mechanical loading and inflammatory factors, abrogate biomechanical homeostasis and trigger a cell-mediated repair response, whereas cellular senescence limits the self-renewal capacity of somatic cells, leading to the acquisition of senescent phenotype. Limitation of self-renewal potential and continuous affliction of environmental stresses may contribute to a higher percentage of senescent NP cell clusters, failure to maintain IVD tissue homeostasis, inflammatory degeneration and ultimately discogenic pain. Previous studies on NP senescence have focus on aberrant activation of inflammatory reaction cascades involved in sensing senescence-associated patterns and senescence-induced gene expression prolife changes on the pre- or post-translational modification. In this study, we verified that exercise promoted the

senescent phenotypic acquisition of NP cells via excessive cytosolic damaged DNA accumulation and activation of cGAS-STING axis, and accelerated the degeneration of lumbar IVDs from surgical lumbar instability, similar to the clinical degeneration process. TREX1 serves as an ER-associated exonuclease that protects cells from excessive cytosolic accumulated DNA fragments and inhibits the activation of cGAS-STING axis, which may be a promising therapeutic target to block activation of the inflammatory response during the IVDD process. However, unexpectedly, no marked changes of TREX1 expression were observed between normal and senescent NP cells. Additionally, the administration of TREX1 wide-type plasmids increased the level of genomic DNA damage in senescent NP cells but not promoted the elimination of cytosolic damaged DNA. To further investigate the mechanism involved in the dysfunction of TREX1, we found the ER membrane tethering of TREX1 was critical for its nucleolytic activity, and TREX1 deanchoring from the ER accompanied with translocation to the nucleus were responsible for the immune elimination dysfunction of TREX1 and aberrant activation of the cytosolic DNA-sensing pathway involved in NP inflammatory senescence. Mechanistically, we performed endogenous TREX1 interactomics from normal and senescent NP cells, and revealed that disassembly of the TREX1-TRAM1 complex promoted the detachment in ER membrane and nuclear translocation of TREX1, which results in the exonuclease activity of TREX1 destroying genomic integrity. Summarily, TRAM1 is a promising target for remodeling the function of TREX1 and inhibiting the inflammatory response of senescent NP cells during progressive IVDD. An appropriately selective vector is necessary for an excellent gene and drug delivery system to deliver target molecules stably to a specific site. EVs, with their suitable biocompatibility, minimal immunogenicity and intrinsic capacity to target tissues, perform a critical function in transferring bioactive molecules to recipient cells, which has been highlighted as a promising strategy for the delivery of proteins and nucleotides in disease-targeting therapeutics. Therefore, we used EXPLOR technology to load TRAM1 protein to engineered EVs and fabricated triboelectric-responsive MNs to delivery therapeutic EVs.

For clarity, we have added some research progress involved in the TRAM1-TREX1 complex and NP cell senescence during the development of IVDD, according to the insightful advice

of the reviewer.

6. In Fig. 5, there are several aspects that require clarification. Firstly, the meaning of the arrows in Fig. 5J, Fig. 5L, and Fig. 5N should be clearly labeled to avoid any confusion for readers. Secondly, in Fig. 5K, noticeable differences seem to exist between group TENG-responsive eEVs-MNs and the other groups at 0 weeks. Additionally, differences between group eEVs-MNs and group TENG-responsive nEVs-MNs at 4 weeks are also observed. An explanation of the factors contributing to these differences would be helpful in understanding the results.

Response:

- We appreciate the reviewer for pointing it out. Firstly, the arrows in Fig. 5J (Revised Extended Fig. 13C) indicated the p-p53-stained positive NP cells (dark staining for eEV-MNs group and TENG-responsive nEV-MNs group, light staining for TENG-responsive eEV-MNs group), the arrows in Fig. 5L (Revised Fig. 5G) indicated PKH26- or mCherry-stained positive NP cells (PKH26 staining for TENG-responsive nEV-MNs, mCherry staining for TENG-responsive eEV-MNs), and the arrows in Fig. 5N (Revised Fig. 5H) indicated the TRAM1-stained positive NP cells (light staining for eEV-MNs group and TENG-responsive nEV-MNs group, dark staining for TENG-responsive eEV-MNs group). For clarify, we have corrected some orders of Fig 5 panels and labell the meaning of the arrows in the relevant panels to avoid any confusion for readers. Secondly, p-p53 is the key protein of cell cycle arrest, acting as a marker of cellular senescence, and we performed IHC staining to evaluate the senescent degrees of NP cells during rat coccygeal IVDD. The staining of p-p53 at 0 week is darked in TENG-responsive eEV-MNs group, and that at 4 weeks is lighter in TENG-responsive eEV-MNs group, which indicated the TRAM1-carring EVs released from triboelectric-responsive MNs could obviously decrease the expression of p-p53, even the expression baseline of p-p53 at 0 week in TENG-responsive eEV-MNs group was highest. Thirdly, the experimental process was kept the same in the single molecule staining, and the internal heterogeneity of rat coccygeal IVD samples such as water content, fibrosis level and captured regions might affect the results of staining. To eliminate the effects of confounding factors from a single molecule and a single

experiment, we used multi-experimental technologies and multi-molecules analysis to evaluate the therapeutic efficiency of different treatments.

Thirdly, considering that protein degradation rates are typically rapid, even when packaged in EVs, it is essential to understand how the authors managed to maintain the TRAM1 protein for 4 weeks in EVs with only one operation. A detailed description of the methodology used to achieve consistency in the degradation and supplementation amount of the TRAM1 protein within a four-week period in the group TENG-responsive eEVs-MNs presented in Fig. 50 would be valuable for the readers and researchers in the field.

Response:

➤ Many thanks for the reviewer for point out this. Our pre-experiments reveal that TRAM1 protein degraded spontaneously, and when EVs-carrying MNs were left at room temperature for 11 days, the amount of protein decreased obviously in EVs released from the MNs (Response Fig. 2A). Additionally, the TRAM1-engineered EVs failed to remodeling the cytosolic localization of TREX1, which were released from the MNs were left at room temperature for 7 days (Response Fig. 2B). As the reviewer said, TRAM1 protein packaged in EVs could maintain its amount and physiologic function for a short period, 7 days. To ensure the therapeutic efficacy, we replaced a new eEV-MNs for 7 days, and the old MNs were used to re-electropolymerize for EVs loading after cleaning, indicating that the PLA-based MNs could be sustainably recycled with reduced carbon emission to suit development of human civilization, which acted as an advantage of “renewability” for our repair system.

Response Fig. 2 The characteristics involved in the degradation and bioactivity of recombinant TRAM1 protein packaged in engineered EVs released from MNs.

Engineered EV-carrying MNs were left at room temperature with the time gradient. TENG was

used to drive the EVs released from these MNs for 30 min, and then senescent NP cells were treated with these EVs for 72 h. A) Representative western blotting images showing the amount of recombinant TRAM1 protein in engineered EVs released from the MNs, which were left at room temperature with the time gradient. B) Representative western blotting images showing nuclear and cytosolic TREX1 of senescent NP cells after treated with TRAM1-engineered EVs released by the MNs, which were left at room temperature with the time gradient.

Additionally, EVs had the capacity of delivering TRAM1 cargo into NP cytosol might slow down and TRAM1 could exhibit the biological function to regulate NP cell activity due to the NP cellular microenvironment. Actually, IHC staining confirmed that the increase of TRAM1 protein in TENG-responsive eEV-MNs group (Revised Fig. 5H). Radiological and histological staining showed that TRAM1-carrying EVs released from triboelectric-response MNs efficiently the degenerative progression of IVDs (Revised Fig. 5C-F) (Extended Data Fig. 13A-F).

Figure 5 C) Representative X-ray images and EVs in vivo imaging showing the EV release kinetics in coccygeal IVDs at different time points. D) Representative μ CT images of coccygeal IVDs at different time points. E) MR T2-weighted images and Pfirrmann MRI grades of coccygeal 7th-coccygeal 8th (C7-C8) IVDs from rats with indicated treatments for 4, 8, and 12 weeks ($n = 5$). F) SO&FG staining of C7-C8 IVDs from rats with the indicated treatments for 4, 8, and 12 weeks ($n = 5$). H) IHC staining of TRAM1 in C7-C8 IVDs from rats with the indicated treatments for 4, 8 and 12 weeks ($n = 5$). The arrows indicated TRAM1⁺ NP cells. I) IF staining of p-STING in C7-C8 IVDs from rats with the indicated treatments for 4, 8 and 12 weeks ($n = 5$).

Extended Data Fig. 13 A) H&E staining of C7-C8 IVDs from rats with the indicated treatments for 4, 8, 12 weeks (n = 5). B) IF staining of type II collagen in C7-C8 IVDs from rats with the indicated treatments for 4, 8 and 12 weeks (n = 5). C) IHC staining of p-p53 in C7-C8 IVDs from rats with the indicated treatments for 4, 8 and 12 weeks (n = 5). The arrows indicated p-p53⁺ NP cells. D) IF staining of IL-1 β in C7-C8 IVDs from rats with the indicated treatments for 4, 8 and 12 weeks (n = 5). E) Quantitative analysis of the p-p53⁺ NP cell proportion in C7-C8 IVDs from rats with the indicated treatments for 4, 8 and 12 weeks (n = 5). F) Quantitative analysis of the fluorescence intensity of IL-1 β in C7-C8 IVDs from rats with the indicated treatments for 4, 8 and 12 weeks (n = 5).

Minor concerns:

I strongly recommend that the author should rewrite the manuscript carefully, because:

1. The description of Fig. 1B, as well as Extended Data 7 and 8, contains an abundance of repetitive statements, making the presentation unnecessarily lengthy. In particular, the author reiterates certain points multiple times, which could be condensed for a more concise and

effective communication of the findings.

Response:

- Many thanks for the constructive advice of the reviewer. We have corrected some description for a more concise and effective communication.

2. The manuscript lacks explanations for key abbreviations, such as "eEV" and "nEV". Additionally, the values corresponding to the abbreviations "NEMT group", "TEMT group", and "TEM group" are not provided because the authors did not use these abbreviations in the figures. Moreover, there appears to be a mistake in the usage of the "TEM group" in line 647.

Response:

- We appreciated the useful advice of the reviewer. "eEV" means "engineered extracellular vesicle", and "nEV" means "native extracellular vesicle", "NEMT group" means "native EV-carrying MNs connected to the TENG group", "TEMT group" means "TRAM1-engineered EV-carrying MNs connected to the TENG group", and "TEM group" means "TRAM1-engineered EV-carrying MNs without connection to the TENG group". We have given up to use these abbreviations and corrected the full name involved in extracellular vesicles and experimental groups for clarity and readability.

3. In line 322, 326 and 331, there should be Extended Data Fig. 3E, 3F and 3G. Please pretend the similar mistakes.

Response:

- Many thanks for careful review. We are so sorry for the figure citation mistakes and have corrected the relevant mistakes.

4. In Fig. 1a, it seems that the author mistakenly used a human as the experimental subject instead of a rat, which could cause confusion or misinterpretation of the results.

Response:

- Many thank for the insightful advice of the reviewer. Fig. 1a aims to envisage the clinical translation of our results. We are dedicated to fabricate the smart and efficient repair systems for IVDD and low back pain. In this study, we designed exercise self-powered

triboelectric-responsive MNs for controllable EV release in IVDD targeting treatment. Fig. 1a shows our envisagement in clinical application: human spinal motion will provide the mechanic energy, and then wearable TENG converts human mechanic energy to triboelectricity and drives the release of EVs carrying in the MNs, which not only maintains the regular activity of patient but also alleviates IVDD progression and discogenic low back pain.

5. The authors seem to have confused between "PHK26" and "PKH26." It is crucial to conduct a careful review and rectify this inconsistency to ensure the accuracy of the information presented.

Response:

- We appreciate the reviewer for careful review. We are so sorry for our spelling mistakes, and we have corrected the mistakes of this term in our manuscript.

6. There are inconsistencies in the format of references. Such as line 1173, 1186 and 1198, there are three versions of "ACS nano".

Response:

- Many thanks for the useful suggestion of the reviewer. We have corrected the formation of references according to the formation requirement of *Nature Communications*.

7. The authors are expected to include comprehensive materials and methods concerning flow cytometry.

Response:

- Many thanks for the constructive suggestion of the reviewer. We have added the experimental procedures and materials of flow cytometry in the "Materials and Methods" part.

Reviewer #4 (Remarks to the Author):

The manuscript entitled ‘Self-powered triboelectric-responsive microneedles of optogenetically engineered-EV controllable release for intervertebral disc degeneration repair’ by Zhang et al. describes a multi-layered system with which EVs, engineered to carry the TRAM1 protein can be delivered to nucleus pulposus cells, using a self-powered triboelectric-responsive microneedle platform, in order to alleviate degenerative processes.

Response:

- We thank the reviewer so much for the positive comments and valuable suggestions on our work involved in the mechanism of NP cell senescence and IVDD treatment. We are very pleased to further edit this manuscript according to your insightful advice. Based on the comments, we have carefully done some experiments and revised the manuscript with all changes highlighted in red. Please read our point-by-point responses below. Thank again for giving us this chance to revise our study.

Novelty.

To date, EVs from various sources have been used extensively in a multitude of regenerative medicine approaches, including for IVDs (reviewed in doi: 10.3389/fbioe.2020.00311 and doi: 10.1002/adhm.202100596 a.o.). The current work I believe has some novelty in the field, by using an engineered EV type, in-depth characterization of biological samples and a delivery system that may provide some advantages. It would be nice for the authors to also describe whether they observe functional recovery in their animal models.

Response:

- Many thanks for the insightful advice of the reviewer. We are dedicated to fabricate the smart and efficient repair systems for IVDD and low back pain. In this study, we investigated the master regulators which were responsive for the activation of cGAS-STING axis and the senescent phenotypic acquisition of NP cell, and designed exercise self-powered triboelectric-responsive MNs for controllable EV release in IVDD targeting treatment. EVs from various sources have been used extensively in a multitude of regenerative medicine approaches for IVDD, and those previous studies have focused on the application of EVs in IVDD. However, given the EV-based treatment in IVDD, some aspects are needed to be

solved for improving therapeutic efficacy of EVs. Firstly, EVs lacking the desired cargoes have poor ability to target master regulators involved in the degenerative process of IVDs. Additionally, an excellent delivery system for targeting EVs to degenerative sites and responsive triggers for controllable EV release are key considerations when designing EV-based therapeutic strategies to minimize off-target effects and enhance therapeutic efficiency. Therefore, we designed triboelectric-responsive EXPLOR-engineered EV release system for biologically targeted IVDD treatment.

Still, the whole scheme seems aimed at the delivery of TRAM1 protein into coccygeal IVDs. In that respect, the use of a rather complicated system (i.e. EVs) may not have benefit over other protein delivery vehicles or even the 'naked' TRAM1 protein. In itself, the EVs itself seem to have limited function.

Response:

- Many thanks for the insightful advice from the reviewer. To evaluate the therapeutic effects of different delivery strategies including TRAM1 protein injection, lentivirus (LV) transfection and EV-based delivery, we directly injected exogenous TRAM1 protein into rat coccygeal NP tissues or used TRAM1-overexpressing plasmid-carrying LVs to transfect coccygeal NP cells (Extended Data Fig. 15A-E). It was difficult that the exogenously injected TRAM1 protein to be stored in NP tissues and exogenous protein administration failed to deliver TRAM1 into NP cells and alleviate needle picture-induced degeneration of coccygeal IVDs (Extended Data Fig. 15A-E). LV transfection exhibited an efficient delivery for improving the number of TRAM1-positive NP cells and migrating the coccygeal IVDD (Extended Data Fig. 15A-E). Interestingly, via IHC staining, we found that the TRAM1-positive NP cells increased until 8 weeks, and subsequently decreased at 12 weeks, and we inferred the increasement of TRAM1 gene copies had limited efficiency for upregulating the expression of TRAM1 and remodeling the function of TRAM1 because the endogenous etiological factors driving the degenerative progression were not eliminated and the endogenous molecular mechanism for TRAM1 deficiency or dysfunction was still unknown (Extended Data Fig. 15A-E). Similarly, the EV-based strategy exhibited the characters in delivering functional protein to NP cells to overcome the delivery problem of

protein injection and the limited efficiency of LVs in regulating endogenous protein expression (Extended Data Fig. 15D-E).

Extended Data Fig. 15 The therapeutic efficacy of different delivery strategies for alleviating rat coccygeal IVDD.

To compare the delivery efficacy of different delivery strategies, we used exogenous GST-tagged TRAM1 protein solution or TRAM1-overexpressing plasmid-carrying LVs to inject into the NP tissues of needle puncture-induced rat coccygeal IVDD model. A) H&E staining of C7-C8 IVDs from rats with the indicated treatments for 4, 8, 12 weeks (n = 5). B) SO&FG staining of C7-C8 IVDs from rats with the indicated treatments for 4, 8 and 12 weeks (n = 5). B) IHC staining of TRAM1 in C7-C8 IVDs from rats with the indicated treatments for 4, 8 and 12 weeks (n = 5). C) IF staining of GST or GFP in C7-C8 IVDs from rats with the indicated treatments for 4, 8, 12 weeks (n = 5). D) Histological score of C7-C8 IVDs from rats with the indicated treatments for 4, 8, and 12 weeks (n = 5). E) Quantitative analysis of the TRAM1⁺ NP cell proportion in C7-C8 IVDs from rats with the indicated treatments for 4, 8 and 12 weeks (n = 5).

Although the EVs itself seem to have limited function, we utilized the EV-based repair system to exhibit some advantages in alleviating IVDD progression: optogenetical

engineered technology was used to integrate reversible optical-responsive interactions with cargo loading into EVs to target the degenerative mechanism of IVDs. Additionally, self-powered triboelectric-responsive MNs harvested the mechanic energy to efficiently control the release of engineered EVs, and we envisage to achieve the clinical translation of human spinal motion-driven treatment.

Lack of experimental detail provided.

Essential experimental detail is lacking. There is no mention of the EV concentration used in any of the experiments (in vitro nor in vivo). This makes it impossible to tell whether EV numbers used are in any way relevant or achievable. It is not mentioned what number of HEK cells was used for EV production. Again, this makes it impossible to gauge the value of the results provided.

Response:

- Many thanks for the constructive advice from the reviewer. The EV concentration used in Fig. 4A-I and Extended Data Fig. 6 to compare the therapeutic efficiency of native EVs and engineered EVs was 1×10^6 particle/ml; the EV concentration used for electropolymerization in Fig. 4J was 1.1×10^{11} particle/ml; and the EV concentrations released from MNs with different triboelectric stimulation duration in Fig. 4K-N and Extended Data Fig. 10C-L is 0 (0 min), 1.4×10^5 (10 min), 1.1×10^6 (20 min), 1.9×10^8 (30 min), 5×10^8 (40 min), 1.2×10^9 (50 min) particle/ml. It was difficult to measure the accurate concentrations of EVs in rat coccygeal IVDs due to lack of methods, but EV dynamics detected by in vivo imaging confirmed the enrichment and accumulation of native EVs or engineered EVs in rat coccygeal NP tissues, and revealed the release characteristic of EVs from triboelectric-responsive MNs. Additionally, IF staining showed NP cells had similar uptake rates of native EVs and engineered EVs. Multi-experimental technologies have verified that the therapeutic efficiency of TRAM1-engineered EVs in alleviating IVDD.
- 1.0×10^9 HEK-293T cell was used to transfect with CIBN-conjugated EGFP-tagged CD9 and CRY2-conjugated mCherry-tagged TRAM1 plasmids for engineered EV production.

The authors use a transient overexpression in HEK cells of their plasmids. What was the efficacy of transfection? I assume both plasmids need to be taken up by one cell in order for that cell to

generate the EVs with both desired proteins? How many cells (and therefore EVs), did not carry the constructs and could this negatively impact the results?

Response:

➤ We appreciate the insightful advice from the reviewer. In our pre-experiment, the efficacy of double transfection of HEK-293T cells is 93.7% (Response Fig. 3). As the reviewer considered, both plasmids need to be taken up by one cell in order to generate EVs with both desired proteins. To evaluate the efficacy of engineered EVs, we used the native EVs from HEK-293T cells without transfected with CRY2-tagged TRAM1 and CIBN-tagged CD9 plasmids as the negative control. We found the characteristics of EVs including size, morphology, EV-associated markers were similar between native and engineered EVs. NP cells were able to efficiently take up both types of EVs without any particular preference. And native EVs didn't exhibit the capacity to inhibit the degenerative progression of IVDs or damaged effects to accelerate NP cell senescence. Additionally, to avoid the effects of cofounding factors of illumination stimulation and photoreactive proteins, we also verified that CIBN-CD9-engineered EV-carrying MNs system without illumination-induced TRAM1 loading failed to inhibit NP cell senescence or alleviate the degenerative progression of coccygeal IVDs (Extended Data Fig. 14A-N). Correctly, native EVs or CIBN-CD9-engineered EVs didn't negatively impact the results, and the delivery of TRAM1 protein into senescent NP cells was essential for the therapeutic effects of optogenetically engineered EVs for inhibiting NP cell inflammatory response.

Response Fig. 3 The transfection efficiency of HEK-293T cells

Flow cytometer showing the proportion of CD9-EGFP⁺/TRAM1-mCherry⁺ cells.

Extended Data Fig. 14 Illumination-induced TRAM1 loading into EVs was essential for the therapeutic effects of engineered-EV carrying MN system for alleviating needle puncture-induced rat coccygeal IVDD.

A) Isolated engineered EVs from HEK-293T cells after cotransfection with CIBN-EGFP-CD9 and CRY2-mCherry-TRAM1 plasmids with (“Blue light ON”) or without (“Blue light OFF”) 460-nm laser stimulation ($20 \mu\text{W}/\text{cm}^2$) were loaded into triboelectric-responsive MN system. And rat coccygeal IVDs for needle puncture model wearing with different equipments (TENG-responsive “Blue light ON” EV-MNs or TENG-responsive “Blue light OFF” EV-MNs) ($n = 5$) were performed. A) H&E staining of C7-C8 IVDs from rats with the indicated treatments for 4, 8, 12 weeks ($n = 5$). B) SO&FG staining of C7-C8 IVDs from rats with the indicated treatments for 4, 8 and 12 weeks ($n = 5$). C) IHC staining of TRAM1 in C7-C8 IVDs from rats with the indicated

treatments (n = 5) for 4, 8 and 12 weeks. D) IHC staining of p-p53 in C7-C8 IVDs from rats with the indicated treatments for 4, 8 and 12 weeks (n = 5). G) IF staining of PKH26-labelled EVs and mCherry in C7-C8 IVDs from rats with the indicated treatments for 4, 8, 12 weeks (n = 5). E) IF staining of type II collagen in C7-C8 IVDs from rats with the indicated treatments for 4, 8 and 12 weeks (n = 5). F) IF staining of EGFP-CD9 in C7-C8 IVDs from rats with the indicated treatments for 4, 8, 12 weeks (n = 5). G) IF staining of p-STING in C7-C8 IVDs from rats with the indicated treatments for 4, 8 and 12 weeks (n = 5). H) Quantitative analysis of the fluorescence intensity of p-STING in C7-C8 IVDs from rats with the indicated treatments for 4, 8 and 12 weeks (n = 5). I) Histological score of C7-C8 IVDs from rats with the indicated treatments for 4, 8, and 12 weeks (n = 5). J) Quantitative analysis of the TRAM1⁺ NP cell proportion in C7-C8 IVDs from rats with the indicated treatments for 4, 8 and 12 weeks (n = 5). K) Quantitative analysis of the p-p53⁺ NP cell proportion in C7-C8 IVDs from rats with the indicated treatments for 4, 8 and 12 weeks (n = 5). L) Quantitative analysis of EGFP-CD9⁺ NP cell proportions in C7-C8 IVDs from rats with the indicated treatments for 4, 8 and 12 weeks (n = 5). M) Quantitative analysis of the fluorescence intensity of p-STING in C7-C8 IVDs from rats with the indicated treatments for 4, 8 and 12 weeks (n = 5). N) Quantitative analysis of the fluorescence intensity of IL-1 β in C7-C8 IVDs from rats with the indicated treatments for 4, 8 and 12 weeks (n = 5).

Tracking of native and engineered EVs relied on different labelling strategies. The native EVs were labelled with PKH26 or DiO, the engineered expressed mCherry or EGFP. This is a potential concern as it has recently been shown that the label used can affect results (see doi: 10.1021/acsnano.0c09873). A better control for the engineered EVs could be a truncated/non-functional version of TRAM1 to avoid results being influenced by different labelling strategies. This would at the same time solve the potential issue of the Cry2 protein having bioactivity. This is currently not controlled for.

Response:

- Many thanks for the insightful advice from the reviewer. According to current studies, there lacks of the understanding involved in the functional domains of TRAM1, which is sufficient for the binding and regulation of TREX1. Therefore, it's difficult to design the truncated/non-functional version of TRAM1, which failed to remodel the elimination of

TREX1. To solve the tracking of EVs and avoid the confounding factors including illumination stimulation, optogenetic proteins and the other cargoes in EVs, we used engineered EVs collected from double-transfected HEK-293T cells without illumination stimulation as the control, and verified that the lack of illumination stimulation failed to load the TRAM1 into CIBN-CD9-engineered EVs, and the delivery of TRAM1 protein into senescent NP cells was essential for the therapeutic effects of optogenetically engineered EV-carrying MNs for and inhibiting NP cell senescent acquisition in vitro and alleviating the degenerative progression of coccygeal IVDs in vivo (Extended Data Fig. 11) (Extended Data Fig. 14).

Extended Data Fig. 11 Illumination-induced TRAM1 loading into EVs was essential for the

therapeutic effects of engineered EVs for remodeling TREX1 function and inhibiting NP cell senescent phenotypic acquisition.

A) Representative western blotting images of CRY2-mCherry-TRAM1, CIBN-EGFP-CD9 and EV markers in HEK-293T cells and isolated engineered EVs from HEK-293T cells after cotransfection with CIBN-EGFP-CD9 and CRY2-mCherry-TRAM1 plasmids with (“Blue light ON”) or without (“Blue light OFF”) 460-nm laser stimulation ($20 \mu\text{W}/\text{cm}^2$). B) Representative IF staining images of EGFP and mCherry in P8 NP cells after treatment with engineered EVs for 72 h, bar: $20 \mu\text{m}$. C) Flow cytometry images to analyze the uptake of EVs in P8 NP cells after treatment with engineered EVs for 72 h. D) Representative western blotting images of CRY2-mCherry-TRAM1, CIBN-EGFP-CD9 from P8 NP cells after treated with PBS or engineered EVs for 72 h. E) Representative western blotting images of p-p53 and p21 in P8 NP cells after treated with PBS or engineered EVs for 72 h. F) SA- β -gal staining and G) quantitative analysis of SA- β -gal⁺ NP cell proportion in P8 NP cells after treated with PBS, native or engineered EVs for 72 h, bar: $200 \mu\text{m}$. H) Differential expression heatmap of SASP in P8 NP cells after treated with PBS or engineered EVs for 72 h. I) Representative western blotting and J) quantitative analysis of cytosolic and nuclear TREX1 protein in P8 NP cells after treated with PBS or engineered EVs for 72 h. K) Representative agarose gel electrophoresis images and L) quantitative analysis of DNA fragments from the cytosolic component in P8 NP cells after treatment with PBS or engineered EVs for 72 h. M) Representative western blotting images of cGAS, STING and γH2A in P8 NP cells after treated with PBS or engineered EVs for 72 h. N) Representative western blotting images of CRY2-mCherry-TRAM1 protein from P8 NP cells after cultured in the medium with engineered EV-carrying MNs connected to TENG with different triboelectric stimulation durations. O) Representative electrophoresis images of DNA fragments from the cytosolic component in P8 NP cells after culture in medium with engineered EV-carrying MNs connected to TENGs with different triboelectric stimulation durations. P) Representative western blotting images of cGAS, STING and γH2A in P8 NP cells after cultured in the medium with engineered EV-carrying MNs connected to TENG with different triboelectric stimulation durations. Q) Representative western blotting images of p-p53 and p21 in P8 NP cells after cultured in the medium with engineered EV-carrying MNs connected to TENG with different triboelectric stimulation durations. R) Differential expression heatmap of SASP in P8 NP cells

after cultured in the medium with engineered EV-carrying MNs connected to TENG with different triboelectric stimulation durations.

Extended Data Fig. 14 Illumination-induced TRAM1 loading into EVs was essential for the therapeutic effects of engineered-EV carrying MN system for alleviating needle puncture-induced rat coccygeal IVDD.

A) Isolated engineered EVs from HEK-293T cells after cotransfection with CIBN-EGFP-CD9 and CRY2-mCherry-TRAM1 plasmids with (“Blue light ON”) or without (“Blue light OFF”) 460-nm laser stimulation ($20 \mu\text{W}/\text{cm}^2$) were loaded into triboelectric-responsive MN system. And rat coccygeal IVDs for needle puncture model wearing with different equipments (TENG-responsive “Blue light ON” EV-MNs or TENG-responsive “Blue light OFF” EV-MNs) ($n = 5$) were performed. A) H&E staining of C7-C8 IVDs from rats with the indicated treatments for 4, 8, 12

weeks (n = 5). B) SO&FG staining of C7-C8 IVDs from rats with the indicated treatments for 4, 8 and 12 weeks (n = 5). C) IHC staining of TRAM1 in C7-C8 IVDs from rats with the indicated treatments (n = 5) for 4, 8 and 12 weeks. D) IHC staining of p-p53 in C7-C8 IVDs from rats with the indicated treatments for 4, 8 and 12 weeks (n = 5). G) IF staining of PKH26-labelled EVs and mCherry in C7-C8 IVDs from rats with the indicated treatments for 4, 8, 12 weeks (n = 5). E) IF staining of type II collagen in C7-C8 IVDs from rats with the indicated treatments for 4, 8 and 12 weeks (n = 5). F) IF staining of EGFP-CD9 in C7-C8 IVDs from rats with the indicated treatments for 4, 8, 12 weeks (n = 5). G) IF staining of p-STING in C7-C8 IVDs from rats with the indicated treatments for 4, 8 and 12 weeks (n = 5). H) Quantitative analysis of the fluorescence intensity of p-STING in C7-C8 IVDs from rats with the indicated treatments for 4, 8 and 12 weeks (n = 5). I) Histological score of C7-C8 IVDs from rats with the indicated treatments for 4, 8, and 12 weeks (n = 5). J) Quantitative analysis of the TRAM1⁺ NP cell proportion in C7-C8 IVDs from rats with the indicated treatments for 4, 8 and 12 weeks (n = 5). K) Quantitative analysis of the p-p53⁺ NP cell proportion in C7-C8 IVDs from rats with the indicated treatments for 4, 8 and 12 weeks (n = 5). L) Quantitative analysis of EGFP-CD9⁺ NP cell proportions in C7-C8 IVDs from rats with the indicated treatments for 4, 8 and 12 weeks (n = 5). M) Quantitative analysis of the fluorescence intensity of p-STING in C7-C8 IVDs from rats with the indicated treatments for 4, 8 and 12 weeks (n = 5). N) Quantitative analysis of the fluorescence intensity of IL-1 β in C7-C8 IVDs from rats with the indicated treatments for 4, 8 and 12 weeks (n = 5).

Potential issues with the EV prep used.

The authors should consider declaring whether they adhere to the MISEV guidelines and what the purity of their EV prep is. As they have used differential centrifugation with no additional purification step to isolate their EVs, there is significant risk of the presence of other, non-EV associated secreted proteins. Furthermore, with EV-depleted serum additional carry-over of proteins is an inherent risk that needs to be assessed, but currently isn't. Involvement of non-EV factors cannot be excluded completely in my opinion.

Response:

- Many thanks for pointing it out. We applied EXPLOR technology to generate engineered EVs according to the previous studies (Sheller-Miller S, et al. *Sci Adv* 7, (2021).; Kim S, et al.

Kidney Int 100, 570-584 (2021).; Yim N, et al. *Nat Commun.* 7, 12277 (2016)). According to the experimental protocol published in the MISEV guidelines, the differential centrifugation method was used to isolate EVs, and a force of around $20000 \times g$ was applied for between 10 and 90 min to enrich putatively larger/denser EVs. The suspended EVs were then filtered through a syringe filter (0.22 μm , Sartorius, Germany). Some experiment procedures have been added into the “Methods and Materials” part.

To avoid the confounding factors including cargoes carrying in EVs, illumination stimulation, optogenetic proteins, non-EV associated secreted proteins and other non-EV factors, we used native EVs and engineered EVs derived from plasmid-transfected cells without illumination stimulation as the control, and confirmed the TRAM1 loading was sufficient for the therapeutic efficiency of engineered EVs (Fig. 4C-N, Fig. 5A-N) (Extended Data Fig. 6A-F, Extended Data Fig. 11A-Q, Extended Data Fig. 14A-N).

Minor issues.

In line 511, it is not clear to me whose parts of the body the authors are referring to. Are these the animals (rats) that were used?

Response:

- We appreciate the reviewer for pointing it out. We used different motion formations of experiments' bodies including beathing, clenching, shaking and bending, to drive TENG for evaluating the output of wearable TENG, and we aimed to confirm TENG had the capacity of converting mechanic energy from human motion to triboelectricity.

In line 616, Fig. 6A and 6B are referenced, but I don't think these exist.

Response:

- Many thanks for careful reviewing from the reviewer. We are so sorry for our citation mistake, and we have corrected the figure citation to Fig. 5A, 5B.

Nomenclature could be more consistent. E.g. Use TEMT and NEMT also in the labels in figure 5.

Response:

- We appreciated the useful advice of the reviewer. “NEMT group” means “native

EV-carrying MNs connected to the TENG group” and “TEMT group” means “TRAM1-engineered EV-carrying MNs connected to the TENG group”. We have given up to use these abbreviations and corrected the full name involved in experimental groups for clarity and readability.

Amount of data presented in one manuscript.

Though the work reflects an impressive amount of work, reading the paper, I found it challenging to fully comprehend all the information that was provided. It is a complex manuscript with several elements, that makes it challenging to see what the main message and novelty is. I sincerely wonder whether the authors may have put too much information in the main body of the manuscript.

Response:

➤ We appreciated the reviewer for your efforts and careful review. Many thanks for the constructive advice and insightful comments. Given the research mode of “phenotype-mechanism-treatment”, our work showed the molecular mechanism of IVDD and a promising strategy for targeting master regulators involved in IVDD, which integrated high-throughput multi-omics analysis, optogenetic technology, MN-based EV delivery and wearable TENG-driven therapeutic system.

Recently, studies on IVDD have emphasized the senescence of resident cell populations and aberrant cell-mediated response to progressive IVD tissue integrity destruction and senescence-associated phenotype alteration in NP cells. In this study, we verified that exercise promoted the senescent phenotypic acquisition of NP cells via excessive cytosolic damaged DNA accumulation and activation of cGAS-STING axis, and accelerated the degeneration of lumbar IVDs from surgical lumbar instability, similar to the clinical degeneration process. TREX1 serves as an ER-associated exonuclease that protects cells from excessive cytosolic accumulated DNA and inhibits the activation of cGAS-STING axis, which may be a promising therapeutic target to block activation of the inflammatory response during the IVDD process. However, the administration of TREX1 wide-type plasmids increased the level of genomic DNA damage in senescent NP cells but not promoted the elimination of cytosolic damaged DNA. To further investigate the mechanism involved in the

dysfunction of TREX1, we performed endogenous TREX1 interactomics from normal and senescent NP cells, and revealed that the disassembly of the TREX1-TRAM1 complex promoted the deanchoring in ER membrane and nuclear translocation of TREX1, which results in the exonuclease activity of TREX1 destroying genomic integrity. Summarily, TRAM1 is a promising target for remodeling the function of TREX1 and inhibiting the inflammatory response of senescent NP cells during progressive IVDD. An appropriately selective vector is necessary for an excellent gene and drug delivery system to deliver target molecules stably to a specific site. EVs, with their suitable biocompatibility, minimal immunogenicity and intrinsic capacity to target tissues, perform a critical function in transferring active molecules to recipient cells, which has been highlighted as a promising strategy for the delivery of proteins and nucleotides in disease-targeting therapeutics. Therefore, we used EXPLOR technology to load TRAM1 protein to engineered EVs and fabricated triboelectric-responsive MNs to delivery therapeutic EVs for minimizing off-target effects and enhancing molecular targeting efficiency.

Language.

The manuscript contains numerous typos in various places (also in the axes of the figures), which should be addressed. English language should also be improved. In some instances it even hinders understanding with the reader. The abstract comes across as quite incoherent and could do with additional connection between the different topics.

Response:

➤ We appreciate the reviewer for reviewing our manuscript carefully. We are so sorry for typos in our manuscript, and we have read through the manuscript and corrected the typos.

Our work contained several elements involved in molecular biology, EVs and biomaterials, and the reader might be unfamiliar with some concepts. We have invited the professional team from *Spring Press* to improve our English language, and rephrased some descriptions in the manuscript, which might make our work understood by any scientists.

According to the insightful advice of the reviewer, in the revised abstract, we have rephrased the abstract and focused on the mechanism research and current therapeutic progress of IVDD, and showed our work about IVDD mechanism and targeting therapeutic strategy.

Overall, I think publication in its current form would be premature for this manuscript.

Response:

- We sincerely appreciate the reviewer for efforts and constructive comments, and we are very happy to revise the manuscript according to the insightful advice. We have done some experiments carefully to support these concerns. Thanks again for the reviewer's effort.

REVIEWERS' COMMENTS

Reviewer #1 (Remarks to the Author):

In this manuscript, Zhang et. al. fabricated a self-powered triboelectric-responsive microneedle array driving the controllable release of optogenetically engineered extracellular vesicles (EVs) for intervertebral disc degeneration (IVDD) repair. The self-powered microneedle device combines wearable triboelectric nanogenerator (TENG) with engineered EVs-based targeting molecular modulation, aiming to block etiology-associated IVD degenerative process. Authors revealed the degenerative-associated molecular atlas involved in nucleus pulposus cell senescence via the analysis of transcriptomics, interactomics and gain-/loss-of-function experiment and designed an optogenetically controlled EVs-targeting treatment strategy. Overall, this study is a creative and interesting with a significance of transitional medicine, and the experiments are meticulously performed.

The authors have revised the manuscript carefully according to the reviewer's comments. The quality of this work is greatly improved. The reviewer is satisfied with the revision and has no more other suggestion for this work. I think that this manuscript should be accepted by the journal of Nature Communications.

Reviewer #2 (Remarks to the Author):

Thank you to the authors to have improved the manuscript accordingly with reviewer's suggestions. My recommendation is that the manuscript can now be published.

Reviewer #3 (Remarks to the Author):

They have addressed all my concerns. Thanks for the revision work. I have no more concerns.

Reviewer #4 (Remarks to the Author):

The authors have addressed my concerns sufficiently and I would endorse this manuscript for publication.

I thank the authors for their efforts.